# NON-NEGATIVE BREGMAN DIVERGENCE MINIMIZATION FOR DEEP DIRECT DENSITY RATIO ESTIMATION

## ABSTRACT

This paper aims to estimate the ratio of probability densities using flexible models, such as state-of-the-art deep neural networks. The density ratio estimation (DRE) has garnered attention as the *density ratio* is useful in various machine learning tasks, such as anomaly detection and domain adaptation. For estimating the density ratio, methods collectively known as direct DRE have been explored. These methods are based on the minimization of the *Bregman* (BR) divergence between a density ratio model and the true density ratio. However, when using *flexible models*, such as deep neural networks, existing direct DRE suffers from serious *train-loss hacking*, which is a kind of *over-fitting* caused by the form of an empirical risk function. In this paper, we introduce a *non-negative correction* for empirical risk using only the prior knowledge of the *upper bound* of the density ratio. This correction makes a DRE method robust against train-loss hacking. It enables the use of flexible models, such as state-of-the-art deep neural networks. In the theoretical analysis, we show the generalization error bound of the BR divergence minimization. In our experiments, the proposed methods show favorable performance in inlier-based outlier detection and covariate shift adaptation.

## 1 INTRODUCTION

The density ratio estimation (DRE) problem has attracted a great deal of attention as an essential task in data science for its various industrial applications, such as domain adaptation (Shimodaira, 2000; Plank et al., 2014; Reddi et al., 2015), learning with noisy labels (Liu & Tao, 2014; Fang et al., 2020), anomaly detection (Smola et al., 2009; Hido et al., 2011; Abe & Sugiyama, 2019), two-sample testing (Keziou & Leoni-Aubin, 2005; Kanamori et al., 2010; Sugiyama et al., 2011a), causal inference (Kato et al., 2020), change point detection in time series (Kawahara & Sugiyama, 2009), and binary classification only from positive and unlabeled data (PU learning; Kato et al., 2019). For example, anomaly detection is not easy to perform based on standard machine learning methods such as binary classification since anomalous data is often scarce, but it can be solved by estimating the density ratio when training data without anomaly as well as unlabeled test data are available (Hido et al., 2008).

Among various approaches for DRE, we focus on the Bregman (BR) divergence minimization framework (Bregman, 1967; Sugiyama et al., 2011b) that is a generalization of various DRE methods, e.g., the moment matching (Huang et al., 2007; Gretton et al., 2009), the probabilistic classification (Qin, 1998; Cheng & Chu, 2004), the density matching (Nguyen et al., 2010; Yamada et al., 2010), and the density-ratio fitting (Kanamori et al., 2009). Recently, Kato et al. (2019) also proposed using the risk of PU learning for DRE, which also can be generalized from the BR divergence minimization viewpoint, as we show below.

However, existing DRE methods mainly adopt a linear-in-parameter model for nonparametric DRE (Kanamori et al., 2012) and rarely discussed the use of more flexible models, such as deep neural networks, while recent developments in machine learning suggest that deep neural networks can significantly improve the performances for various tasks, such as computer vision (Krizhevsky et al., 2012) and natural language processing (Bengio et al., 2001). This motivates us to use deep neural networks for DRE. However, existing DRE studies have not fully discussed using such state-of-the-art deep neural networks. For instance, although Nam & Sugiyama (2015) and Abe & Sugiyama (2019) proposed using neural networks for DRE, their neural networks are simple and shallow.

When using deep neural networks in combination with empirical minimization of BR divergence, we often observe a serious *over-fitting* problem as demonstrated through experiments in Figure 2 of Section 5. We hypothesize that this is mainly because there is no lower bound in the empirically BR divergence approximating by finite samples, i.e., we can achieve an infinitely negative value in minimization. This hypothesis is based on Kiryo et al. (2017), which reports a similar problem in PU learning. While Kiryo et al. (2017) call this phenomena *over-fitting*, we refer to it as *train-loss hacking* because the nuance is a bit different from the standard meaning of overfitting.

Here, we briefly introduce the train-loss hacking discussed in the PU learning literature Kiryo et al. (2017). In a standard binary classification problem, we train a classifier $\psi$ by minimizing the following empirical risk using $\{(y_i, X_i)\}_{i=1}^n$:

$$\frac{1}{n} \sum_{i=1}^n \mathbb{1}[y_i = +1]\ell(\psi(X_i)) + \frac{1}{n} \sum_{i=1}^n \mathbb{1}[y_i = -1]\ell(-\psi(X_i)), \tag{1}$$

where $y_i \in \{\pm 1\}$ is a binary label, $X_i$ is a feature, and $\ell$ is a loss function. On the other hand, in PU learning formulated by du Plessis et al. (2015), because we only have positive data $\{(y_i' = +1, X_i')\}_{i=1}^{n'}$ and unlabeled data $\{(\boldsymbol{x}_j'')\}_{j=1}^{n''}$, we minimize the following alternative empirical risk:

$$\frac{\pi}{n'} \sum_{i=1}^{n'} \ell(\psi(X_i')) \underbrace{- \frac{\pi}{n'} \sum_{i=1}^{n'} \ell(-\psi(X_i'))}_{\text{Cause of train-loss hacking.}} + \frac{1}{n''} \sum_{j=1}^{n''} \ell(-\psi(X_j'')), \tag{2}$$

where $\pi$ is a hyper-parameter representing $p(y = +1)$. Note that the empirical risk (2) is unbiased to the population binary classification risk (1) (du Plessis et al., 2015). While the the empirical risk (1) of the standard binary classification is lower bounded under an appropiate choise of $\ell$, the empirical risk (2) of PU learning proposed by du Plessis et al. (2015) is not lower bounded owing to the existence of the second term. Therefore, if a model is sufficiently flexible, we can significantly minimize the empirical risk only by minimizing the second term $-\frac{\pi}{n'} \sum_{i=1}^{n'} \ell(-\psi(X_i'))$ without increasing the other terms. Kiryo et al. (2017) proposed non-negative risk correction for avoiding this problem when using neural networks. We discuss this problem again in Section 2 and Figure 1.

In existing DRE literature, this train-loss hacking has rarely been discussed, although we often face this problem when using neural networks, as mentioned in Section 5. One reason for this is that the existing method assumes a linear-in-parameter model for a density ratio model (Kanamori et al., 2012), which is not so flexible as neural networks and do not cause the phenomenon.

To mitigate the train-loss hacking, we propose a general procedure to modify the empirical BR divergence using the prior knowledge of the upper bound of the density ratio. Our idea of the correction is inspired by Kiryo et al. (2017). However, their idea of non-negative correction is only immediately applicable to the binary classification; thus we require a non-trivial rewriting of the BR divergence to generalize the approach to our problem. We call the proposed empirical risk the *non-negative BR* (nnBR) divergence, and it is a generalization of the method of Kiryo et al. (2017). In addition, for a special case of DRE, we can still use a lower bounded loss for DRE (See bounded uLSIF and BKL introduced in the following section). However, such a loss also suffers from the train-loss hacking (bounded uLSIF of Figure 2 and BKL-NN of Figure 4). In the case, the train-loss hacking is caused because the loss sticks to the lower bound. This type of train-loss hacking is also avoided by using the proposed nnBR divergence.

Our main contributions are: (1) the proposal of a general procedure to modify a BR divergence to enable DRE with flexible models, (2) theoretical justification of the proposed estimator, and (3) the experimental validation of the proposed method using benchmark data.

## 2 PROBLEM SETTING

Let $\mathcal{X}^{\mathrm{nu}} \subseteq \mathbb{R}^d$ and $\mathcal{X}^{\mathrm{de}} \subseteq \mathbb{R}^d$ be the spaces of the $d$-dimensional *covariates* $\{X_i^{\mathrm{nu}}\}_{i=1}^{n_{\mathrm{nu}}}$ and $\{X_i^{\mathrm{de}}\}_{i=1}^{n_{\mathrm{de}}}$, respectively, which are independent and identically distributed (i.i.d.) as $\{X_i^{\mathrm{nu}}\}_{i=1}^{n_{\mathrm{nu}}} \overset{\mathrm{i.i.d.}}{\sim} p_{\mathrm{nu}}(X)$ and $\{X_i^{\mathrm{de}}\}_{i=1}^{n_{\mathrm{de}}} \overset{\mathrm{i.i.d.}}{\sim} p_{\mathrm{de}}(X)$, where $p_{\mathrm{nu}}(X)$ and $p_{\mathrm{de}}(X)$ are probability densities over $\mathcal{X}^{\mathrm{nu}}$

Table 1: Summary of methods for DRE(Sugiyama et al., 2011b). For PULogLoss, we use $C < \frac{1}{\overline{R}}$.

| Method | $f(t)$ | Reference |
|---|---|---|
| LSIF | $(t-1)^2/2$ | Kanamori et al. (2009) |
| Kernel Mean Matching | $(t-1)^2/2$ | Gretton et al. (2009) |
| UKL | $t\log(t) - t$ | Nguyen et al. (2010) |
| KLIEP | $t\log(t) - t$ | Sugiyama et al. (2008) |
| Logistic Regression (BKL) | $t\log(t) - (1+t)\log(1+t)$ | Hastie et al. (2001) |
| PULogLoss | $C\log(1-t) + Ct\left(\log(t) - \log(1-t)\right)$ for $0 < t < 1$ | Kato et al. (2019) |

and $\mathcal{X}^{\mathrm{de}}$, respectively. Here, "nu" and "de" indicate the numerator and the denominator. Our goal is to estimate the density ratio $r^*(X) = \frac{p_{\mathrm{nu}}(X)}{p_{\mathrm{de}}(X)}$. To identify the density ratio, we assume the following:

**Assumption 1.** The density $p_{\mathrm{nu}}(X)$ is strictly positive over the space $\mathcal{X}^{\mathrm{nu}}$, the density $p_{\mathrm{de}}(X)$ is strictly positive over the space $\mathcal{X}^{\mathrm{de}}$, and $\mathcal{X}^{\mathrm{nu}} \subseteq \mathcal{X}^{\mathrm{de}}$. In addition, the density ratio $r^*$ is bounded from above on $\mathcal{X}^{\mathrm{de}}$: $\overline{R} = \sup_{X \in \mathcal{X}^{\mathrm{de}}} r^*(X) < \infty$.

Note that the assumption $\mathcal{X}^{\mathrm{nu}} \subseteq \mathcal{X}^{\mathrm{de}}$ is typical in the context of DRE. For instance, in anomaly detection with unlabeled test data, $\mathcal{X}^{\mathrm{de}}$ corresponds to a sample space including clean and anomaly data and $\mathcal{X}^{\mathrm{de}}$ corresponds to a sample space only with clean data. Here, we introduce the notation of this paper. Let $\mathbb{E}_{\mathrm{nu}}$ and $\mathbb{E}_{\mathrm{de}}$ denote the expectations over $p_{\mathrm{nu}}(X)$ and $p_{\mathrm{de}}(X)$, respectively. Let $\hat{\mathbb{E}}_{\mathrm{nu}}$ and $\hat{\mathbb{E}}_{\mathrm{de}}$ denote the sample average over $\left\{X_i^{\mathrm{nu}}\right\}_{i=1}^{n_{\mathrm{nu}}}$ and $\left\{X_i^{\mathrm{de}}\right\}_{i=1}^{n_{\mathrm{de}}}$, respectively. Let $\mathcal{H} \subset \{r : \mathbb{R}^d \to (b_r, B_r)\}$ be the hypothesis class of the density ratio, where $0 \le b_r < \overline{R} < B_r$.

### 2.1 DENSITY RATIO MATCHING UNDER THE BREGMAN DIVERGENCE

A naive way to implement DRE would be to estimate the numerator and the denominator densities separately and take the ratio. However, according to Vapnik's principle, we should avoid solving a more difficult intermediate problem than the target problem (Vapnik, 1998). Therefore, various methods for directly estimating the density ratio model have been proposed (Gretton et al., 2009; Sugiyama et al., 2008; Kanamori et al., 2009; Nguyen et al., 2010; Yamada et al., 2010; Kato et al., 2019). Sugiyama et al. (2011b) showed that these methods can be generalized as the *density ratio matching under the BR divergence*.

The BR divergence is an extension of the Euclidean distance to a class of divergences that share similar properties (Bregman, 1967). Formally, let $f : (b_r, B_r) \to \mathbb{R}$ be a twice continuously differentiable convex function with a bounded derivative. Then, the point-wise BR divergence associated with $f$ from $t^*$ to $t$ is defined as $\ddot{\mathrm{BR}}_f(t^* \| t) := f(t^*) - f(t) - \partial f(t)(t^* - t)$, where $\partial f$ is the derivative of $f$. Now, the discrepancy from the true density ratio function $r^*$ to a density ratio model $r$ is measured by integrating the point-wise BR divergence as follows (Sugiyama et al., 2011b):

$$\ddot{\mathrm{BR}}_f(r^* \| r) := \int p_{\mathrm{de}}(X)\big(f(r^*(X)) - f(r(X)) - \partial f(r(X))\{r^*(X) - r(X)\}\big)dX. \quad (3)$$

We estimate the density ratio by finding a function $r$ that minimizes the BR divergence defined in (3). Here, we subtract the constant $\overline{\mathrm{BR}} = \mathbb{E}_{\mathrm{de}}\big[f(r^*(X))\big]$ from (3) to obtain

$$\mathrm{BR}_f(r^* \| r) := \int p_{\mathrm{de}}(X)\Big(\partial f(r(X))r(X) - f(r(X))\Big)dX - \int p_{\mathrm{nu}}(X)\partial f(r(X))dX. \quad (4)$$

Here, Sugiyama et al. (2012) used $r^*(X)p_{\mathrm{de}} = p_{\mathrm{nu}}$ for removing $r^*(X)$, which is a common technique in the DRE literature. Since $\overline{\mathrm{BR}}$ is constant with respect to $r$, we have $\arg\min_r \mathrm{BR}'_f(r^* \| r) = \arg\min_r \mathrm{BR}_f(r^* \| r)$. Then, let us define the sample analogue of (4) as

$$\widehat{\mathrm{BR}}_f(r) := \hat{\mathbb{E}}_{\mathrm{de}}\Big[\partial f\big(r(X_i)\big)r(X_i) - f\big(r(X_i)\big)\Big] - \hat{\mathbb{E}}_{\mathrm{nu}}\Big[\partial f\big(r(X_j)\big)\Big]. \quad (5)$$

For a hypothesis class $\mathcal{H}$, we estimate the density ratio by solving $\min_{r \in \mathcal{H}} \widehat{\mathrm{BR}}_f(r^* \| r)$.

### 2.2 EXAMPLES OF DRE

Sugiyama et al. (2011b) showed that various DRE methods can be unified from the viewpoint of BR divergence minimization. Furthermore, Menon & Ong (2016) showed an equivalence between conditional probability estimation and DRE by BR divergence minimization. In addition,

we can derive a novel method for DRE from the proposed method of du Plessis et al. (2015) and Kato et al. (2019). We summarize the DRE methods in Table 1. Here, the empirical risks of *least-square importance fitting* (LSIF), the *Kullback–Leibler importance estimation procedure* (KLIEP), logistic regression (LR), and *PU learning with log Loss* (PULogLoss) is given as $\widehat{\mathrm{BR}}_{\mathrm{LSIF}}(r) := -\hat{\mathbb{E}}_{\mathrm{nu}}[r(X_j)] + \frac{1}{2}\hat{\mathbb{E}}_{\mathrm{de}}[(r(X_i))^2]$, $\widehat{\mathrm{BR}}_{\mathrm{UKL}}(r) := \hat{\mathbb{E}}_{\mathrm{de}}[r(X_i)] - \hat{\mathbb{E}}_{\mathrm{nu}}\left[\log\left(r(X_j)\right)\right]$, $\widehat{\mathrm{BR}}_{\mathrm{BKL}}(r) := -\hat{\mathbb{E}}_{\mathrm{de}}\left[\log\left(\frac{1}{1+r(X_i)}\right)\right] - \hat{\mathbb{E}}_{\mathrm{nu}}\left[\log\left(\frac{r(X_j)}{1+r(X_j)}\right)\right]$, and $\widehat{\mathrm{BR}}_{\mathrm{PU}}(r) := -\hat{\mathbb{E}}_{\mathrm{de}}\left[\log\left(1 - r(X_i)\right)\right] + C\hat{\mathbb{E}}_{\mathrm{nu}}\left[-\log\left(r(X_j)\right) + \log\left(1 - r(X_j)\right)\right]$, where $0 < C < \frac{1}{R}$. Here, $\widehat{\mathrm{BR}}_{\mathrm{LSIF}}(r)$ and $\widehat{\mathrm{BR}}_{\mathrm{PU}}(r)$ correspond to LSIF and PULogLoss, respectively. We can derive the KLIEP and LR from $\widehat{\mathrm{BR}}_{\mathrm{UKL}}(r)$ and $\widehat{\mathrm{BR}}_{\mathrm{BKL}}(r)$, which are called unnormalized Kullback–Leibler (UKL) divergence and binary Kullback–Leibler (BKL) divergence, respectively (Sugiyama et al., 2011b). In $\widehat{\mathrm{BR}}_{\mathrm{PU}}(r)$, we restrict the model as $r \in (0, 1)$ and we obtain an estimator of $Cr^*$ as a result of the minimization of the risk. Details of the existing methods are shown in Appendix A.

# 3 DEEP DIRECT DRE BASED ON NON-NEGATIVE RISK ESTIMATOR

In this section, we develop methods for DRE with flexible models such as neural networks.

## 3.1 DIFFICULTIES OF DRE USING NEURAL NETWORKS

Training neural networks for DRE by minimizing the empirical BR divergence tends to suffer from an overfitting phenomenon. For example, we show in Section 5 that the LSIF with neural networks (Nam & Sugiyama, 2015) suffers from a serious overfitting issue.

One possible cause of overfitting of flexible models has been identified to be the train-loss hacking as shown by Kiryo et al. (2017) in the context of PU learning (Figure 1). In DRE, the train-loss hacking is also sensible: the term $-\hat{\mathbb{E}}_{\mathrm{nu}}\left[\partial f\left(r(X_j)\right)\right]$ can easily diverge to $-\infty$ if we try to minimize empirical BR divergence (5) over a highly flexible hypothesis class $\mathcal{H}$. This is because the term is not lower bounded, unlike the loss functions for other machine learning tasks such as classification. Here, note that the other term $\hat{\mathbb{E}}_{\mathrm{de}}\left[\partial f\left(r(X_i)\right)r(X_i) - f\left(r(X_i)\right)\right]$ does not introduce a sufficient trade-off to prevent the divergence to $-\infty$ when the hypothesis class is highly flexible (Figure 1).

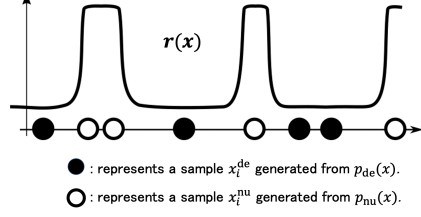

$r(x)$

● : represents a sample $x_i^{\mathrm{de}}$ generated from $p_{\mathrm{de}}(x)$.

○ : represents a sample $x_i^{\mathrm{nu}}$ generated from $p_{\mathrm{nu}}(x)$.

Figure 1: Illustration of the train-loss hacking phenomenon. Given finite data points, a sufficiently flexible model $r$ can easily make the training loss, e.g., $\widehat{\mathrm{BR}}_{\mathrm{LSIF}}(r)$, diverge to $-\infty$.

## 3.2 NON-NEGATIVE BR DIVERGENCE

Although DRE using flexible models suffers from serious train-loss hacking, we still have a strong motivation to use those models for analyzing data such as computer vision and text data. For example, Nam & Sugiyama (2015) and Abe & Sugiyama (2019) approximated the density ratio with neural networks, and Uehara et al. (2016) applied direct DRE for generative adversarial nets (GANs; Goodfellow et al., 2014), both by minimizing the empirical BR divergence (5). To alleviate the train-loss hacking problem, we propose the *non-negative BR divergence estimator*. The proposed method is inspired by Kiryo et al. (2017), which suggested a non-negative correction to the empirical risk of PU learning based on the knowledge that a part of the population risk is non-negative. On the other hand, in DRE, it is not obvious how to correct the empirical risk because we do not know which part of the population risk (4) is non-negative. However, this can be alleviated by determining the upper bound $\overline{R}$ of the density ratio $r^*$. With the prior knowledge of $\overline{R}$, we can determine part of the risk of DRE (4) is non-negative and conduct a non-negative correction to the empirical risk (5) based on the non-negativity of the population risk. Let us define $\tilde{f}$ to be a function such that $\partial f(t) = C\left(\partial f(t)t - f(t)\right) + \tilde{f}(t)$, where $0 < C < \frac{1}{R}$ and put the following assumption.

**Assumption 2.** Assume that $\tilde{f}(t)$ is bounded from above, and that there exists a constant $A$ such that $\partial f(t)t - f(t) + A \geq 0$ for $t \in (b_r, B_r)$.

Assumption 2 is satisfied by most of the loss functions which appear in the previously proposed DRE methods (see Appendix B for examples). Under Assumption 2, because $\tilde{f}(t)$ is bounded above, the train-loss hacking $-\hat{\mathbb{E}}_{\mathrm{nu}}\left[\partial f\left(r(X_j)\right)\right] \to -\infty$ in the minimization of the empirical risk (5) is caused by $-\hat{\mathbb{E}}_{\mathrm{nu}}\left[C\left\{\partial f(r(X_j))r(X_j) - f(r(X_j))\right\}\right] \to -\infty$ because

$$\underbrace{-\hat{\mathbb{E}}_{\mathrm{nu}}\left[\partial f\left(r(X_j)\right)\right]}_{\to -\infty} = \underbrace{-\hat{\mathbb{E}}_{\mathrm{nu}}\left[\tilde{f}(r(X_j))\right]}_{\text{Bounded}} \underbrace{-\hat{\mathbb{E}}_{\mathrm{nu}}\left[C\left\{\partial f(r(X_j))r(X_j) - f(r(X_j))\right\}\right]}_{\to -\infty}.$$

Thus, we succeeded in identifying the part of the empirical risk causing the train-loss hacking. Therefore, by preventing the term from going to infinite negative, we can avoid the train-loss hacking. For achieving this purpose, we impose a non-negative correction to the empirical risk (5); that is, under Assumption 2, we restrict the behavior of the problematic term. To incorporate the assumption, we first rewrite the population risk (4) as $\mathrm{BR}_f(r^*\|r) =$

$$\underbrace{\int \left\{p_{\mathrm{de}}(X) - Cp_{\mathrm{nu}}(X)\right\}\left\{\partial f(r(X))r(X) - f(r(X)) + A\right\}dX}_{(*)} - \int p_{\mathrm{nu}}(X)\left\{\tilde{f}(r(X))\right\}dX.$$

Note that we introduce a constant $A$, which is irrelevant to the original optimization problem. Thus, the problematic term is incorporated into $(*)$. Therefore, we next consider a constraint condition on the term $(*)$. Here, let us define $\ell_1(t) := \partial f(t)t - f(t) + A$, and $\ell_2(t) := -\tilde{f}(t)$. In the above equation, since Assumption 2 implies $\ell_1(t) \geq 0$, and $0 < C < \frac{1}{\overline{R}}$ and $\frac{p_{\mathrm{nu}}(X)}{p_{\mathrm{de}}(X)} \leq \overline{R}$ imply $p_{\mathrm{de}}(X) - Cp_{\mathrm{nu}}(X) > 0$ for all $X \in \mathcal{X}$, we have $\int \left\{p_{\mathrm{de}}(X) - Cp_{\mathrm{nu}}(X)\right\}\left\{\ell_1(r(X))\right\}dX > 0$. Motivated by this inequality, we propose an empirical risk with the non-negative correction as

$$\widehat{\mathrm{nnBR}}_f(r) := \hat{\mathbb{E}}_{\mathrm{nu}}\left[\ell_2(r(X_j))\right] + \left(\hat{\mathbb{E}}_{\mathrm{de}}\left[\ell_1(r(X_i))\right] - C\hat{\mathbb{E}}_{\mathrm{nu}}\left[\ell_1(r(X_j))\right]\right)_+,$$

where $(\cdot)_+ := \max\{0, \cdot\}$. Note that the constraint on $(*)$ is not broken in population (infinite samples) but broken in finite samples with causing the train-loss hacking. Our *deep direct DRE* (D3RE) is based on minimizing $\widehat{\mathrm{nnBR}}_f(r)$.

**Remark 1** (Choice of $C$). In practice, selecting the hyper-parameter $C$ does not require accurate knowledge of $\overline{R}$ because any $0 < C < 1/\overline{R}$ is sufficient to justify the non-negative correction. However, selecting $C$ that is relatively much smaller than $1/\overline{R}$ may damage the empirical performance. See Section G.1.1. This remark does not mean that $1/\overline{R}$ should not be small; that is, $\overline{R}$ is note be small. If $1/\overline{R}$ is small, $C$ also can be small. Therefore, we recommend use larger $C$ more than smaller $C$.

**nnBR Divergence with Existing Methods:** The above strategy can be instantiated in various methods previously proposed for DRE. Here, we introduce the non-negative BR divergence estimators that correspond to LSIF, UKL, BKL, and PULogLoss as follows:

$$\widehat{\mathrm{nnBR}}_{\mathrm{LSIF}}(r) := -\hat{\mathbb{E}}_{\mathrm{nu}}\left[r(X_j) - \frac{C}{2}r^2(X_j)\right] + \left(\frac{1}{2}\hat{\mathbb{E}}_{\mathrm{de}}\left[r^2(X_i)\right] - \frac{C}{2}\hat{\mathbb{E}}_{\mathrm{nu}}\left[r^2(X_j)\right]\right)_+.$$

$$\widehat{\mathrm{nnBR}}_{\mathrm{UKL}}(r) = -\hat{\mathbb{E}}_{\mathrm{nu}}\left[\log\left(r(X_j)\right) - Cr(X_j)\right] + \left(\hat{\mathbb{E}}_{\mathrm{de}}\left[r(X_i)\right] - C\hat{\mathbb{E}}_{\mathrm{nu}}\left[r(X_j)\right]\right)_+,$$

$$\widehat{\mathrm{nnBR}}_{\mathrm{BKL}}(r) = -\hat{\mathbb{E}}_{\mathrm{nu}}\left[\log\left(\frac{r(X_j)}{1 + r(X_j)}\right) + C\log\left(\frac{1}{1 + r(X_j)}\right)\right]$$
$$+ \left(-\hat{\mathbb{E}}_{\mathrm{de}}\left[\log\left(\frac{1}{1 + r(X_i)}\right)\right] + C\hat{\mathbb{E}}_{\mathrm{nu}}\left[\log\left(\frac{1}{1 + r(X_j)}\right)\right]\right)_+,$$

$$\widehat{\mathrm{nnBR}}_{\mathrm{PU}}(r) := -C\hat{\mathbb{E}}_{\mathrm{nu}}\left[\log\left(r(X_j)\right)\right] + \left(C\hat{\mathbb{E}}_{\mathrm{nu}}\left[\log\left(1 - r(X_j)\right)\right] - \hat{\mathbb{E}}_{\mathrm{de}}\left[\log\left(1 - r(X_i)\right)\right]\right)_+.$$

More detailed derivation of $\tilde{f}$ is in Appendix B. In Appendix C, we provide the pseudo code of D3RE. For improving the performance heuristically, we use gradient ascending in our main experiments. Note that the gradient ascent only slightly improves the performance and the use is not essential. In the experiments in Appendix E and G.1.3, we also show the experimental results without the gradient ascent for readers concerning the effect of the gradient ascent.

## 4 THEORETICAL JUSTIFICATION OF D3RE

In this section, we confirm the validity of the proposed method by providing a generalization error bound. Given $n \in \mathbb{N}$ and a distribution $p$, we define the *Rademacher complexity* $\mathcal{R}_n^p$ of a function class $\mathcal{H}$ as $\mathcal{R}_n^p(\mathcal{H}) := \mathbb{E}_p \mathbb{E}_\sigma \left[ \sup_{r \in \mathcal{H}} \left| \frac{1}{n} \sum_{i=1}^n \sigma_i r(X_i) \right| \right]$, where $\{\sigma_i\}_{i=1}^n$ are independent uniform sign variables and $\{X_i\}_{i=1}^n \overset{\text{i.i.d.}}{\sim} p$. We omit $r^*$ from the notation of $\mathrm{BR}_f$ when there is no ambiguity.

### 4.1 GENERALIZATION ERROR BOUND ON BR DIVERGENCE

Theorem 4 in Appendix I provides a generalization error bound under the following assumption.

**Assumption 3.** Let $I_r := (b_r, B_r)$. Assume that there exists an empirical risk minimizer $\hat{r} \in \arg\min_{r \in \mathcal{H}} \widehat{\mathrm{nnBR}}_f(r)$ and a population risk minimizer $\bar{r} \in \arg\min_{r \in \mathcal{H}} \mathrm{BR}_f(r)$. Assume $B_\ell := \sup_{t \in I_r} \{\max\{|\ell_1(t)|, |\ell_2(t)|\}\} < \infty$. Also assume $\ell_1$ (resp. $\ell_2$) is $L_{\ell_1}$-Lipschitz (resp. $L_{\ell_2}$-Lipschitz) on $I_r$. Assume also that $\inf_{r \in \mathcal{H}} (\mathbb{E}_{\mathrm{de}} - C\mathbb{E}_{\mathrm{nu}}) \ell_1(r(X)) > 0$ holds.

In order for the boundedness and Lipschitz continuity in Assumption 3 to hold for the loss functions involving a logarithm (UKL, BKL, PU), a technical assumption $b_r > 0$ is sufficient. We obtain a theoretical guarantee for D3RE from Theorem 4 in Appendix I by additionally imposing Assumption 4 to bound the Rademacher complexities using a previously known result (Golowich et al., 2019, Theorem 1).

**Assumption 4** (Neural networks with bounded complexity)**.** Assume that $p_{\mathrm{nu}}$ and $p_{\mathrm{de}}$ have bounded supports: $\sup_{x \in \mathcal{X}^{\mathrm{de}}} \|x\| < \infty$. Also assume that $\mathcal{H}$ consists of real-valued neural networks of depth $L$ over the domain $\mathcal{X}$, where each parameter matrix $W_j$ has the Frobenius norm at most $B_{W_j} \geq 0$ and with 1-Lipschitz activation functions $\varphi_j$ that are positive-homogeneous (i.e., $\varphi_j$ is applied element-wise and $\varphi_j(\alpha t) = \alpha \varphi_j(t)$ for all $\alpha \geq 0$).

Under Assumption 4, Lemma 3 in Appendix J reveals $\mathcal{R}_{n_{\mathrm{nu}}}^{p_{\mathrm{nu}}}(\mathcal{H}) = \mathcal{O}(1/\sqrt{n_{\mathrm{nu}}})$ and $\mathcal{R}_{n_{\mathrm{de}}}^{p_{\mathrm{de}}}(\mathcal{H}) = \mathcal{O}(1/\sqrt{n_{\mathrm{de}}})$. By combining these with Theorem 4 in Appendix I, we obtain the following theorem:

**Theorem 1** (Generalization error bound for D3RE)**.** *Under Assumptions 3 and 4, for any $\delta \in (0,1)$, we have with probability at least $1 - \delta$,*

$$\mathrm{BR}_f(\hat{r}) - \mathrm{BR}_f(\bar{r}) \leq \frac{\kappa_1}{\sqrt{n_{\mathrm{de}}}} + \frac{\kappa_2}{\sqrt{n_{\mathrm{nu}}}} + 2\Phi_C^f(n_{\mathrm{nu}}, n_{\mathrm{de}}) + B_\ell \sqrt{8 \left( \frac{1}{n_{\mathrm{de}}} + \frac{(1+C)^2}{n_{\mathrm{nu}}} \right) \log \frac{1}{\delta}},$$

*where $\kappa_1, \kappa_2$ are constants that depend on $C, f, B_{p_{\mathrm{de}}}, B_{p_{\mathrm{nu}}}, L,$ and $B_{W_j}$.*

See Remark 6 in Appendix I for the explicit form of this bound. By transforming the BR divergence, Theorem 1 tells us generalization error bounds for various problems. For instance, by defining $f(t)$ as $\log(1 - t) + Ct(\log(t) - \log(1 - t))$ for $0 < t < 1$, the BR divergence $\mathrm{BR}_f(r)$ becomes the same risk functional of PU learning (see Appendix A for the derivation). Then, the generalization bound is similar to the one shown by Kiryo et al. (2017), i.e., the generalization bounds matches the classification error bound of classification only from positive and unlabeled data. Note that the dependency of the bound is standard for classification risk bound with Lipschitz function (See Corollary 15 of Bartlett & Mendelson (2003)). Thus, this result implies that the BR divergence minimization with D3RE also minimize the generalization error bound of binary classification.

### 4.2 ESTIMATION ERROR BOUND ON $L^2$ NORM

Next, we derive the estimation error bound of $\hat{r}$ on the $L^2$ norm. We aim to derive the standard convergence rate of non-parametric regression; that is, under appropriate conditions, the order of $\|\hat{r} - r^*\|_{L^2(p_{\mathrm{de}})}$ is nearly $\mathcal{O}_{\mathbb{P}}(1/(n_{\mathrm{de}} \wedge n_{\mathrm{nu}}))$ (Kanamori et al., 2012). Note that unlike the generalization error bound of the BR divergence, which is related to classification problems, we need to restrict the neural network model to achieve such a convergence rate in general. To the best of our knowledge, it is difficult for showing the convergence rate for any neural network models such as ResNet (Schmidt-Hieber, 2020) in a general way. In the following Theorem 1, for a multi-layer perception with ReLU activation function (Definition 3), we derive a converge rate of $L^2$ distance,

which is the same rate with the nonparametric regression using radial basis function with the Gaussian kernel and LSIF loss (Kanamori et al., 2012). This result also corresponds to a tighter generalization error bound than Theorem 1 under the model restriction. Note that without such a model restriction in Theorem 2, the convergence rate will slower based on the rate of Theorem 1. The proof is shown in Appendix K. To support this result, we empirically investigate the estimator error using an artificially generated dataset with the known true density ratio in Appendix E.

**Theorem 2** ($L^2$ Convergence rate). *Assume $f$ is $\mu$-strongly convex. Let $\mathcal{H}$ be defined as in Definition 3, and assume $r^* = \frac{p_{\text{nu}}}{p_{\text{de}}} \in \mathcal{H}$. Also assume the same conditions as Theorem 3. Then, for any $0 < \gamma < 2$, we have $\|\hat{r} - r^*\|_{L^2(p_{\text{de}})} \leq \mathcal{O}_{\mathbb{P}}\left((\min\{n_{\text{de}}, n_{\text{nu}}\})^{-1/(2+\gamma)}\right)$ $(n_{\text{de}}, n_{\text{nu}} \to \infty)$.*

## 5 EXPERIMENT WITH IMAGE DATA

In this section, we experimentally show how the existing estimators fail to estimate the density ratio when using neural networks and that our proposed estimators succeed. To investigate the performances, we consider PU learning. For a binary classification problem with labels $y \in \{-1, +1\}$, we consider training a classifier only from $p(X \mid y = +1)$ and $p(X)$ to find a positive data point in test data sampled from $p(X)$. The goal is to maximize the *area under the receiver operating characteristic* (AUROC) curve, which is a criterion used for anomaly detection, by estimating the density ratio $r^*(X) = p(X \mid y = +1)/p(X)$. We construct the positive and negative dataset from CIFAR-10[1](Krizhevsky, 2009) dataset with 10 classes. The positive dataset comprises 'airplane', 'automobile', 'ship', and 'truck'; the negative dataset comprises 'bird', 'cat', 'deer', 'dog', 'frog', and 'horse'. We use $1,000$ positive data sampled from $p(X \mid y = +1)$ and $1,000$ unlabeled data sampled from $p(X)$ to train the models. Then, we calculate the AUROCs using $10,000$ test data sampled from $p(X)$. In this case, it is desirable to set $C < \frac{1}{2}$ because $\frac{p(X|y=+1)}{0.5p(X|y=+1)+0.5p(X|y=-1)} = \frac{1}{0.5+0.5\frac{p(X|y=-1)}{p(X|y=+1)}}$. For demonstrative purposes, we use the CNN architecture from the PyTorch tutorial (Paszke et al., 2019). Details of the network structure are shown in Appendix D.2. The model is trained using Adam without weight decay and the parameters $(\beta_1, \beta_2, \epsilon)$ in Kingma & Ba (2015) are fixed at the default of PyTorch, namely $(0.9, 0.999, 10^{-8})$.

First, we compare two of the proposed estimators, $\widehat{\text{nnBR}}_{\text{PU}}$ (nnBR-PU) and $\widehat{\text{nnBR}}_{\text{LSIF}}$ (nnBR-LSIF), with the existing estimators, $\widehat{\text{BR}}_{\text{PU}}$ (PU-NN) and $\widehat{\text{BR}}_{\text{LSIF}}$ (uLSIF-NN). We use the logistic loss for PULogLoss. Additionally, we conduct an experiment with uLSIF-NN using a naively capped model $\tilde{r}(X) = \min\{r(X), 1/C\}$ (Bounded uLSIF). We fix the hyperparameter $C$ at $1/3$. We report the results for two learning rates, $1 \times 10^{-4}$ and $1 \times 10^{-5}$. We conducted 10 trials, and calculated the average AUROCs. We also compute $\hat{\mathbb{E}}_{\text{de}}[\hat{r}(X)]$, which should be close to 1 if we successfully estimate the density ratio, because $\int (p(x|y = +1)/p(X))p(X)dx = 1$. The results are in Figure 2. In all cases, the proposed estimators outperform the other methods. In contract, the unstable behaviors of PU-NN and LISF-NN are caused by the train-loss hacking (also see (Kiryo et al., 2017) Appendix G.1). The experiment also demonstrates that naive capping (Bounded uLSIF) fails to prevent train-loss hacking and leads to suboptimal behavior. A naive capping is insufficient because an unreasonable model such that $r(X_i^{\text{de}}) = 0$ and $r(X_i^{\text{nu}}) = 1/C$ still can be a minimizer by making one part of the empirical BR divergence largely negative, e.g., as $\frac{1}{2}\hat{\mathbb{E}}_{\text{de}}\left[r^2(X)\right] - \frac{C}{2}\hat{\mathbb{E}}_{\text{nu}}\left[r^2(X)\right] = 0 - \frac{1}{2C}$.

Additional experimental results for sensitivity analysis on the upper bound of the density ratio and comparison with various estimators using nnBR divergence are shown in Appendix G.1.

## 6 INLIER-BASED OUTLIER DETECTION

As an applications of D3RE, we introduce *inlier-based outlier detection* with experiments using benchmark datasets. Moreover, we introduce other applications such as *covariate shift adaptation* in Appendix H. In addition to CIFAR-10, we use MNIST[2] (LeCun et al., 1998) and fashion-MNIST (FMNIST)[3] (Xiao et al., 2017). Hido et al. (2008; 2011) applied the direct DRE

---

[1]See https://www.cs.toronto.edu/~kriz/cifar.html.

[2]MNIST has 10 classes from 0 to 9. See http://yann.lecun.com/exdb/mnist/.

[3]FMNIST has 10 classes. See https://github.com/zalandoresearch/fashion-mnist.

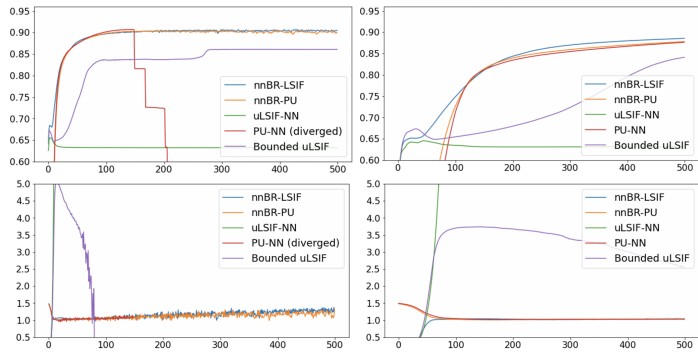

Figure 2: Experimental results of Section 5. The horizontal axis is epoch, and the vertical axis is AUROC. The learning rates of the left and right graphs are $1 \times 10^{-4}$ and $1 \times 10^{-5}$, respectively. The upper graphs show the AUROCs and the lower graphs show $\hat{\mathbb{E}}_{\mathrm{de}}[\hat{r}(X)]$, which will approach 1 when we successfully estimate the density ratio.

Table 2: Average AUROC curve (Mean) with the standard deviation (SD) over 5 trials of anomaly detection methods. For all datasets, each model was trained on a single class and tested against all other classes. The best figure is in bold.

| CIFAR-10 | uLSIF-NN | | nnBR-LSIF | | nnBR-PU | | nnBR-LSIF | | nnBR-PU | | Deep SAD | | GT | |
| Network | LeNet | | LeNet | | LeNet | | WRN | | WRN | | LeNet | | WRN | |
| Inlier Class | Mean | SD | Mean | SD | Mean | SD | Mean | SD | Mean | SD | Mean | SD | Mean | SD |
|---|---|---|---|---|---|---|---|---|---|---|---|---|---|---|
| plane | 0.745 | 0.056 | 0.934 | 0.002 | **0.943** | 0.001 | 0.925 | 0.004 | 0.923 | 0.001 | 0.627 | 0.066 | 0.697 | 0.009 |
| car | 0.758 | 0.078 | 0.957 | 0.002 | **0.968** | 0.001 | 0.965 | 0.002 | 0.960 | 0.001 | 0.606 | 0.018 | 0.962 | 0.003 |
| bird | 0.768 | 0.012 | 0.850 | 0.007 | **0.878** | 0.004 | 0.844 | 0.004 | 0.858 | 0.004 | 0.404 | 0.006 | 0.752 | 0.002 |
| cat | 0.745 | 0.037 | 0.820 | 0.003 | **0.856** | 0.002 | 0.810 | 0.009 | 0.841 | 0.002 | 0.517 | 0.018 | 0.727 | 0.014 |
| deer | 0.758 | 0.036 | 0.886 | 0.004 | **0.909** | 0.002 | 0.864 | 0.008 | 0.872 | 0.002 | 0.704 | 0.052 | 0.863 | 0.014 |

| FMNIST | uLSIF-NN | | nnBR-LSIF | | nnBR-PU | | nnBR-LSIF | | nnBR-PU | | Deep SAD | | GT | |
| Network | LeNet | | LeNet | | LeNet | | WRN | | WRN | | LeNet | | WRN | |
| Inlier Class | Mean | SD | Mean | SD | Mean | SD | Mean | SD | Mean | SD | Mean | SD | Mean | SD |
|---|---|---|---|---|---|---|---|---|---|---|---|---|---|---|
| T-shirt/top | 0.960 | 0.005 | 0.981 | 0.001 | **0.985** | 0.000 | 0.984 | 0.001 | 0.982 | 0.000 | 0.558 | 0.031 | 0.890 | 0.007 |
| Trouser | 0.961 | 0.010 | 0.998 | 0.000 | **1.000** | 0.000 | 0.998 | 0.000 | 0.998 | 0.000 | 0.758 | 0.022 | 0.974 | 0.004 |
| Pullover | 0.944 | 0.012 | 0.976 | 0.001 | 0.980 | 0.001 | **0.983** | 0.002 | 0.972 | 0.001 | 0.617 | 0.046 | 0.902 | 0.005 |
| Dress | 0.973 | 0.006 | 0.986 | 0.001 | **0.992** | 0.000 | 0.991 | 0.001 | 0.986 | 0.000 | 0.525 | 0.038 | 0.843 | 0.014 |
| Coat | 0.958 | 0.006 | 0.978 | 0.001 | **0.983** | 0.000 | 0.981 | 0.002 | 0.974 | 0.000 | 0.627 | 0.029 | 0.885 | 0.003 |

for inlier-based outlier detection: finding outliers in a test set based on a training set consisting only of inliers by using the ratio of training and test data densities as an outlier score. Nam & Sugiyama (2015) and Abe & Sugiyama (2019) proposed using neural networks with DRE for this problem. In relation to the experimental setting of Section 5, the problem setting can be seen as a transductive variant of PU learning (Kato et al., 2019).

We follow the setting proposed by Golan & El-Yaniv (2018) using MNIST, CIFAR-10, and FMNIST. There are ten classes in each dataset; we use one class as the inlier class and all other classes as the outliers. For example, in the case of CIFAR-10, there are $5,000$ train data per class. On the other hand, there are $1,000$ test data for each class, which consists of $1,000$ inlier samples and $9,000$ outlier samples. The AUROC is used as a metric to evaluate whether an outlier class can be detected in the test data. We compare the proposed methods with benchmark methods of deep semi-supervised anomaly detection (DeepSAD) (Ruff et al., 2020) and geometric transformations (GT) (Golan & El-Yaniv, 2018). The details of each method are shown in Appendix F. To compare the methods fairly, we use the same architectures of neural networks from Golan & El-Yaniv (2018) and Ruff et al. (2020), LeNet and Wide Resnet, for D3RE. The details of the structures are shown in Appendix D. A part of the experimental results with CIFAR-10 and FMNIST are shown in Table 2 due to the limitation of the space. The full results are shown in Table 4 in Appendix G.2. The proposed methods result in better AUROCs than the existing methods. The largest performance gain is seen in the CIFAR-10: the mean AUROC is improved by $0.157$ on average between the uLSIF-NN and nnBR-LSIF. Although GT and DeepSAD are designed for a different problem, to the best of our knowledge, these are the state-of-the-art algorithms, which are not based on DRE.

# 7 CONCLUSION

We proposed a non-negative correction to the empirical BR divergence for DRE. Using the prior knowledge of the upper bound of the density ratio, we can prevent train-loss hacking when using flexible models. In our theoretical analysis, we showed the generalization bound of the algorithm.

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

## A  DETAILS OF EXISTING METHODS FOR DRE

In this section, we overview examples of DRE methods in the framework of the density ratio matching under BR divergence.

**Least Squares Importance Fitting (LSIF):**   LSIF minimizes the squared error between a density ratio model $r$ and the true density ratio $r^*$ defined as follows (Kanamori et al., 2009):

$$R_{\mathrm{LSIF}}(r) = \mathbb{E}_{\mathrm{de}}[(r(X) - r^*(X))^2] = \mathbb{E}_{\mathrm{de}}[(r^*(X))^2] - 2\mathbb{E}_{\mathrm{nu}}[r(X)] + \mathbb{E}_{\mathrm{de}}[(r(X))^2].$$

In the *unconstrained LSIF* (uLSIF) (Kanamori et al., 2009), we ignore the first term in the above equation and estimate the density ratio by the following minimization problem:

$$\hat{r} = \arg\min_{r \in \mathcal{H}} \left[ \frac{1}{2}\hat{\mathbb{E}}_{\mathrm{de}}[(r(X))^2] - \hat{\mathbb{E}}_{\mathrm{nu}}[r(X)] + \mathcal{R}(r) \right], \qquad (6)$$

where $\mathcal{R}$ is a regularization term. This empirical risk minimization is equal to minimizing the empirical BR divergence defined in (5) with $f(t) = (t-1)^2/2$.

**Unnormalized Kullback–Leibler (UKL) Divergence and KL Importance Estimation Procedure (KLIEP):**   The KL importance estimation procedure (KLIEP) is derived from the unnormalized Kullback–Leibler (UKL) divergence objective (Sugiyama et al., 2008; Nguyen et al., 2010; Tsuboi et al., 2009; Yamada & Sugiyama, 2009; Yamada et al., 2010), which uses $f(t) = t\log(t) - t$. Ignoring the terms which are irrelevant for the optimization, we obtain the unnormalized Kullback–Leibler (UKL) divergence objective (Nguyen et al., 2010; Sugiyama et al., 2012) as

$$\mathrm{BR}_{\mathrm{UKL}}(r) = \mathbb{E}_{\mathrm{de}}\left[r(X)\right] - \mathbb{E}_{\mathrm{nu}}\left[\log\left(r(X)\right)\right].$$

Directly minimizing UKL is proposed by Nguyen et al. (2010). The KLIEP also solves the same problem with further imposing a constraint that the ratio model $r(X)$ is non-negative for all $X$ and is normalized as

$$\hat{\mathbb{E}}_{\mathrm{de}}\left[r(X)\right] = 1.$$

Then, following is the optimization criterion of KLIEP (Sugiyama et al., 2008):

$$\max_r \hat{\mathbb{E}}_{\mathrm{nu}}\left[\log\left(r(X)\right)\right]$$
$$\text{s.t. } \hat{\mathbb{E}}_{\mathrm{de}}\left[r(X)\right] = 1 \text{ and } r(X) \geq 0 \text{ for all } X.$$

**Logistic Regression:**   By using $f(t) = \log(t) - (1+t)\log(1+t)$, we obtain the following BR divergence called the binary Kullback–Leibler (BKL) divergence:

$$\mathrm{BR}_{\mathrm{BKL}}(r) = -\mathbb{E}_{\mathrm{de}}\left[\log\left(\frac{1}{1+r(X)}\right)\right] - \mathbb{E}_{\mathrm{nu}}\left[\log\left(\frac{r(X)}{1+r(X)}\right)\right].$$

This BR divergence is derived from a formulation based on the logistic regression (Hastie et al., 2001; Sugiyama et al., 2011b).

**PU Learning with the Log Loss:**   Consider a binary classification problem and let $X$ and $y \in \{\pm1\}$ be the feature and the label of a sample, respectively. In PU learning, the goal is to train a classifier only using positive data sampled from $p(X \mid y = +1)$, and unlabeled data sampled from $p(X)$ in binary classification (Elkan & Noto, 2008). More precisely, this problem setting of PU learning is called the *case-control scenario* (Elkan & Noto, 2008; Niu et al., 2016). Let $\mathcal{G}$ be the set of measurable functions from $\mathcal{X}$ to $[\epsilon, 1-\epsilon]$, where $\epsilon \in (0, 1/2)$ is a small positive value. For a loss function $\ell : \mathbb{R} \times \{\pm1\} \to \mathbb{R}^+$, du Plessis et al. (2015) showed that the classification risk of $g \in \mathcal{G}$ in the PU problem setting can be expressed as

$$R_{\mathrm{PU}}(g) = \pi \int \Big(\ell(g(X), +1) - \ell(g(X), -1)\Big) p(X \mid y = +1) dX + \int \ell(g(X), -1)] p(X) dX. \tag{7}$$

According to Kato et al. (2019), we can derive the following risk for DRE from the risk for PU learning (7) as follows:

$$\mathrm{BR}_{\mathrm{PU}}(g) = \frac{1}{R}\mathbb{E}_{\mathrm{nu}}\left[-\log\left(g(X)\right) + \log\left(1 - g(X)\right)\right] - \mathbb{E}_{\mathrm{de}}\left[\log\left(1 - g(X)\right)\right],$$

and Kato et al. (2019) showed that $g^* = \arg\min_{g \in \mathcal{G}} \mathrm{BR}_{\mathrm{PU}}(g)$ satisfies the following:

**Proposition 1.** It holds almost everywhere that

$$g^*(X) = \begin{cases} 1 - \varepsilon & (X \notin D_2), \\ C\frac{p_{\mathrm{nu}}(X)}{p_{\mathrm{de}}(X)} & (X \in D_1 \cap D_2), \\ \varepsilon & (X \notin D_1), \end{cases}$$

where $C = \frac{1}{R}$, $D_1 = \{X \mid Cp_{\mathrm{nu}}(X) \geq \epsilon p_{\mathrm{de}}(X)\}$, and $D_2 = \{X \mid Cp_{\mathrm{nu}}(X) \leq (1 - \epsilon)p_{\mathrm{de}}(X)\}$.

Using this result, we define the empirical version of $\mathrm{BR}_{\mathrm{PU}}(g)$ as follows:

$$\widehat{\mathrm{BR}}_{\mathrm{PU}}(r^*\|r) := C\hat{\mathbb{E}}_{\mathrm{nu}}\left[-\log\left(r(X_i)\right) + \log\left(1 - r(X_j)\right)\right] - \hat{\mathbb{E}}_{\mathrm{de}}\left[\log\left(1 - r(X_i)\right)\right].$$

Note that for $f(t)$ in the BR divergence, we use

$$f(t) = C\log\left(1 - t\right) + Ct\left(\log\left(t\right) - \log\left(1 - t\right)\right).$$

Then, we have

$$\partial f(t) = -\frac{C}{1 - t} + C(\log(t) - \log(1 - t)) + Ct\left(\frac{1}{t} + \frac{1}{1 - t}\right).$$

Therefore, we have

$$\mathrm{BR}_f(r) := \mathbb{E}_{\mathrm{de}}\left[\partial f\left(r(X_i)\right)r(X_i) - f\left(r(X_i)\right)\right] - \mathbb{E}_{\mathrm{nu}}\left[\partial f\left(r(X_j)\right)\right]$$

$$= \mathbb{E}_{\mathrm{de}}\left[-\frac{Cr(X_i)}{1 - r(X_i)} + Cr(X_i)(\log(r(X_i)) - \log(1 - r(X_i))) + Cr^2(X_i)\left(\frac{1}{r(X_i)} + \frac{1}{1 - r(X_i)}\right)\right]$$

$$\quad - \mathbb{E}_{\mathrm{de}}\left[\log\left(1 - r(X_i)\right) + Cr(X_i)\left(\log\left(r(X_i)\right) - \log\left(1 - r(X_i)\right)\right)\right]$$

$$\quad - \mathbb{E}_{\mathrm{nu}}\left[-\frac{C}{1 - r(X_i)} + C(\log(r(X_i)) - \log(1 - r(X_i))) + Cr(X_i)\left(\frac{1}{r(X_i)} + \frac{1}{1 - r(X_i)}\right)\right]$$

$$= \mathbb{E}_{\mathrm{de}}\left[-\frac{Cr(X_i)}{1 - r(X_i)} + Cr(X_i)(\log(r(X_i)) - \log(1 - r(X_i))) + \frac{Cr(X_i)}{1 - r(X_i)}\right]$$

$$\quad - \mathbb{E}_{\mathrm{de}}\left[\log\left(1 - r(X_i)\right) + Cr(X_i)\left(\log\left(r(X_i)\right) - \log\left(1 - r(X_i)\right)\right)\right]$$

$$\quad - \mathbb{E}_{\mathrm{nu}}\left[-\frac{C}{1 - r(X_i)} + C(\log(r(X_i)) - \log(1 - r(X_i))) + \frac{C}{1 - r(X_i)}\right]$$

$$= \mathbb{E}_{\mathrm{de}}\left[\log\left(1 - r(X_i)\right)\right] - C\mathbb{E}_{\mathrm{nu}}\left[\log(r(X_i)) - \log(1 - r(X_i))\right].$$

**Remark 2** (DRE and PU learning). Menon & Ong (2016) showed that minimizing a proper CPE loss is equivalent to minimizing a BR divergence to the true density ratio, and demonstrated the viability of using existing losses from one problem for the other for CPE and DRE. Kato et al. (2019) pointed out the relation between the PU learning and density ratio estimation and leveraged it to solve a sample selection bias problem in PU learning. In this paper, we introduced the BR divergence with $f(t) = \log\left(1 - Ct\right) + Ct\left(\log\left(Ct\right) - \log\left(1 - Ct\right)\right)$, inspired by the objective function of PU learning with the log loss. In the terminology of Menon & Ong (2016), this $f$ results in a DRE objective without a *link function*. In other words, it yields a direct DRE method.

## B   EXAMPLES OF $\tilde{f}$

Here, we show the examples of $\tilde{f}$ such that $\partial f(t) = C\left(\partial f(t)t - f(t)\right) + \tilde{f}(t)$, where $\tilde{f}(t)$ is bounded from above, and $\partial f(t)t - f(t) + A$ is non-negative.

First, we consider $f(t) = (t-1)^2/2$, which results in the LSIF objective. Because $\partial f(t) = t - 1$, we have

$$t - 1 = C\big((t-1)t - (t-1)^2/2\big) + \tilde{f}(t)$$

$$\Leftrightarrow \tilde{f}(t) = -C\big((t-1)t - (t-1)^2/2\big) + t - 1 = -\frac{C}{2}t^2 + \frac{C}{2} + t - 1.$$

The function is a concave quadratic function, therefore it is upper bounded. Note that uLSIF satisfies the Assumption 2 with $A = \frac{1}{2}$ since $\partial f(t)t + f(t) + \frac{1}{2} = \frac{1}{2}t^2$, and Assumption 3 automatically holds. Also note that the term $A$ is irrelevant to the optimization.

Second, we consider $f(t) = t\log(t) - t$, which results in the UKL or KLIEP objective. Because $\partial f(t) = \log(t)$, we have

$$\log(t) = C\big(\log(t)t - t\log(t) + t\big) + \tilde{f}(t)$$

$$\Leftrightarrow \tilde{f}(t) = -tC + \log(t).$$

We can easily confirm that the function is upper bounded by taking the derivative and finding that $t = 1/C$ gives the maximum. Note that UKL satisfies the Assumption 2 with $A = 0$ since $\partial f(t)t - f(t) = t\log(t) - t\log(t) + t = t$, and Assumption 3 automatically holds.

Third, we consider $f(t) = t\log(t) - (1+t)\log(1+t)$, which is used for DRE based on LR or BKL. Because $\partial f(t) = \log(t) - \log(1+t)$, we have

$$\log(t) - \log(1+t) = C\big((\log(t) - \log(1+t))t - t\log(t) + (1+t)\log(1+t)\big) + \tilde{f}(t)$$

$$\Leftrightarrow \tilde{f}(t) = -C\big(\log(1+t)\big) + \log(t) - \log(1+t) = \log\left(\frac{C}{1+t}\right) + \log\left(\frac{t}{1+t}\right).$$

We can easily confirm that the function is upper bounded as the terms involving $t$ always add up to be negative. Note that BKL satisfies the Assumption 2 with $A = 0$ since $\partial f(t)t - f(t) = (\log(t) - \log(1+t))t - t\log(t) + (1+t)\log(1+t) = \log(1+t)$, and Assumption 3 automatically holds.

Fourth, we consider DRE based on PULog. By setting $f(t) = \log(1-t) + Ct(\log(t) - \log(1-t))$, we can obtain the same risk functional introduced in Kiryo et al. (2017). Note that PULog satisfies the Assumption 2 with $A = 0$ since $\partial f(t)t - f(t) = -\log(1-t)$ with $t < 1$, and Assumption 3 automatically holds.

## C  IMPLEMENTATION

The algorithm for D3RE is described in Algorithm 1. For training with a large amount of data, we adopt the *stochastic optimization* by splitting the dataset into mini-batches. In stochastic optimization, we separate the samples into $N$ mini-batches as $(\{X_i^{\mathrm{nu}}\}_{i=1}^{n_{\mathrm{nu},j}}, \{X_i^{\mathrm{de}}\}_{i=1}^{n_{\mathrm{de},j}})$ $(j = 1, \ldots, N$, where $n_{\mathrm{nu},j}$ and $n_{\mathrm{de},j}$ are the sample sizes in each mini-batch. Then, we consider taking sample average in each mini-batch. Let $\hat{\mathbb{E}}_{\mathrm{nu}}^j$ and $\hat{\mathbb{E}}_{\mathrm{de}}^j$ be sample averages over $\{X_i^{\mathrm{nu}}\}_{i=1}^{n_{\mathrm{nu},j}}$ and $\{X_i^{\mathrm{de}}\}_{i=1}^{n_{\mathrm{de},j}}$. In addition, we use regularization such as L1 and L2 penalties, denoted by $\mathcal{R}(r)$. For improving the performance, we heuristically employ *gradient ascent* from Kiryo et al. (2017) when $\hat{\mathbb{E}}_{\mathrm{de}}\big[\ell_1(r(X))\big] - C\hat{\mathbb{E}}_{\mathrm{nu}}\big[\ell_1(r(X))\big]$ becomes less than 0, i.e., update the model in the direction that increases the term. Let us note that our proposed method is agnostic to the optimization procedure, and other methods such as plain gradient descent can be combined with our method. We consider that further theoretical investigation of the optimization procedure is out of scope.

## D  NETWORK STRUCTURE USED IN SECTIONS 5 AND 6

We explain the structures of neural networks used in the experiments.

### D.1  NETWORK STRUCTURE USED IN SECTIONS 5

In Section 5, we used `CIFAR-10` datasets. The model was a convolutional net (Springenberg et al., 2015): $(32 \times 32 \times 3)$-$C(3 \times 6, 3)$-$C(3 \times 16, 3)$-128-84-1, where the input is a $32 \times 32$ RGB image,

---

**Algorithm 1** D3RE

---

**Input:** Training data $\left\{X_i^{\mathrm{nu}}\right\}_{i=1}^{n_{\mathrm{nu}}}$ and $\left\{X_i^{\mathrm{de}}\right\}_{i=1}^{n_{\mathrm{de}}}$, the algorithm for stochastic optimization such as Adam (Kingma & Ba, 2015), the learning rate $\gamma$, the regularization coefficient $\lambda$.
**Output:** A density ratio estimator $\hat{r}$.
**while** No stopping criterion has been met: **do**
    Create $N$ mini-batches $\left\{\left(\left\{X_i^{\mathrm{nu}}\right\}_{i=1}^{n_{\mathrm{nu},j}}, \left\{X_i^{\mathrm{de}}\right\}_{i=1}^{n_{\mathrm{de},j}}\right)\right\}_{j=1}^{N}$.
    **for** $i = 1$ to $N$ **do**
      **if** $\hat{\mathbb{E}}_{\mathrm{de}}\left[\ell_1(r(X))\right] - C\hat{\mathbb{E}}_{\mathrm{nu}}\left[\ell_1(r(X))\right] \geq 0$: **then**
        Gradient decent: set gradient $\nabla_r\left\{\hat{\mathbb{E}}_{\mathrm{nu}}^j\left[\ell_2(r(X))\right] + \hat{\mathbb{E}}_{\mathrm{de}}^j\left[\ell_1(r(X))\right] - C\hat{\mathbb{E}}_{\mathrm{nu}}^j\left[\ell_1(r(X))\right] + \lambda\mathcal{R}(r)\right\}$.
      **else**
        Gradient ascent: set gradient $\nabla_r\left\{-\hat{\mathbb{E}}_{\mathrm{de}}^j\left[\ell_1(r(X))\right] + C\hat{\mathbb{E}}_{\mathrm{nu}}^j\left[\ell_1(r(X))\right] + \lambda\mathcal{R}(r)\right\}$.
      **end if**
      Update $r$ with the gradient and the learning rate $\gamma$.
    **end for**
**end while**

---

$C(3 \times 6, 3)$ indicates that 3 channels of $3 \times 6$ convolutions followed by ReLU is used. This structure has been adopted from the tutorial of Paszke et al. (2019).

### D.2 NETWORK STRUCTURE USED IN SECTIONS 6

**Inlier-based Outlier Detection:** We used the same LeNet-type CNNs proposed in Ruff et al. (2020). In the CNNs, each convolutional module consists of a convolutional layer followed by leaky ReLU activations with leakiness $\alpha = 0.1$ and $(2 \times 2)$-max-pooling. For MNIST, we employ a CNN with two modules: $(32 \times 32 \times 3)$-$C(3 \times 32, 5)$-$C(32 \times 64, 5)$-$C(64 \times 128, 5)$-1. For CIFAR-10 we employ the following architecture: $(32 \times 32 \times 1)$-$C(1 \times 8, 5)$-$C(8 \times 4, 5)$-1 with a batch normalization (Ioffe & Szegedy, 2015) after each convolutional layer.

The WRN architecture was proposed in Zagoruyko & Komodakis (2016) and it is also used in Golan & El-Yaniv (2018). This structure improved the performance of image recognition by decreasing the depth and increasing the width of the residual networks (He et al., 2015). We omit the detailed description of the structure here.

**Covariate Shift Adaptation:** We used the 5-layer perceptron with ReLU activations. The structure is 10000-1000-1000-1000-1000-1.

## E   EXPERIMENTS FOR REGRESSION ERROR USING SYNTHETIC DATASET

In this experiment, we investigate the regression error of the proposed D3RE. We compare our method with the uLSIF (Kanamori et al., 2009) with reproducing kernel Hilbert space (Kanamori et al., 2012). For uLSIF, we use an open code of https://github.com/hoxo-m/densratio_py. For D3RE, we use nnBR-LSIF and 3-layer perceptron with ReLU activation function, where the number of the nodes of the middle layer is 100. We conducted nnBR-LSIF for all $C \in \{0.8, 1, 2, 3, 4, 5, 10, 15, 20\}$. We also compare these method with naively implemented LSIF with the 3-layer perceptron.

Let the dimension of the domain be $d$ and

$$p_{\mathrm{nu}}(X) = \mathcal{N}(X; \mu^{\mathrm{nu}}, I_d), \tag{8}$$

$$p_{\mathrm{de}}(X) = \mathcal{N}(X; \mu^{\mathrm{de}}, I_d), \tag{9}$$

where $\mathcal{N}(\mu, \Sigma)$ denotes the multivariate normal distribution with mean $\mu$ and $\Sigma$, $\mu^{\mathrm{nu}}$ and $\mu^{\mathrm{de}}$ are $d$-dimensional vector such that $\mu^{\mathrm{nu}} = (1, 0, \ldots, 0)^{\top}$ and $\mu^{\mathrm{de}} = (0, 0, \ldots, 0)^{\top}$, and $I_d$ is a $d$-dimensional identity matrix. We fix the sample sizes at $n_{\mathrm{nu}} = n_{\mathrm{de}} = 1,000$ and estimate the density ratio using uLSIF, LSIF, and D3RE (nnBR-LSIF).

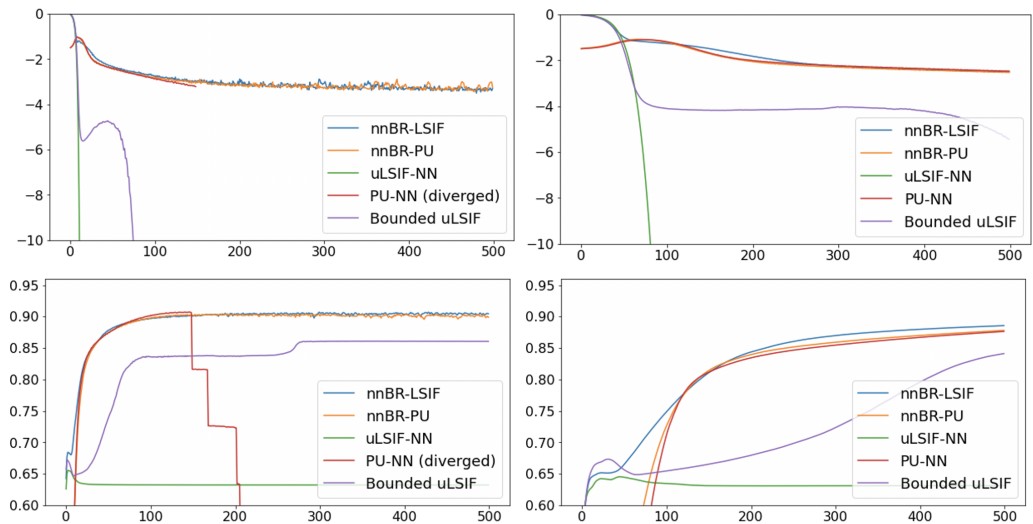

Figure 3: The learning curves of the experiments in Section 5. The horizontal axis is epoch. The vertical axes of the top figures indicate the training losses. The vertical axes of the bottom figures show the AURPC for the test data. The bottom figures are identical to the ones displayed in Section 5.

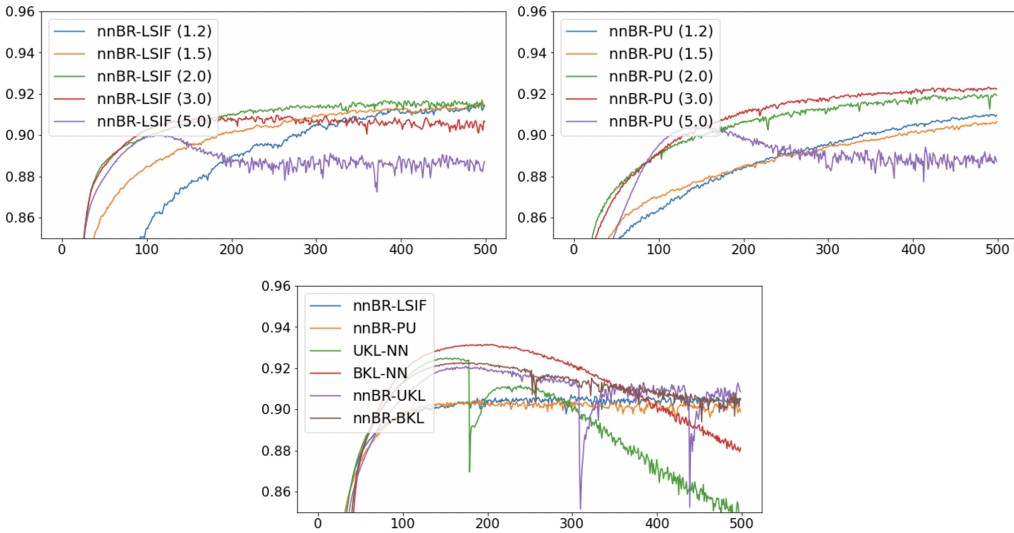

Figure 4: Top figures: the detailed experimental results for Section G.1.1. Bottom figure: the detailed experimental results for Section G.1.2. The horizontal axis is epoch, and the vertical axis is AUROC.

For a performance metric, we use the mean squared error (MSE) and the standard deviation (SD) calculated over 50 trials. Note that in this setting, we can calculate the true density ratio $r^*$. The results are shown in Table 3. The lowest MSE methods are highlighted in bold. As shown in Table 3, the proposed nnBR-LSIF methods estimate the density rate better than the other methods; that is, achieve lower MSEs. Note that in this setting, the upper bound of $r^*$ is infinite because we do not restrict the support of $X$ for simplicity. In the many cases of the results, nnBR-LSIF achieve the best performance around $C = 2$. This result implies that we do not need to know the exact $C$ for better estimation; that is, we can ignore some "out-lier" samples, which cause train-loss hacking.

Table 3: Experimental results (MSEs and SDs) of DRE using synthetic datasets.

| | | uLSIF | LSIF-NN | D3RE (nnBR-LSIF) | | | | | | | | |
|---|---|---|---|---|---|---|---|---|---|---|---|---|
| | | | | $C = 0.8$ | $C = 1$ | $C = 2$ | $C = 3$ | $C = 4$ | $C = 5$ | $C = 10$ | $C = 15$ | $C = 20$ |
| dim = 10 | MSE | 2.378 | 1.272 | 1.750 | 1.695 | 1.191 | 0.964 | 0.873 | **0.833** | 0.948 | 1.079 | 1.170 |
| | SD | 1.143 | 0.413 | 0.570 | 0.563 | 0.523 | 0.487 | 0.459 | 0.424 | 0.370 | 0.331 | 0.387 |
| dim = 20 | MSE | 1.684 | 2.694 | 1.704 | 1.646 | 1.307 | **1.272** | 1.337 | 1.444 | 2.066 | 2.697 | 3.098 |
| | SD | 0.372 | 0.409 | 0.380 | 0.368 | 0.328 | 0.297 | 0.283 | 0.288 | 0.285 | 0.346 | 0.374 |
| dim = 30 | MSE | 1.786 | 3.724 | 1.811 | 1.747 | **1.488** | 1.577 | 1.798 | 2.019 | 3.238 | 4.306 | 5.432 |
| | SD | 0.456 | 0.460 | 0.459 | 0.449 | 0.411 | 0.400 | 0.401 | 0.379 | 0.370 | 0.464 | 0.543 |
| dim = 50 | MSE | 1.791 | 8.717 | 1.817 | 1.753 | **1.609** | 1.818 | 2.194 | 2.614 | 4.848 | 6.955 | 8.798 |
| | SD | 0.562 | 1.518 | 0.571 | 0.555 | 0.513 | 0.503 | 0.484 | 0.465 | 0.488 | 0.597 | 0.672 |
| dim = 100 | MSE | 1.723 | 4.849 | 1.748 | 1.693 | **1.626** | 1.860 | 2.226 | 2.709 | 5.528 | 8.605 | 11.557 |
| | SD | 0.574 | 4.182 | 0.575 | 0.571 | 0.540 | 0.532 | 0.495 | 0.563 | 0.672 | 0.790 | 1.140 |

## F  EXISTING METHODS FOR ANOMALY DETECTION

This section introduces the existing methods for anomaly detection. DeepSAD is a method for semi-supervised anomaly detection, which tries to take advantage of labeled anomalies (Ruff et al., 2020). GT proposed by Golan & El-Yaniv (2018) trains neural networks based on a self-labeled dataset by performing 72 geometric transformations. The anomaly score based on GT is calculated based on the Dirichlet distribution obtained by maximum likelihood estimation using the softmax output from the trained network.

In the problem setting of the DeepSAD, we have access to a small pool of labeled samples, e.g. a subset verified by some domain expert as being normal or anomalous. In the experimental results shown in Ruff et al. (2020) indicate that, when we can use such samples, the DeepSAD outperforms the other methods. However, in our experimental results, such samples are not assumed to be available, hence the method does not perform well. The problem setting of Ruff et al. (2020) and ours are both termed *semi-supervised learning* in anomaly detection, but the two settings are different.

## G  DETAILS OF EXPERIMENTS

The details of experiments are shown in this section. The description of the data is as follows:

MNIST: The MNIST database is one of the most popular benchmark datasets for image classification, which consists of $28 \times 28$ pixel handwritten digits from 0 to 9 with $60,000$ train samples and $10,000$ test samples (LeCun et al., 1998).

CIFAR-10: The CIFAR-10 dataset consists of $60,000$ color images of size $32 \times 32$ from 10 classes, each having 6000. There are $50,000$ training images and $10,000$ test images (Krizhevsky et al., 2012).

fashion-MNIST: The fashion-MNIST dataset consists of $70,000$ grayscale images of size $28 \times 28$ from 10 classes. There are $60,000$ training images and $10,000$ test images (Xiao et al., 2017).

Amazon Review Dataset: Blitzer et al. (2007) published the text data of Amazon review. The data originally consists of a rating (0-5 stars) for four different genres of products in the electronic commerce site Amazon.com: books, DVDs, electronics, and kitchen appliances. Blitzer et al. (2007) also released the pre-processed and balanced data of the original data. The pre-processed data consists of text data with four labels 1, 2, 4, and 5. We map the text data into $10,000$ dimensional data by the TF-IDF mapping with that vocabulary size. In the experiment, for the pre-processed data, we solve the regression problem where the text data are the inputs and the ratings 1, 2, 4, and 5 are the outputs. When evaluating the performance, following Menon & Ong (2016), we calculate PD (=1-AUROUC) by regarding 4 and 5 ratings as positive labels and 1 and 2 ratings as negative labels.

### G.1  EXPERIMENTS WITH IMAGE DATA

We show the additional results of Section 5. Figure 3, we show the training loss of LSIF-based methods to demonstrate the train-loss hacking phenomenon caused by the objective function without

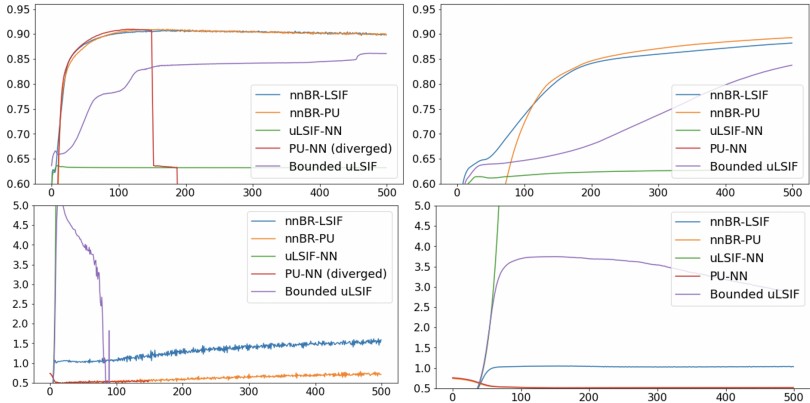

Figure 5: Experimental results of Section 5 without gradient ascent/descent heuristic. The horizontal axis is epoch, and the vertical axis is AUROC. The learning rates of the left and right graphs are $1 \times 10^{-4}$ and $1 \times 10^{-5}$, respectively. The upper graphs show the AUROCs and the lower graphs show $\hat{\mathbb{E}}_{\mathrm{de}}[\hat{r}(X)]$, which will approach 1 when we successfully estimate the density ratio.

a lower bound. In Figure 3, even though the training loss of uLSIF-NN and that of bounded uLSIF decrease more rapidly than that of nnBR-LSIF, the test AUROC score (the higher the better) either drops or fails to increase. These graphs are the manifestations of the severe train-loss hacking in DRE without our proposed device.

### G.1.1 EMPIRICAL SENSITIVITY ANALYSIS ON THE UPPER BOUNDS OF THE DENSITY RATIO

Next, we investigate the sensitivity of D3RE to the hyperparameter $C$. We use nnBR-LSIF and nnBR-PU as in Section 5, but vary the hyperparameter $C$ in $\{1/1.2, 1/1.5, 1/2.0.1/3.0, 1/5.0\}$. The other settings remain unchanged from the previous section. These results are shown in Figure 4. For $\overline{R} = 2.0$, the estimator show a better performance when $1/C$ is close to 2.0.

### G.1.2 COMPARISON WITH VARIOUS ESTIMATORS USING NNBR DIVERGENCE

Let UKL-NN and BKL-NN be DRE method with the UKL and BKL losses with neural networks without non-negative correction. Finally, we examine the performances of nnBR-LSIF, nnBR-PU, UKL-NN, BKL-NN, nnBR-UKL, and nnBR-BKL. The learning rate was $1 \times 10^{-4}$, and the other settings were identical to those in the previous experiments. These results are shown in Figure 4. UKL-NN and BKL-NN also suffer train-loss hacking although BKL loss seems to be more robust against the train-loss hacking than the other loss functions . Although nnBR-UKL and nnBR-BKL show better performance in earlier epochs, nnBR-LSIF and nnBR-PU appear more stable.

### G.1.3 RESULTS WITHOUT GRADIENT ASCENT

We also show the experimental results without the gradient ascent heuristic. Figure 5 corresponds to the Figure 2 without the gradient ascent heuristic. Figure 6 corresponds to the Figure 3 without the gradient ascent heuristic. Figure 7 corresponds to the Figure 4 without the gradient ascent heuristic. As shown these experiments, although the gradient ascent/descent heuristic improve the performance, there is no significant difference between empirical performance with and without the heuristic. Therefore, we recommend practitioners to use the gradient ascent/descent heuristic, but if readers concern the theoretical guarantee, they can use the plain gradient descent algorithm; that is, naively minimize the proposed original empirical nnBR risk.

### G.2 EXPERIMENTS OF INLIER-BASED OUTLIER DETECTION

In Table 4, we show the full results of inlier-based outlier detection. In almost all the cases, D3RE for inlier-based outlier detection outperforms the other methods. As explained in Section F, we consider that DeepSAD does not work well because the method assumes the availability of the labeled anomaly data, which is not available in our problem setting.

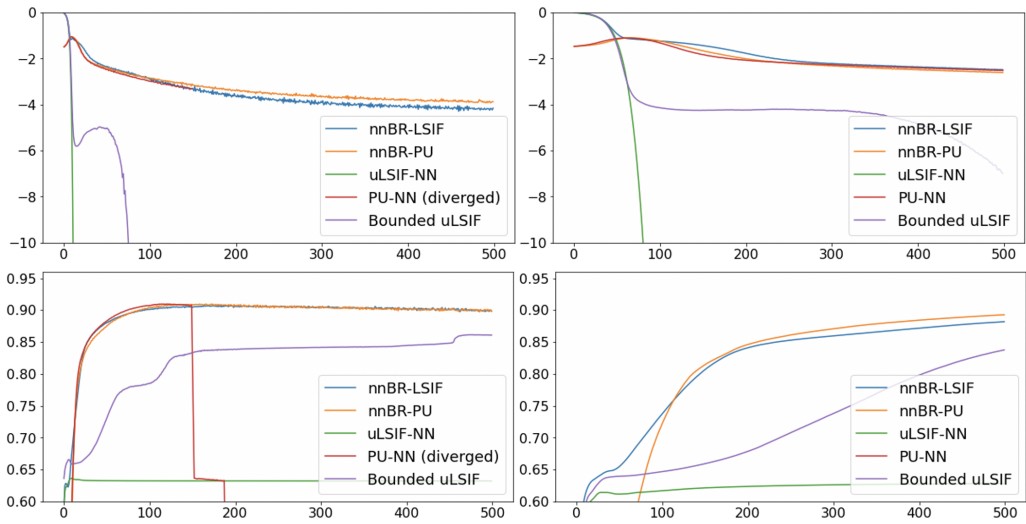

Figure 6: The learning curves of the experiments in Section 5 without gradient ascent/descent heuristic. The horizontal axis is epoch. The vertical axes of the top figures indicate the training losses. The vertical axes of the bottom figures show the AURPC for the test data. The bottom figures are identical to the ones displayed in Section 5.

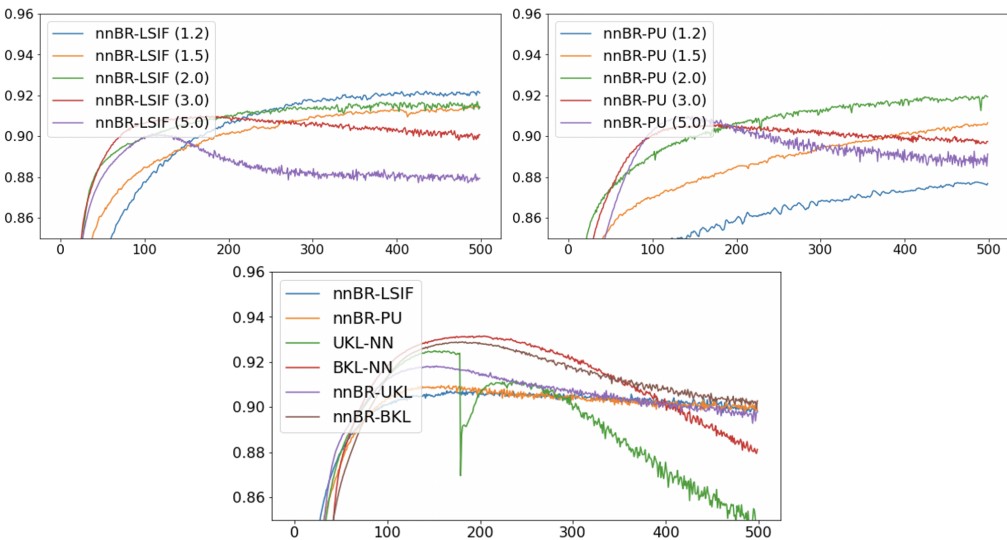

Figure 7: Top figures: the detailed experimental results for Section G.1.1 without gradient ascent/descent heuristic. Bottom figure: the detailed experimental results for Section G.1.2. The horizontal axis is epoch, and the vertical axis is AUROC.

**Remark 3** (Benchmark Methods). Although GT is outperformed by our proposed method, the problem setting for the comparison is not in favor of GT as it does not assume the access to the test data. Recently proposed methods for semi-supervised anomaly detection by Ruff et al. (2020) did not perform well without using other side information used in Ruff et al. (2020). On the other hand, there is no other competitive methods in this problem setting, to the best of our knowledge.

Table 4: Average area under the ROC curve (Mean) of anomaly detection methods averaged over 5 trials with the standard deviation (SD). For all datasets, each model was trained on the single class, and tested against all other classes. The best performing method in each experiment is in bold. *SD*: Standard deviation.

| MNIST Network Inlier Class | uLSIF-NN LeNet Mean | SD | nnBR-LSIF LeNet Mean | SD | nnBR-PU LeNet Mean | SD | nnBR-LSIF WRN Mean | SD | nnBR-PU WRN Mean | SD | Deep SAD LeNet Mean | SD | GT WRN Mean | SD |
|---|---|---|---|---|---|---|---|---|---|---|---|---|---|---|
| 0 | 0.999 | 0.000 | 0.997 | 0.000 | 0.999 | 0.000 | **1.000** | 0.000 | **1.000** | 0.000 | 0.592 | 0.051 | 0.963 | 0.002 |
| 1 | **1.000** | 0.000 | 0.999 | 0.000 | **1.000** | 0.000 | **1.000** | 0.000 | **1.000** | 0.000 | 0.942 | 0.016 | 0.517 | 0.039 |
| 2 | 0.997 | 0.001 | 0.994 | 0.000 | 0.997 | 0.001 | **1.000** | 0.000 | **1.000** | 0.001 | 0.447 | 0.027 | 0.992 | 0.001 |
| 3 | 0.997 | 0.000 | 0.995 | 0.001 | 0.998 | 0.000 | **1.000** | 0.000 | **1.000** | 0.000 | 0.562 | 0.035 | 0.974 | 0.001 |
| 4 | 0.998 | 0.000 | 0.997 | 0.001 | 0.999 | 0.000 | **1.000** | 0.000 | **1.000** | 0.000 | 0.646 | 0.015 | 0.989 | 0.001 |
| 5 | 0.997 | 0.001 | 0.996 | 0.001 | 0.998 | 0.000 | **1.000** | 0.000 | **1.000** | 0.000 | 0.502 | 0.046 | 0.990 | 0.001 |
| 6 | 0.997 | 0.001 | 0.997 | 0.001 | 0.999 | 0.000 | **1.000** | 0.000 | **1.000** | 0.000 | 0.671 | 0.027 | 0.998 | 0.000 |
| 7 | 0.996 | 0.001 | 0.993 | 0.001 | 0.998 | 0.001 | **1.000** | 0.000 | **1.000** | 0.001 | 0.685 | 0.032 | 0.927 | 0.004 |
| 8 | 0.997 | 0.000 | 0.994 | 0.001 | 0.997 | 0.000 | **0.999** | 0.000 | **0.999** | 0.000 | 0.654 | 0.026 | 0.949 | 0.002 |
| 9 | 0.993 | 0.002 | 0.990 | 0.002 | 0.994 | 0.001 | **0.998** | 0.001 | **0.998** | 0.001 | 0.786 | 0.021 | 0.989 | 0.001 |

| CIFAR-10 Network Inlier Class | uLSIF-NN LeNet Mean | SD | nnBR-LSIF LeNet Mean | SD | nnBR-PU LeNet Mean | SD | nnBR-LSIF WRN Mean | SD | nnBR-PU WRN Mean | SD | Deep SAD LeNet Mean | SD | GT WRN Mean | SD |
|---|---|---|---|---|---|---|---|---|---|---|---|---|---|---|
| plane | 0.745 | 0.056 | 0.934 | 0.002 | **0.943** | 0.001 | 0.925 | 0.004 | 0.923 | 0.001 | 0.627 | 0.066 | 0.697 | 0.009 |
| car | 0.758 | 0.078 | 0.957 | 0.002 | **0.968** | 0.001 | 0.965 | 0.002 | 0.960 | 0.001 | 0.606 | 0.018 | 0.962 | 0.003 |
| bird | 0.768 | 0.012 | 0.850 | 0.007 | **0.878** | 0.004 | 0.844 | 0.004 | 0.858 | 0.004 | 0.404 | 0.006 | 0.752 | 0.002 |
| cat | 0.745 | 0.037 | 0.820 | 0.003 | **0.856** | 0.002 | 0.810 | 0.009 | 0.841 | 0.002 | 0.517 | 0.018 | 0.727 | 0.014 |
| deer | 0.758 | 0.036 | 0.886 | 0.004 | **0.909** | 0.002 | 0.864 | 0.008 | 0.872 | 0.002 | 0.704 | 0.052 | 0.863 | 0.014 |
| dog | 0.728 | 0.103 | 0.875 | 0.004 | **0.906** | 0.002 | 0.887 | 0.005 | 0.896 | 0.002 | 0.490 | 0.025 | 0.873 | 0.002 |
| frog | 0.750 | 0.060 | 0.944 | 0.003 | **0.958** | 0.001 | 0.948 | 0.004 | 0.948 | 0.001 | 0.744 | 0.014 | 0.879 | 0.008 |
| horse | 0.782 | 0.048 | 0.928 | 0.003 | 0.948 | 0.002 | 0.921 | 0.007 | 0.927 | 0.002 | 0.519 | 0.015 | **0.953** | 0.001 |
| ship | 0.780 | 0.048 | 0.958 | 0.003 | **0.965** | 0.001 | 0.964 | 0.002 | 0.957 | 0.001 | 0.430 | 0.062 | 0.921 | 0.009 |
| truck | 0.708 | 0.081 | 0.939 | 0.003 | **0.955** | 0.001 | 0.952 | 0.003 | 0.949 | 0.001 | 0.393 | 0.008 | 0.911 | 0.003 |

| FMNIST Network Inlier Class | uLSIF-NN LeNet Mean | SD | nnBR-LSIF LeNet Mean | SD | nnBR-PU LeNet Mean | SD | nnBR-LSIF WRN Mean | SD | nnBR-PU WRN Mean | SD | Deep SAD LeNet Mean | SD | GT WRN Mean | SD |
|---|---|---|---|---|---|---|---|---|---|---|---|---|---|---|
| T-shirt/top | 0.960 | 0.005 | 0.981 | 0.001 | **0.985** | 0.000 | 0.984 | 0.001 | 0.982 | 0.000 | 0.558 | 0.031 | 0.890 | 0.007 |
| Trouser | 0.961 | 0.010 | 0.998 | 0.000 | **1.000** | 0.000 | 0.998 | 0.000 | 0.998 | 0.000 | 0.758 | 0.022 | 0.974 | 0.004 |
| Pullover | 0.944 | 0.012 | 0.976 | 0.001 | 0.980 | 0.001 | **0.983** | 0.002 | 0.972 | 0.001 | 0.617 | 0.046 | 0.902 | 0.005 |
| Dress | 0.973 | 0.006 | 0.986 | 0.001 | **0.992** | 0.000 | 0.991 | 0.001 | 0.986 | 0.000 | 0.525 | 0.038 | 0.843 | 0.014 |
| Coat | 0.958 | 0.006 | 0.978 | 0.001 | **0.983** | 0.000 | 0.981 | 0.002 | 0.974 | 0.000 | 0.627 | 0.029 | 0.885 | 0.003 |
| Sandal | 0.968 | 0.011 | 0.997 | 0.001 | **0.999** | 0.000 | **0.999** | 0.000 | **0.999** | 0.000 | 0.681 | 0.023 | 0.949 | 0.005 |
| Shirt | 0.919 | 0.005 | 0.952 | 0.001 | **0.958** | 0.001 | 0.944 | 0.005 | 0.932 | 0.001 | 0.618 | 0.015 | 0.842 | 0.004 |
| Sneaker | 0.991 | 0.001 | 0.994 | 0.002 | **0.998** | 0.000 | **0.998** | 0.000 | **0.998** | 0.000 | 0.802 | 0.054 | 0.954 | 0.006 |
| Bag | 0.980 | 0.005 | 0.994 | 0.001 | **0.999** | 0.000 | 0.998 | 0.000 | **0.999** | 0.000 | 0.447 | 0.034 | 0.973 | 0.006 |
| Ankle boot | 0.992 | 0.001 | 0.985 | 0.015 | **0.999** | 0.000 | 0.997 | 0.000 | 0.996 | 0.000 | 0.583 | 0.023 | 0.996 | 0.000 |

### G.3 EXPERIMENTS OF COVARIATE SHIFT ADAPTATION

In Table 5, we show the detailed results of experiments of covariate shift adaptation. Even when the training data and the test data follow the same distribution, the covariate shift adaptation based on D3RE improves the mean PD. We consider that this is because the importance weighting emphasizes the loss in the empirical higher-density regions of the test examples.

## H OTHER APPLICATIONS

In this section, we explain other potential applications of the proposed method.

### H.1 COVARIATE SHIFT ADAPTATION BY IMPORTANCE WEIGHTING

We consider training a model using input distribution different from the test input distribution, which is called *covariate shift*, (Bickel et al., 2009). To solve this problem, the density ratio has been used via importance weighting (IW) (Shimodaira, 2000; Yamada et al., 2010; Reddi et al., 2015).

Table 5: Average PD (Mean) with standard deviation (SD) over 10 trials with different seeds per method. The best performing method in terms of the mean PD is specified by bold face.

| Domains (Train → Test) | books → books | | dvd → books | | dvd → dvd | | elec → books | | elec → dvd | |
|---|---|---|---|---|---|---|---|---|---|---|
| DRE method | Mean | SD | Mean | SD | Mean | SD | Mean | SD | Mean | SD |
| w/o IW | 0.093 | 0.003 | 0.128 | 0.008 | 0.100 | 0.005 | 0.212 | 0.012 | 0.187 | 0.008 |
| Kernel uLSIF | 0.089 | 0.002 | 0.114 | 0.006 | 0.094 | 0.004 | 0.200 | 0.009 | 0.179 | 0.006 |
| Kernel KLIEP | 0.089 | 0.002 | 0.116 | 0.006 | 0.094 | 0.004 | 0.205 | 0.011 | 0.184 | 0.008 |
| uLSIF-NN | 0.093 | 0.003 | 0.128 | 0.008 | 0.100 | 0.005 | 0.212 | 0.012 | 0.187 | 0.008 |
| PU-NN | 0.093 | 0.003 | 0.128 | 0.008 | 0.100 | 0.005 | 0.212 | 0.012 | 0.187 | 0.008 |
| nnBR-LSIF | **0.086** | 0.002 | **0.113** | 0.005 | **0.091** | 0.004 | **0.199** | 0.009 | **0.176** | 0.005 |
| nnBR-PU | 0.090 | 0.003 | **0.113** | 0.006 | 0.096 | 0.004 | **0.199** | 0.009 | **0.176** | 0.006 |

| Domains (Train → Test) | elec → elec | | kitchen → books | | kitchen → dvd | | kitchen → elec | | kitchen → kitchen | |
|---|---|---|---|---|---|---|---|---|---|---|
| DRE method | Mean | SD | Mean | SD | Mean | SD | Mean | SD | Mean | SD |
| w/o IW | 0.079 | 0.005 | 0.202 | 0.013 | 0.185 | 0.006 | 0.073 | 0.004 | 0.062 | 0.002 |
| Kernel uLSIF | 0.072 | 0.003 | 0.192 | 0.007 | 0.178 | 0.008 | 0.071 | 0.003 | 0.060 | 0.003 |
| Kernel KLIEP | 0.072 | 0.003 | 0.195 | 0.005 | 0.182 | 0.007 | 0.072 | 0.004 | 0.060 | 0.002 |
| uLSIF-NN | 0.079 | 0.005 | 0.202 | 0.013 | 0.185 | 0.006 | 0.073 | 0.004 | 0.062 | 0.002 |
| PU-NN | 0.079 | 0.005 | 0.202 | 0.013 | 0.185 | 0.006 | 0.073 | 0.004 | 0.062 | 0.002 |
| nnBR-LSIF | **0.071** | 0.003 | **0.189** | 0.008 | **0.174** | 0.008 | **0.068** | 0.003 | **0.058** | 0.003 |
| nnBR-PU | 0.074 | 0.004 | 0.190 | 0.008 | **0.174** | 0.008 | **0.068** | 0.003 | 0.062 | 0.005 |

Table 6: Average PD (Mean) with standard deviation (SD) over 10 trials with different seeds per method. The best performing method in terms of the mean PD is specified by bold face.

| Domains (Train → Test) | book → dvd | | book → elec | | book → kitchen | | dvd → elec | | dvd → kitchen | | elec → kitchen | |
|---|---|---|---|---|---|---|---|---|---|---|---|---|
| DRE method | Mean | SD | Mean | SD | Mean | SD | Mean | SD | Mean | SD | Mean | SD |
| w/o IW | 0.126 | 0.008 | 0.174 | 0.010 | 0.166 | 0.009 | 0.162 | 0.006 | 0.146 | 0.010 | 0.074 | 0.005 |
| Kernel uLSIF | 0.122 | 0.009 | 0.162 | 0.009 | 0.159 | 0.007 | 0.153 | 0.006 | 0.142 | 0.007 | 0.068 | 0.005 |
| Kernel KLIEP | 0.130 | 0.010 | 0.164 | 0.009 | 0.161 | 0.007 | 0.154 | 0.006 | 0.143 | 0.006 | 0.070 | 0.005 |
| uLSIF-NN | 0.126 | 0.008 | 0.174 | 0.010 | 0.166 | 0.009 | 0.162 | 0.006 | 0.146 | 0.010 | 0.074 | 0.005 |
| PU-NN | 0.126 | 0.008 | 0.174 | 0.010 | 0.166 | 0.009 | 0.162 | 0.006 | 0.146 | 0.010 | 0.074 | 0.005 |
| nnBR-LSIF | 0.120 | 0.008 | **0.160** | 0.008 | 0.157 | 0.008 | **0.148** | 0.006 | **0.138** | 0.007 | **0.066** | 0.005 |
| nnBR-PU | **0.119** | 0.008 | **0.160** | 0.008 | **0.156** | 0.007 | **0.148** | 0.005 | **0.138** | 0.007 | **0.066** | 0.005 |

We use a document dataset of Amazon[4] (Blitzer et al., 2007) for multi-domain sentiment analysis (Blitzer et al., 2007). This data consists of text reviews from four different product domains: book, electronics (elec), dvd, and kitchen. Following Chen et al. (2012) and Menon & Ong (2016), we transform the text data using TF-IDF to map them into the instance space $\mathcal{X} = \mathbb{R}^{10000}$ (Salton & McGill, 1986). Each review is endowed with four labels indicating the positivity of the review, and our goal is to conduct regression for these labels. To achieve this goal, we perform kernel ridge regression with the polynomial kernel. We compare regression without IW (w/o IW) with regression using the density ratio estimated by PU-NN, uLSIF-NN, nnBR-LSIF, nnBR-PU, uLSIF with Gaussian kernels (Kernel uLSIF), and KLIEP with Gaussian kernels (Kernel KLIEP). We conduct experiments on $2,000$ samples from one domain, and test $2,000$ samples. Following Menon & Ong (2016), we reduce the dimension into $100$ dimensions by principal component analysis when using Kernel uLSIF, Kernel KLEIP, and regressions. Following Menon & Ong (2016) and Cortes & Mohri (2011), the mean and standard deviation of the pairwise disagreement (PD), $1 - \mathrm{AUROC}$, is reported. A part of results is in Table 6. The full results are in Appendix G.3. The methods with D3RE show preferable performance, but the improvement is not significant compared with the image data. We consider this is owing to the difficulty of the covariate shift problem in this dataset.

**$f$-divergence Estimation:** $f$-divergences (Ali & Silvey, 1966; Csiszár, 1967) are the discrepancy measures of probability densities based on the density ratio, hence the proposed method can be used for their estimation. They include the KL divergence (Kullback & Leibler, 1951), the Hellinger distance (Hellinger, 1909), and the Pearson divergence (Pearson, 1900), as examples.

**Two-sample Homogeneity Test:** The purpose of a homogeneity test is to determine if two or more datasets come from the same distribution (Loevinger, 1948). For two-sample testing, using a semiparametric $f$-divergence estimator with nonparametric density ratio models has been studied

---
[4] http://john.blitzer.com/software.html

(Keziou., 2003; Keziou & Leoni-Aubin, 2005). Kanamori et al. (2010) and Sugiyama et al. (2011a) employed direct DRE for the nonparametric DRE.

**Generative Adversarial Networks:** Generative adversarial networks (GANs) are successful deep generative models, which learns to generate new data with the same distribution as the training data Goodfellow et al. (2014). Various GAN methods have been proposed, amongst which Nowozin et al. (2016) proposed *f-GAN*, which minimizes the variational estimate of $f$-divergence. Uehara et al. (2016) extended the idea of Nowozin et al. (2016) to use BR divergence minimization for DRE. The estimator proposed in this paper also has a potential to improve the method of Uehara et al. (2016).

**Average Treatment Effect Estimation and Off-policy Evaluation:** One of the goals in causal inference is to estimate the expected treatment effect, which is a *counterfactual* value. Therefore, following the causality formulated by Rubin (1974), we consider estimating the average treatment effect (ATE). Recently, from machine learning community, off-policy evaluation (OPE) is also proposed, which is a generalization of ATE (Dudík et al., 2011; Imai & Ratkovic, 2014; Wang et al., 2017; Narita et al., 2019; Bibaut et al., 2019; Kallus & Uehara, 2019; Oberst & Sontag, 2019). OPE has garnered attention in applications such as advertisement design selection, personalized medicine, search engines, and recommendation systems (Beygelzimer & Langford, 2009; Li et al., 2010; Athey & Wager, 2017).

The problem in ATE estimation and OPE is sample selection bias. For removing the bias, the density ratio has a critical role. An idea of using the density ratio dates back to (Rosenbaum, 1987), which proposed an inverse probability weighting (IPW) method (Horvitz & Thompson, 1952) for ATE estimation. In the IPW method, we approximate the parameter of interest with the sample average with inverse assignment probability of treatment (action), which is also called propensity score. Here, it is known that using the true assignment probability yields higher variance than the case where we use an estimated assignment probability even if we know the true value (Hirano et al., 2003; Henmi & Eguchi, 2004; Henmi et al., 2007). This property can be explained from the viewpoint of semiparametric efficiency (Bickel et al., 1998). While the asymptotic variance of the IPW estimator with an estimated propensity score can achieve the efficiency bound, that of the IPW estimator with the true propensity score does not.

By extending the IPW estimator, more robust ATE estimators are proposed by Rosenbaum (1983), which is known as a doubly robust (DR) estimator. The doubly robust estimator is not only robust to model misspecification but also useful in showing asymptotic normality. In particular, when using the density ratio and the other nuisance parameters estimated from the machine learning method, the conventional IPW and DR estimators do not have asymptotic normality (Chernozhukov et al., 2018). This is because the nuisance estimators do not satisfy Donsker's condition, which is required for showing the asymptotic normality of semiparametric models. However, by using the sample splitting method proposed by Klaassen (1987), Zheng & van der Laan (2011), and Chernozhukov et al. (2018), we can show the asymptotic normality when using the DR estimator. Note that for the IPW estimator, we cannot show the asymptotic normality even if using sample-splitting.

When using the IPW and DR estimator, we often consider a two-stage approach: in the first stage, we estimate the nuisance parameters, including the density ratio; in the second stage, we construct a semiparametric ATE estimator including the first-stage nuisance estimators. This is also called two-step generalized method of moments (GMM). On the other hand, from the causal inference community, there are also weighting-based covariate balancing methods (Qin & Zhang, 2007; Tan, 2010; Hainmueller, 2012; Imai & Ratkovic, 2014). In particular, Imai & Ratkovic (2014) proposed a covariate balancing propensity score (CBPS), which simultaneously estimates the density ratio and ATE. The idea of CBPS is to construct moment conditions, including the density ratios, and estimate the ATE and density ratio via GMM simultaneously. Although the asymptotic property of the CBPS is the same as other conventional estimators, existing empirical studies report that the CBPS outperforms them (Wyss et al., 2014).

Readers may feel that the CBPS has a close relationship with the direct DRE. However, Imai & Ratkovic (2014) is less relevant to the context of the direct DRE. From the DRE perspective, the method of Imai & Ratkovic (2014) boils down to the method of Gretton et al. (2009), which proposed direct DRE through moment matching. The research motivation of Imai & Ratkovic (2014)

is to estimate the ATE with estimating a nuisance density ratio estimator simultaneously. Therefore, the density ratio itself is *nuisance* parameter; that is, they are not interested in the estimation performance of the density ratio. Under their motivation, they are interested in a density ratio estimator satisfying the moment condition for estimating the ATE, not in a density ratio estimator predicting the true density ratio well. In addition, while the direct DRE method adopts linear-in-parameter models and neural networks (our work), it is not appropriate to use those methods with the CBPS (Chernozhukov et al., 2018). This is because the density ratio estimator does not satisfy Donsker's condition. Even naive Ridge and Lasso regression estimators do not satisfy the Donsker's condition. Therefore, when using machine learning methods for estimating the density ratio, we cannot show asymptotic normality of an ATE estimator obtained by the CBPS; therefore, we need to use the sample-splitting method by (Chernozhukov et al., 2018). This means that when using the CBPS, we can only use a naive parametric linear model without regularization or classic nonparametric kernel regression. Recently, for GMM with such non-Donsker nuisance estimators, Chernozhukov et al. (2016) also proposed a new GMM method based on the conventional two-step approach. For these reasons, the CBPS is less relevant to the direct DRE context.

**Off-policy Evaluation with External Validity:**    By the problem setting of combining causal inference and domain adaptation, Kato et al. (2020) recently proposed using covariate shift adaptation to solve the *external validity* problem in OPE, i.e., the case that the distribution of covariates is the same between the historical and evaluation data (Cole & Stuart, 2010; Pearl & Bareinboim, 2014).

**Change Point Detection:**    The methods for *change-point detection* try to detect abrupt changes in time-series data (Basseville & Nikiforov, 1993; Brodsky & Darkhovsky, 1993; Gustafsson, 2000; Nguyen et al., 2011). There are two types of problem settings in change-point detection, namely the real-time detection (Adams, 2007; Garnett et al., 2009; Paquet, 2007) and the retrospective detection (Basseville & Nikiforov, 1993; Yamanishi & Takeuchi, 2002). In retrospective detection, which requires longer reaction periods, Liu et al. (2012) proposed using techniques of direct DRE. Whereas the existing methods rely on linear-in-parameter models, our proposed method enables us to employ more complex models for change point detection.

**Similarity-based Sentiment Analysis:**    Kato (2019) used the density ratio estimated from PU learning for sentiment analysis of text data based on similarity.

# I  GENERALIZATION ERROR BOUND

The generalization error bound can be proved by building upon the proof techniques in Kiryo et al. (2017); Lu et al. (2020).

**Notations for the Theoretical Analysis:**    We denote the set of real values by $\mathbb{R}$ and that of positive integers by $\mathbb{N}$. Let $\mathcal{X} \subset \mathbb{R}^d$. Let $p_{\mathrm{nu}}(x)$ and $p_{\mathrm{de}}(x)$ be probability density functions over $\mathcal{X}$, and assume that the density ratio $r^*(x) := \frac{p_{\mathrm{nu}}(x)}{p_{\mathrm{de}}(x)}$ is existent and bounded: $\overline{R} := \|r^*\|_\infty < \infty$. Assume $0 < C < \frac{1}{\overline{R}}$. Since $\overline{R} \geq 1$ (because $1 = \int p_{\mathrm{de}}(x) r^*(x) dx \leq 1 \cdot \|r^*\|_\infty$), we have $C \in (0, 1]$ and hence $p_{\mathrm{mod}} := p_{\mathrm{de}} - C p_{\mathrm{nu}} > 0$.

**Problem Setup:**    Let the hypothesis class of density ratio be $\mathcal{H} \subset \{r : \mathbb{R}^D \to (b_r, B_r) =: I_r\}$, where $0 \leq b_r < \overline{R} < B_r$. Let $f : I_r \to \mathbb{R}$ be a twice continuously-differentiable convex function with a bounded derivative. Define $\tilde{f}$ by $\partial f(t) = C(\partial f(t)t - f(t)) + \tilde{f}(t)$, where $\partial f$ is the derivative of $f$ continuously extended to 0 and $B_r$. Recall the definitions $\ell_1(t) := \partial f(t)t - f(t) + A$, $\ell_2(t) :=$

$-\tilde{f}(t)$, and

$$\text{BR}_f(r) := \mathbb{E}_{\text{de}}\left[\partial f(r(X))r(X) - f(r(X)) + A\right] - \mathbb{E}_{\text{nu}}\left[\partial f(r(X))\right]$$

$$= \mathbb{E}\hat{\mathbb{E}}_{\text{mod}}\left[\partial f(r(X))r(X) - f(r(X)) + A\right] - \mathbb{E}_{\text{nu}}\left[\tilde{f}(r(X))\right]$$

$$= \mathbb{E}\hat{\mathbb{E}}_{\text{mod}}\ell_1(r(X)) + \mathbb{E}_{\text{nu}}\ell_2(r(X))$$

$$\left(= (\mathbb{E}_{\text{de}} - C\mathbb{E}_{\text{nu}})\ell_1(r(X)) + \mathbb{E}_{\text{nu}}\ell_2(r(X))\right),$$

$$\widehat{\text{nnBR}}_f(r) := \rho\left(\hat{\mathbb{E}}_{\text{mod}}\ell_1(r(X))\right) + \hat{\mathbb{E}}_{\text{nu}}\ell_2(r(X))$$

$$\left(= \rho((\hat{\mathbb{E}}_{\text{de}} - C\hat{\mathbb{E}}_{\text{nu}})\ell_1(r(X)) + \hat{\mathbb{E}}_{\text{nu}}\ell_2(r(X)))\right),$$

where we denoted $\hat{\mathbb{E}}_{\text{mod}} = \hat{\mathbb{E}}_{\text{de}} - C\hat{\mathbb{E}}_{\text{nu}}$ and $\rho$ is a consistent correction function with Lipschitz constant $L_\rho$ (Definition 1).

**Remark 4.** The true density ratio $r^*$ minimizes $\text{BR}_f$.

**Definition 1** (Consistent correction function Lu et al. (2020)). A function $f : \mathbb{R} \to \mathbb{R}$ is called a consistent correction function if it is Lipschitz continuous, non-negative and $f(x) = x$ for all $x \geq 0$.

**Definition 2** (Rademacher complexity). Given $n \in \mathbb{N}$ and a distribution $p$, define the Rademacher complexity $\mathcal{R}_n^p(\mathcal{H})$ of a function class $\mathcal{H}$ as

$$\mathcal{R}_n^p(\mathcal{H}) := \mathbb{E}_p\mathbb{E}_\sigma\left[\sup_{r \in \mathcal{H}}\left|\frac{1}{n}\sum_{i=1}^{n}\sigma_i r(X_i)\right|\right],$$

where $\{\sigma_i\}_{i=1}^n$ are Rademacher variables (i.e., independent variables following the uniform distribution over $\{-1, +1\}$) and $\{X_i\}_{i=1}^n \overset{\text{i.i.d.}}{\sim} p$.

The theorem in the paper is a special case of Theorem 3 with $\rho(\cdot) := \max\{0, \cdot\}$ (in which case $L_\rho = 1$) and Theorem 4.

**Theorem 3** (Generalization error bound). *Assume that $B_\ell := \sup_{t \in I_r}\{\max\{|\ell_1(t)|, |\ell_2(t)|\}\} < \infty$. Assume $\ell_1$ is $L_{\ell_1}$-Lipschitz and $\ell_2$ is $L_{\ell_2}$-Lipschitz. Assume that there exists an empirical risk minimizer $\hat{r} \in \arg\min_{r \in \mathcal{H}} \widehat{\text{nnBR}}_f(r)$ and a population risk minimizer $\bar{r} \in \arg\min_{r \in \mathcal{H}} \text{BR}_f(r)$. Also assume $\inf_{r \in \mathcal{H}} \mathbb{E}\hat{\mathbb{E}}_{\text{mod}}\ell_1(r(X)) > 0$ and that $(\rho - \text{Id})$ is $(L_{\rho-\text{Id}})$-Lipschitz. Then for any $\delta \in (0, 1)$, with probability at least $1 - \delta$, we have*

$$\text{BR}_f(\hat{r}) - \text{BR}_f(\bar{r}) \leq 8L_\rho L_{\ell_1}\mathcal{R}_{n_{\text{de}}}^{p_{\text{de}}}(\mathcal{H}) + 8(L_\rho C L_{\ell_1} + L_{\ell_2})\mathcal{R}_{n_{\text{nu}}}^{p_{\text{nu}}}(\mathcal{H})$$

$$+ 2\Phi_{(C,f,\rho)}(n_{\text{nu}}, n_{\text{de}}) + B_\ell\sqrt{8\left(\frac{L_\rho^2}{n_{\text{de}}} + \frac{(1+L_\rho C)^2}{n_{\text{nu}}}\right)\log\frac{1}{\delta}},$$

*where $\Phi_{(C,f,\rho)}(n_{\text{nu}}, n_{\text{de}})$ is defined as in Lemma 2.*

*Proof.* Since $\hat{r}$ minimizes $\widehat{\text{nnBR}}_f$, we have

$$\text{BR}_f(\hat{r}) - \text{BR}_f(\bar{r}) = \text{BR}_f(\hat{r}) - \widehat{\text{nnBR}}_f(\hat{r}) + \widehat{\text{nnBR}}_f(\hat{r}) - \text{BR}_f(\bar{r})$$

$$\leq \text{BR}_f(\hat{r}) - \widehat{\text{nnBR}}_f(\hat{r}) + \widehat{\text{nnBR}}_f(\bar{r}) - \text{BR}_f(\bar{r})$$

$$\leq 2\sup_{r \in \mathcal{H}}|\widehat{\text{nnBR}}_f(r) - \text{BR}_f(r)|$$

$$\leq \underbrace{2\sup_{r \in \mathcal{H}}|\widehat{\text{nnBR}}_f(r) - \mathbb{E}\widehat{\text{nnBR}}_f(r)|}_{\text{Maximal deviation}} + \underbrace{2\sup_{r \in \mathcal{H}}|\mathbb{E}\widehat{\text{nnBR}}_f(r) - \text{BR}_f(r)|}_{\text{Bias}}.$$

We apply McDiarmid's inequality McDiarmid (1989); Mohri et al. (2018) to the maximal deviation term. The absolute value of the difference caused by altering one data point in the maximal deviation term is bounded from above by $2B_\ell\frac{L_\rho}{n_{\text{de}}}$ if the altered point is a sample from $p_{\text{de}}$ and $2B_\ell\frac{1+L_\rho C}{n_{\text{nu}}}$ if

it is from $p_{\mathrm{nu}}$. Therefore, McDiarmid's inequality implies, with probability at least $1 - \delta$, that we have

$$\sup_{r \in \mathcal{H}} |\widehat{\mathrm{nnBR}}_f(r) - \mathbb{E}\widehat{\mathrm{nnBR}}_f(r)|$$

$$\leq \underbrace{\mathbb{E}\left[\sup_{r \in \mathcal{H}} |\widehat{\mathrm{nnBR}}_f(r) - \mathbb{E}\widehat{\mathrm{nnBR}}_f(r)|\right]}_{\text{Expected maximal deviation}} + B_\ell \sqrt{2\left(\frac{L_\rho^2}{n_{\mathrm{de}}} + \frac{(1 + L_\rho C)^2}{n_{\mathrm{nu}}}\right) \log \frac{1}{\delta}}.$$

Applying Lemma 1 to the expected maximal deviation term and Lemma 2 to the bias term, we obtain the assertion. $\qquad \square$

The following lemma generalizes the symmetrization lemmas proved in Kiryo et al. (2017) and Lu et al. (2020).

**Lemma 1** (Symmetrization under Lipschitz-continuous modification). *Let* $0 \leq a < b$, $J \in \mathbb{N}$, *and* $\{K_j\}_{j=1}^J \subset \mathbb{N}$. *Given i.i.d. samples* $\mathcal{D}_{(j,k)} := \{X_i\}_{i=1}^{n_{(j,k)}}$ *each from a distribution* $p_{(j,k)}$ *over* $\mathcal{X}$, *consider a stochastic process* $\hat{S}$ *indexed by* $\mathcal{F} \subset (a,b)^{\mathcal{X}}$ *of the form*

$$\hat{S}(f) = \sum_{j=1}^J \rho_j \left(\sum_{k=1}^{K_j} \hat{\mathbb{E}}_{(i,j)}[\ell_{(j,k)}(f(X))]\right),$$

*where each* $\rho_j$ *is a* $L_{\rho_j}$*-Lipschitz function on* $\mathbb{R}$, $\ell_{(j,k)}$ *is a* $L_{\ell_{(j,k)}}$*-Lipschitz function on* $(a, b)$, *and* $\hat{\mathbb{E}}_{(i,j)}$ *denotes the expectation with respect to the empirical measure of* $\mathcal{D}_{(j,k)}$. *Denote* $S(f) := \mathbb{E}\hat{S}(f)$ *where* $\mathbb{E}$ *is the expectation with respect to the product measure of* $\{\mathcal{D}_{(j,k)}\}_{(j,k)}$. *Here, the index* $j$ *denotes the grouping of terms due to* $\rho_j$, *and* $k$ *denotes each sample average term. Then we have*

$$\mathbb{E}\sup_{f \in \mathcal{F}} |\hat{S}(f) - S(f)| \leq 4 \sum_{j=1}^J \sum_{k=1}^{K_j} L_{\rho_j} L_{\ell_{(j,k)}} \mathcal{R}_{n_{(j,k)}, p_{(j,k)}}(\mathcal{F}).$$

*Proof.* First, we consider a continuous extension of $\ell_{(j,k)}$ defined on $(a, b)$ to $[0, b)$. Since the functions in $\mathcal{F}$ take values only in $(a, b)$, this extension can be performed without affecting the values of $\hat{S}(f)$ or $S(f)$. We extend the function by defining the values for $x \in [0, a]$ as $\ell_{(j,k)}(x) := \lim_{x' \downarrow a} \ell_{(j,k)}(x')$, where the right-hand side is guaranteed to exist since $\ell_{(j,k)}$ is Lipschitz continuous hence uniformly continuous. Then, $\ell_{(j,k)}$ remains a $L_{\rho_j}$-Lipschitz continuous function on $[0, b)$. Now we perform symmetrization (Vapnik, 1998), deal with $\rho_j$'s, and then bound the symmetrized process by Rademacher complexity. Denoting independent copies of $\{X_{(j,k)}\}$ by $\{X_{j,k}^{(\mathrm{gh})}\}_{(j,k)}$ and

the corresponding expectations as well as the sample averages with $^{(\text{gh})}$,

$$\mathbb{E}\sup_{f\in\mathcal{F}}|\hat{S}(f)-S(f)|$$

$$\leq \sum_{j=1}^{J}\mathbb{E}\sup_{f\in\mathcal{F}}|\rho_j(\sum_{k=1}^{K_j}\hat{\mathbb{E}}_{(i,j)}\ell_{(j,k)}(f(X)))-\mathbb{E}^{(\text{gh})}\rho_j(\sum_{k=1}^{K_j}\hat{\mathbb{E}}_{(j,k)}^{(\text{gh})}\ell_{(j,k)}(f(X^{(\text{gh})})))|$$

$$\leq \sum_{j=1}^{J}\mathbb{E}\mathbb{E}^{(\text{gh})}\sup_{f\in\mathcal{F}}|\rho_j(\sum_{k=1}^{K_j}\hat{\mathbb{E}}_{(i,j)}\ell_{(j,k)}(f(X)))-\rho_j(\sum_{k=1}^{K_j}\hat{\mathbb{E}}_{(j,k)}^{(\text{gh})}\ell_{(j,k)}(f(X^{(\text{gh})})))|$$

$$\leq \sum_{j=1}^{J}L_{\rho_j}\sum_{k=1}^{K_j}\mathbb{E}\mathbb{E}^{(\text{gh})}\sup_{f\in\mathcal{F}}|\hat{\mathbb{E}}_{(i,j)}\ell_{(j,k)}(f(X))-\hat{\mathbb{E}}_{(j,k)}^{(\text{gh})}\ell_{(j,k)}(f(X^{(\text{gh})}))|$$

$$= \sum_{j=1}^{J}L_{\rho_j}\sum_{k=1}^{K_j}\mathbb{E}\mathbb{E}^{(\text{gh})}\sup_{f\in\mathcal{F}}|\hat{\mathbb{E}}_{(i,j)}(\ell_{(j,k)}(f(X))-\ell_{(j,k)}(0))-\hat{\mathbb{E}}_{(j,k)}^{(\text{gh})}(\ell_{(j,k)}(f(X^{(\text{gh})}))-\ell_{(j,k)}(0))|$$

$$\leq \sum_{j=1}^{J}L_{\rho_j}\sum_{k=1}^{K_j}\left(2\mathcal{R}_{n_{(j,k)},p_{(j,k)}}(\{\ell_{(j,k)}\circ f-\ell_{(j,k)}(0):f\in\mathcal{F}\})\right)$$

$$\leq \sum_{j=1}^{J}L_{\rho_j}\sum_{k=1}^{K_j}2\cdot 2L_{\ell_{(j,k)}}\mathcal{R}_{n_{(j,k)},p_{(j,k)}}(\mathcal{F}),$$

where we applied Talagrand's contraction lemma for two-sided Rademacher complexity (Ledoux & Talagrand, 1991; Bartlett & Mendelson, 2001) with respect to $(t\mapsto\ell_{(j,k)}(t)-\ell_{(j,k)}(0))$ in the last inequality. $\qquad\square$

**Lemma 2** (Bias due to risk correction). *Assume* $\inf_{r\in\mathcal{H}}\mathbb{E}\hat{\mathbb{E}}_{\text{mod}}\ell_1(r(X)) > 0$ *and that* $(\rho-\text{Id})$ *is* $(L_{\rho-\text{Id}})$*-Lipschitz on* $\mathbb{R}$*. There exists* $\alpha > 0$ *such that*

$$\sup_{r\in\mathcal{H}}|\widehat{\mathbb{E}\text{nnBR}}_f(r)-\text{BR}_f(r)| \leq (1+C)B_\ell L_{\rho-\text{Id}}\exp\left(-\frac{2\alpha^2}{(B_\ell^2/n_{\text{de}})+(C^2B_\ell^2/n_{\text{nu}})}\right)$$

$$=:\Phi_{(C,f,\rho)}(n_{\text{nu}},n_{\text{de}}).$$

**Remark 5.** Note that we already have $p_{\text{mod}}\geq 0$ and $\ell_1\geq 0$ and hence $\inf_{r\in\mathcal{H}}\mathbb{E}\hat{\mathbb{E}}_{\text{mod}}\ell_1(r(X))\geq 0$. Therefore, the assumption of Lemma 2 is essentially referring to the strict positivity of the infimum. Here, $\mathbb{E}\hat{\mathbb{E}}_{\text{mod}}$ and $\mathbb{P}(\cdot)$ denote the expectation and the probability with respect to the joint distribution of the samples included in $\hat{\mathbb{E}}_{\text{mod}}$.

*Proof.* Fix an arbitrary $r\in\mathcal{H}$. We have

$$|\widehat{\mathbb{E}\text{nnBR}}_f(r)-\text{BR}_f(r)| = |\mathbb{E}[\widehat{\text{nnBR}}_f(r)-\widehat{\text{BR}}_f(r)]|$$

$$= |\mathbb{E}[\rho(\hat{\mathbb{E}}_{\text{mod}}\ell_1(r(X)))-\hat{\mathbb{E}}_{\text{mod}}\ell_1(r(X))]| \leq \mathbb{E}\left[|\rho(\hat{\mathbb{E}}_{\text{mod}}\ell_1(r(X)))-\hat{\mathbb{E}}_{\text{mod}}\ell_1(r(X))|\right]$$

$$= \mathbb{E}\left[\mathbb{1}\{\rho(\hat{\mathbb{E}}_{\text{mod}}\ell_1(r(X)))\neq\hat{\mathbb{E}}_{\text{mod}}\ell_1(r(X))\}\cdot|\rho(\hat{\mathbb{E}}_{\text{mod}}\ell_1(r(X)))-\hat{\mathbb{E}}_{\text{mod}}\ell_1(r(X))|\right]$$

$$\leq \mathbb{E}\left[\mathbb{1}\{\rho(\hat{\mathbb{E}}_{\text{mod}}\ell_1(r(X)))\neq\hat{\mathbb{E}}_{\text{mod}}\ell_1(r(X))\}\right]\left(\sup_{s:|s|\leq(1+C)B_\ell}|\rho(s)-s|\right)$$

where $\mathbb{1}\{\cdot\}$ denotes the indicator function, and we used $|\hat{\mathbb{E}}_{\text{mod}}\ell_1(r(X))|\leq(1+C)B_\ell$. Further, we have

$$\sup_{s:|s|\leq(1+C)B_\ell}|\rho(s)-s| \leq \sup_{s:|s|\leq(1+C)B_\ell}|(\rho-\text{Id})(s)-(\rho-\text{Id})(0)|+|(\rho-\text{Id})(0)|$$

$$\leq \sup_{s:|s|\leq(1+C)B_\ell}L_{\rho-\text{Id}}|s-0|+0 \leq (1+C)B_\ell L_{\rho-\text{Id}},$$

where Id denotes the identity function. On the other hand, since $\inf_{r \in \mathcal{H}} \mathbb{E}\hat{\mathbb{E}}_{\mathrm{mod}}\ell_1(r(X)) > 0$ is assumed, there exists $\alpha > 0$ such that for any $r \in \mathcal{H}$, $\mathbb{E}\hat{\mathbb{E}}_{\mathrm{mod}}\ell_1(r(X)) > \alpha$. Therefore, denoting the support of a function by $\mathrm{supp}(\cdot)$,

$$\mathbb{E}\left[\mathbb{1}\{\rho(\hat{\mathbb{E}}_{\mathrm{mod}}\ell_1(r(X))) \neq \hat{\mathbb{E}}_{\mathrm{mod}}\ell_1(r(X))\}\right] = \mathbb{P}\left(\hat{\mathbb{E}}_{\mathrm{mod}}\ell_1(r(X)) \in \mathrm{supp}(\rho - \mathrm{Id})\right)$$

$$\leq \mathbb{P}\left(\hat{\mathbb{E}}_{\mathrm{mod}}\ell_1(r(X)) < 0\right) \leq \mathbb{P}\left(\hat{\mathbb{E}}_{\mathrm{mod}}\ell_1(r(X)) < \mathbb{E}\hat{\mathbb{E}}_{\mathrm{mod}}\ell_1(r(X)) - \alpha\right)$$

holds. Now we apply McDiarmid's inequality to the right-most quantity. The absolute difference caused by altering one data point in $\hat{\mathbb{E}}_{\mathrm{mod}}\ell_1(r(X))$ is bounded by $\frac{B_\ell}{n_{\mathrm{de}}}$ if the change is in a sample from $p_{\mathrm{de}}$ and $\frac{CB_\ell}{n_{\mathrm{nu}}}$ otherwise. Therefore, McDiarmid's inequality implies

$$\mathbb{P}\left(\hat{\mathbb{E}}_{\mathrm{mod}}\ell_1(r(X)) < \mathbb{E}\hat{\mathbb{E}}_{\mathrm{mod}}\ell_1(r(X)) - \alpha\right) \leq \exp\left(-\frac{2\alpha^2}{(B_\ell{}^2/n_{\mathrm{de}}) + (C^2 B_\ell{}^2/n_{\mathrm{nu}})}\right).$$

$\square$

**Theorem 4** (Generalization error bound). *Under Assumption 3, for any $\delta \in (0,1)$, with probability at least $1 - \delta$, we have $\mathrm{BR}_f(\hat{r}) - \mathrm{BR}_f(\bar{r}) \leq L_{\ell_1}\mathcal{R}_{n_{\mathrm{de}}}^{p_{\mathrm{de}}}(\mathcal{H}) + 8(CL_{\ell_1} + L_{\ell_2})\mathcal{R}_{n_{\mathrm{nu}}}^{p_{\mathrm{nu}}}(\mathcal{H}) + 2\Phi_C^f(n_{\mathrm{nu}}, n_{\mathrm{de}}) + B_\ell\sqrt{8\left(\frac{1}{n_{\mathrm{de}}} + \frac{(1+C)^2}{n_{\mathrm{nu}}}\right)\log\frac{1}{\delta}}$, where $\Phi_C^f(n_{\mathrm{nu}}, n_{\mathrm{de}}) := (1 + C)B_\ell\exp\left(-\frac{2\alpha^2}{(B_\ell{}^2/n_{\mathrm{de}}) + (C^2 B_\ell{}^2/n_{\mathrm{nu}})}\right)$ and $\alpha > 0$ is a constant determined in the proof of Lemma 2 in Appendix I.*

**Remark 6** (Explicit form of the bound in Theorem 1). Here, we show the explicit form of the bound in Theorem 1 as follows:

$$\mathrm{BR}_f(\hat{r}) - \mathrm{BR}_f(\bar{r})$$

$$\leq \frac{\kappa_1}{\sqrt{n_{\mathrm{de}}}} + \frac{\kappa_2}{\sqrt{n_{\mathrm{nu}}}} + 2\Phi_C^f(n_{\mathrm{nu}}, n_{\mathrm{de}}) + B_\ell\sqrt{8\left(\frac{1}{n_{\mathrm{de}}} + \frac{(1+C)^2}{n_{\mathrm{nu}}}\right)\log\frac{1}{\delta}}$$

$$= L_{\ell_1}\frac{B_{p_{\mathrm{de}}}\left(\sqrt{2\log(2)L} + 1\right)\prod_{j=1}^{L} B_{W_j}}{\sqrt{n_{\mathrm{de}}}}$$

$$+ 8(CL_{\ell_1} + L_{\ell_2})\frac{B_{p_{\mathrm{nu}}}\left(\sqrt{2\log(2)L} + 1\right)\prod_{j=1}^{L} B_{W_j}}{\sqrt{n_{\mathrm{nu}}}}$$

$$+ (1 + C)B_\ell\exp\left(-\frac{2\alpha^2}{(B_\ell{}^2/n_{\mathrm{de}}) + (C^2 B_\ell{}^2/n_{\mathrm{nu}})}\right)$$

$$+ B_\ell\sqrt{8\left(\frac{1}{n_{\mathrm{de}}} + \frac{(1+C)^2}{n_{\mathrm{nu}}}\right)\log\frac{1}{\delta}}.$$

## J  RADEMACHER COMPLEXITY BOUND

The following lemma provides an upper-bound on the Rademacher complexity for multi-layer perceptron models in terms of the Frobenius norms of the parameter matrices. Alternatively, other approaches to bound the Rademacher complexity can be employed. The assertion of the lemma follows immediately from the proof of Theorem 1 of Golowich et al. (2019) after a slight modification to incorporate the absolute value function in the definition of Rademacher complexity.

**Lemma 3** (Rademacher complexity bound (Golowich et al., 2019, Theorem 1)). *Assume the distribution $p$ has a bounded support: $B_p := \sup_{x \in \mathrm{supp}(p)} \|x\| < \infty$. Let $\mathcal{H}$ be the class of real-valued networks of depth $L$ over the domain $\mathcal{X}$, where each parameter matrix $W_j$ has Frobenius norm at most $B_{W_j} \geq 0$, and with 1-Lipschitz activation functions $\varphi_j$ which are positive-homogeneous (i.e., $\varphi_j$ is applied element-wise and $\varphi_j(\alpha t) = \alpha\varphi_j(t)$ for all $\alpha \geq 0$). Then*

$$\mathcal{R}_n^p(\mathcal{H}) \leq \frac{B_p\left(\sqrt{2\log(2)L} + 1\right)\prod_{j=1}^{L} B_{W_j}}{\sqrt{n}}.$$

*Proof.* The assertion immediately follows once we modify the beginning of the proof of Theorem 1 by introducing the absolute value function inside the supremum of the Rademacher complexity as

$$\mathbb{E}_\sigma \left[ \sup_{r \in \mathcal{H}} \left| \sum_{i=1}^n \sigma_i r(x_i) \right| \right] \leq \frac{1}{\lambda} \log \mathbb{E}_\sigma \sup_{r \in \mathcal{H}} \exp \left( \lambda \left| \sum_{i=1}^n \sigma_i r(x_i) \right| \right).$$

for $\lambda > 0$. The rest of the proof is identical to that of Theorem 1 of Golowich et al. (2019). $\qquad\square$

## K  PROOF OF THEOREM 2

We consider relating the $L^2$ error bound to the BR divergence generalization error bound in the following lemma.

**Lemma 4** ($L^2$ distance bound). *Let $\mathcal{H} := \{r : \mathcal{X} \to (b_r, B_r) =: I_r | \int |r(x)|^2 \mathrm{d}\boldsymbol{x} < \infty\}$ and assume $r^* \in \mathcal{H}$. If $\inf_{t \in I_r} f''(t) > 0$, then there exists $\mu > 0$ such that for all $r \in \mathcal{H}$,*

$$\|r - r^*\|_{L^2(p_{\mathrm{de}})}^2 \leq \frac{2}{\mu} \left( \mathrm{BR}_f(r) - \mathrm{BR}_f(r^*) \right)$$

*holds.*

*Proof.* Since $\mu := \inf_{t \in I_r} f''(t) > 0$, the function $f$ is $\mu$-strongly convex. By the definition of strong convexity,

$$\mathrm{BR}_f(r) - \mathrm{BR}_f(r^*) = (\mathrm{BR}_f(r) - \mathbb{E}_{\mathrm{de}} f(r^*(X))) - \underbrace{(\mathrm{BR}_f(r^*) + \mathbb{E}_{\mathrm{de}} f(r^*(X)))}_{= 0}$$

$$= \mathbb{E}_{\mathrm{de}} \left[ f(r^*(X)) - f(r(X)) + \partial f(r(X))(r^*(X) - r(X)) \right]$$

$$\geq \mathbb{E}_{\mathrm{de}} \left[ \frac{\mu}{2} (r^*(X) - r(X))^2 \right] = \frac{\mu}{2} \|r^* - r\|_{L^2(p_{\mathrm{de}})}^2.$$

$\square$

**Lemma 5** ($\ell_2$ distance bound). *Fix $r \in \mathcal{H}$. Given $n$ samples $\{x_i\}_{i=1}^n$ from $p_{\mathrm{de}}$, with probability at least $1 - \delta$, we have*

$$\frac{1}{n} \sum_{i=1}^n (r(x_i) - r^*(x_i))^2 \leq \underbrace{\mathbb{E} \left[ (r - r^*)^2 (X) \right]}_{= \|r - r^*\|_{L^2(p_{\mathrm{de}})}^2} + (2\overline{R})^2 \sqrt{\frac{\log \frac{1}{\delta}}{2n}}.$$

*Proof.* The assertion follows from McDiarmid's inequality after noting that altering one sample results in an absolute change bounded by $\frac{1}{n}(2\overline{R})^2$. $\qquad\square$

Thus, a generalization error bound in terms of $\mathrm{BR}_f$ can be converted to that of an $L^2$ distance when the true density ratio and the density ratio model are square-integrable and $f$ is strongly convex. However, when using the result of Theorem 1, the convergence rate shown here is slower than $\mathcal{O}_{\mathbb{P}} \left( (\min \{n_{\mathrm{de}}, n_{\mathrm{nu}}\})^{-1/(4)} \right)$. On the other hand, Kanamori et al. (2012) derived $\mathcal{O}_{\mathbb{P}} \left( (\min \{n_{\mathrm{de}}, n_{\mathrm{nu}}\})^{-1/(2+\gamma)} \right)$ convergence rate. To derive this bound when using neural network, we need to restrict the neural network models. In the following part, we prove Theorem 2 for the following hypothesis class $\mathcal{H}$.

**Definition 3** (ReLU neural networks; Schmidt-Hieber, 2020). *For $L \in \mathbb{N}$ and $p = (p_0, \ldots, p_{L+1}) \in \mathbb{N}^{L+2}$,*

$$\mathcal{F}(L, p) := \{f : x \mapsto W_L \sigma_{v_L} W_{L-1} \sigma_{v_{L-1}} \cdots W_1 \sigma_{v_1} W_0 x :$$

$$W_i \in \mathbb{R}^{p_{i+1} \times p_i}, v_i \in \mathbb{R}^{p_i} (i = 0, \ldots, L)\},$$

*where $\sigma_v(y) := \sigma(y - v)$, and $\sigma(\cdot) = \max\{\cdot, 0\}$ is applied in an element-wise manner. Then, for $s \in \mathbb{N}, F \geq 0, L \in \mathbb{N}$, and $p \in \mathbb{N}^{L+2}$, define*

$$\mathcal{H}(L, p, s, F) := \{f \in \mathcal{F}(L, p) : \sum_{j=0}^L \|W_j\|_0 + \|v_j\|_0 \leq s, \|f\|_\infty \leq F\},$$

where $\|\cdot\|_0$ denotes the number of non-zero entries of the matrix or the vector, and $\|\cdot\|_\infty$ denotes the supremum norm. Now, fixing $\bar{L}, \bar{p}, s \in \mathbb{N}$ as well as $F > 0$, we define

$$\mathrm{Ind}_{\bar{L},\bar{p}} := \{(L,p) : L \in \mathbb{N}, L \le \bar{L}, p \in [\bar{p}]^{L+2}\},$$

and we consider the hypothesis class

$$\bar{\mathcal{H}} := \bigcup_{(L,p) \in \mathrm{Ind}_{\bar{L},\bar{p}}} \mathcal{H}(L,p,s,F)$$

$$\mathcal{H} := \{r \in \bar{\mathcal{H}} : \mathrm{Im}(r) \subset (b_r, B_r)\}.$$

Moreover, we define $I_1 : \mathrm{Ind}_{\bar{L},\bar{p}} \to \mathbb{R}$ and $I : \mathcal{H} \to [0, \infty)$ by

$$I_1(L,p) := 2|\mathrm{Ind}_{\bar{L},\bar{p}}|^{\frac{1}{s+1}}(L+1)V^2,$$

$$I(r) := \max \left\{ \|r\|_\infty, \min_{\substack{(L,p) \in \mathrm{Ind}_{\bar{L},\bar{p}} \\ r \in \mathcal{H}(L,p,s,F)}} I_1(L,p) \right\},$$

where $V := \prod_{l=0}^{L+1}(p_l + 1)$, and we define

$$\mathcal{H}_M := \{r \in \mathcal{H} : I(r) \le M\}.$$

Note that the requirement for the hypothesis class of Theorem 1 is not as tight as that of Theorem 2. Then, we prove Theorem 2 as follows:

*Proof.* Thanks to the strong convexity, by Lemma 4, we have

$$\frac{\mu}{2}\|\hat{r} - r^*\|^2_{L^2(p_{\mathrm{de}})} \le \mathrm{BR}_f(\hat{r}) - \mathrm{BR}_f(r^*)$$
$$= \mathrm{BR}_f(\hat{r}) - \mathrm{BR}_f(r^*)$$
$$\underbrace{-\widehat{\mathrm{BR}}_f(\hat{r}) + \widehat{\mathrm{BR}}_f(\hat{r})}_{=\,0} \underbrace{-\widehat{\mathrm{nnBR}}_f(\hat{r}) + \widehat{\mathrm{nnBR}}_f(\hat{r})}_{=\,0} \underbrace{-\widehat{\mathrm{BR}}_f(r^*) + \widehat{\mathrm{BR}}_f(r^*)}_{=\,0}$$
$$\le \mathrm{BR}_f(\hat{r}) - \widehat{\mathrm{BR}}_f(\hat{r}) + (\widehat{\mathrm{BR}}_f(\hat{r}) - \widehat{\mathrm{nnBR}}_f(\hat{r}))$$
$$+ (\widehat{\mathrm{nnBR}}_f(r^*) - \widehat{\mathrm{BR}}_f(r^*)) + \widehat{\mathrm{BR}}_f(r^*) - \mathrm{BR}_f(r^*)$$
$$\le \underbrace{(\mathrm{BR}_f(\hat{r}) - \mathrm{BR}_f(r^*) + \widehat{\mathrm{BR}}_f(r^*) - \widehat{\mathrm{BR}}_f(\hat{r}))}_{=:\,A} + \underbrace{2\sup_{r \in \mathcal{H}}|\widehat{\mathrm{BR}}_f(r) - \widehat{\mathrm{nnBR}}_f(r)|}_{=:\,B},$$

where we used $\widehat{\mathrm{nnBR}}_f(\hat{r}) \le \widehat{\mathrm{nnBR}}_f(r^*)$. To bound $A$, for ease of notation, let $\ell_1^r = \ell_1(r(X))$ and $\ell_2^r = \ell_2(r(X))$. Then, since

$$\mathrm{BR}_f(r) = \mathbb{E}_{\mathrm{de}}\ell_1(r(X)) - C\mathbb{E}_{\mathrm{nu}}\ell_1(r(X)) + \mathbb{E}_{\mathrm{nu}}\ell_2(r(X)),$$
$$\widehat{\mathrm{BR}}_f(r) = \hat{\mathbb{E}}_{\mathrm{de}}\ell_1(r(X)) - C\hat{\mathbb{E}}_{\mathrm{nu}}\ell_1(r(X)) + \hat{\mathbb{E}}_{\mathrm{nu}}\ell_2(r(X)),$$

we have

$$A = \mathrm{BR}_f(\hat{r}) - \mathrm{BR}_f(r^*) + \widehat{\mathrm{BR}}_f(r^*) - \widehat{\mathrm{BR}}_f(\hat{r})$$
$$= (\mathbb{E}_{\mathrm{de}} - \hat{\mathbb{E}}_{\mathrm{de}})(\ell_1^{\hat{r}} - \ell_1^{r^*}) - C(\mathbb{E}_{\mathrm{nu}} - \hat{\mathbb{E}}_{\mathrm{nu}})(\ell_1^{\hat{r}} - \ell_1^{r^*}) + (\mathbb{E}_{\mathrm{nu}} - \hat{\mathbb{E}}_{\mathrm{nu}})(\ell_2^{\hat{r}} - \ell_2^{r^*})$$
$$\le |(\mathbb{E}_{\mathrm{de}} - \hat{\mathbb{E}}_{\mathrm{de}})(\ell_1^{\hat{r}} - \ell_1^{r^*})| + C|(\mathbb{E}_{\mathrm{nu}} - \hat{\mathbb{E}}_{\mathrm{nu}})(\ell_1^{\hat{r}} - \ell_1^{r^*})| + |(\mathbb{E}_{\mathrm{nu}} - \hat{\mathbb{E}}_{\mathrm{nu}})(\ell_2^{\hat{r}} - \ell_2^{r^*})|$$

By applying Lemma 10, for any $0 < \gamma < 2$, we have

$$A \le \mathcal{O}_{\mathbb{P}}\left(\max\left\{\frac{\|\hat{r} - r^*\|^{1-\gamma/2}_{L^2(p_{\mathrm{de}})}}{\sqrt{\min\{n_{\mathrm{de}}, n_{\mathrm{nu}}\}}}, \frac{1}{(\min\{n_{\mathrm{de}}, n_{\mathrm{nu}}\})^{2/(2+\gamma)}}\right\}\right).$$

On the other hand, by Lemma 12 and Lemma 7, and the assumption $\inf_{r \in \mathcal{H}} \mathbb{E}\hat{\mathbb{E}}_{\mathrm{mod}} \ell_1(r(X)) > 0$, there exists $\alpha > 0$ such that we have $B \leq \mathcal{O}_{\mathbb{P}}\left(\exp\left(-\frac{2\alpha^2}{(B_\ell{}^2/n_{\mathrm{de}})+(C^2 B_\ell{}^2/n_{\mathrm{nu}})}\right)\right)$. Combining the above bounds on $A$ and $B$, for any $0 < \gamma < 2$, we get

$$\|\hat{r} - r^*\|_{L^2(p_{\mathrm{de}})}^2 \leq \mathcal{O}_{\mathbb{P}}\left(\max\left\{\frac{\|\hat{r} - r^*\|_{L^2(p_{\mathrm{de}})}^{1-\gamma/2}}{\sqrt{\min\{n_{\mathrm{de}}, n_{\mathrm{nu}}\}}}, \frac{1}{(\min\{n_{\mathrm{de}}, n_{\mathrm{nu}}\})^{2/(2+\gamma)}}\right\}\right)$$

$$+ \mathcal{O}_{\mathbb{P}}\left(\exp\left(-\frac{2\alpha^2}{(B_\ell{}^2/n_{\mathrm{de}})+(C^2 B_\ell{}^2/n_{\mathrm{nu}})}\right)\right)$$

$$\leq \mathcal{O}_{\mathbb{P}}\left(\max\left\{\frac{\|\hat{r} - r^*\|_{L^2(p_{\mathrm{de}})}^{1-\gamma/2}}{\sqrt{\min\{n_{\mathrm{de}}, n_{\mathrm{nu}}\}}}, \frac{1}{(\min\{n_{\mathrm{de}}, n_{\mathrm{nu}}\})^{2/(2+\gamma)}}\right\}\right).$$

As a result, we have

$$\|\hat{r} - r^*\|_{L^2(p_{\mathrm{de}})} \leq \mathcal{O}_{\mathbb{P}}\left((\min\{n_{\mathrm{de}}, n_{\mathrm{nu}}\})^{-\frac{1}{2+\gamma}}\right).$$

$\square$

Each lemma used in the proof is provided as follows.

### K.1 Complexity of the hypothesis class

For the function classes in Definition 3, we have the following evaluations of their complexities.

**Lemma 6** (Lemma 5 in Schmidt-Hieber (2020)). *For $L \in \mathbb{N}$ and $p \in \mathbb{N}^{L+2}$, let $V := \prod_{l=0}^{L+1}(p_l+1)$. Then, for any $\delta > 0$,*

$$\log \mathcal{N}(\delta, \mathcal{H}(L, p, s, \infty), \|\cdot\|_\infty) \leq (s+1)\log(2\delta^{-1}(L+1)V^2).$$

**Lemma 7.** *There exists $c > 0$ such that*

$$\mathcal{R}_{n_{\mathrm{nu}}}^{p_{\mathrm{nu}}}(\mathcal{H}) \leq cn_{\mathrm{nu}}^{-1/2}, \quad \mathcal{R}_{n_{\mathrm{de}}}^{p_{\mathrm{de}}}(\mathcal{H}) \leq cn_{\mathrm{de}}^{-1/2}.$$

*Proof.* By Dudley's entropy integral bound (Wainwright, 2019, Theorem 5.22) and Lemma 6, we have

$$\mathcal{R}_{n_{\mathrm{nu}}}^{p_{\mathrm{nu}}}(\mathcal{H}(L, p, s, F)) \leq 32 \int_0^{2F} \sqrt{\frac{\log \mathcal{N}(\delta, \mathcal{H}(L, p, s, F), \|\cdot\|_\infty)}{n_{\mathrm{nu}}}} d\delta$$

$$= \left(32 \int_0^{2F} \left((s+1)\log(2\delta^{-1}(L+1)V^2)\right)^{1/2} d\delta\right) n_{\mathrm{nu}}^{-1/2}.$$

Therefore, there exists $c > 0$ such that

$$\mathcal{R}_{n_{\mathrm{nu}}}^{p_{\mathrm{nu}}}(\mathcal{H}) \leq \sum_{(L,p) \in \mathrm{Ind}_{\bar{L}, \bar{p}}} \mathcal{R}_{n_{\mathrm{nu}}}^{p_{\mathrm{nu}}}(\mathcal{H}(L, p, s, F)) \leq cn_{\mathrm{nu}}^{-1/2}.$$

The same argument applies to $\mathcal{R}_{n_{\mathrm{de}}}^{p_{\mathrm{de}}}(\mathcal{H})$, and we obtain the assertion. $\square$

**Lemma 8.** *There exists $c_0 > 0$ such that for any $\gamma > 0$, any $\delta > 0$, and any $M \geq 1$, we have*

$$\log \mathcal{N}(\delta, \mathcal{H}_M, \|\cdot\|_\infty) \leq \frac{s+1}{\gamma}\left(\frac{M}{\delta}\right)^\gamma.$$

*and*

$$\sup_{r \in \mathcal{H}_M} \|r - r^*\|_\infty \leq c_0 M.$$

*Proof.* The first assertion is a result of the following calculation:

$$\log \mathcal{N}\left(\delta, \mathcal{H}_M, \|\cdot\|_\infty\right) \leq \log \sum_{\substack{(L,p)\in \mathrm{Ind}_{\bar{L},\bar{p}} \\ I_1(L,p)\leq M}} \mathcal{N}\left(\delta, \mathcal{H}(L,p,s,M), \|\cdot\|_\infty\right)$$

$$\leq \log \sum_{\substack{(L,p)\in \mathrm{Ind}_{\bar{L},\bar{p}} \\ I_1(L,p)\leq M}} \left(\frac{2}{\delta}(L+1)V^2\right)^{s+1}$$

$$\leq \log |\mathrm{Ind}_{\bar{L},\bar{p}}| \left(\frac{1}{\delta}M|\mathrm{Ind}_{\bar{L},\bar{p}}|^{-\frac{1}{s+1}}\right)^{s+1}$$

$$= (s+1)\log\left(\frac{M}{\delta}\right) < (s+1)\frac{1}{\gamma}\left(\frac{M}{\delta}\right)^\gamma,$$

where the first inequality follows from $\mathcal{H}_M \subset \bigcup_{(L,p)\in \mathrm{Ind}_{\bar{L},\bar{p}} : I_1(L,p)\leq M} \mathcal{H}(L,p,s,F)$, and the last inequality from $\gamma \log x^{\frac{1}{\gamma}} = \log x < x$ that holds for all $x, \gamma > 0$.

The second assertion can be confirmed by noting that for any $r \in \mathcal{H}_M$ with $M \geq 1$,

$$\|r - r^*\|_\infty \leq \|r\|_\infty + \|r^*\|_\infty \leq M + \|r^*\|_\infty$$

$$\leq \left(1 + \frac{\|r^*\|_\infty}{M}\right)M \leq (1 + \|r^*\|_\infty)M$$

holds.

$\square$

**Definition 4** (Derived function class and bracketing entropy). Given a real-valued function class $\mathcal{F}$, define $\ell \circ \mathcal{F} := \{\ell \circ f : f \in \mathcal{F}\}$. By extension, we define $I : \ell \circ \mathcal{H} \to [1,\infty)$ by $I(\ell \circ r) = I(r)$ and $\ell \circ \mathcal{H}_M := \{\ell \circ r : r \in \mathcal{H}_M\}$. Note that, as a result, $\ell \circ \mathcal{H}_M$ coincides with $\{\ell \circ r \in \ell \circ \mathcal{H} : I(\ell \circ r) \leq M\}$.

**Lemma 9.** *Let $\ell : (b_r, B_r) \to \mathbb{R}$ be a $\nu$-Lipschitz continuous function. Let $H_B\left(\delta, \mathcal{F}, \|\cdot\|_{L^2(P)}\right)$ denote the bracketing entropy of $\mathcal{F}$ with respect to a distribution $P$. Then, for any distribution $P$, any $\gamma > 0$, any $M \geq 1$, and any $\delta > 0$, we have*

$$H_B\left(\delta, \ell \circ \mathcal{H}_M, \|\cdot\|_{L^2(P)}\right) \leq \frac{(s+1)(2\nu)^\gamma}{\gamma}\left(\frac{M}{\delta}\right)^\gamma.$$

*Moreover, there exists $c_0 > 0$ such that for any $M \geq 1$ and any distribution $P$,*

$$\sup_{\ell \circ r \in \ell \circ \mathcal{H}_M} \|\ell \circ r - \ell \circ r^*\|_{L^2(P)} \leq c_0 \nu M,$$

$$\sup_{\substack{\ell \circ r \in \ell \circ \mathcal{H}_M \\ \|\ell \circ r - \ell \circ r^*\|_{L^2(P)}\leq \delta}} \|\ell \circ r - \ell \circ r^*\|_\infty \leq c_0 \nu M, \quad \text{for all } \delta > 0.$$

*Proof.* By combining Lemma 2.1 in van de Geer (2000) with Lemma 6, we have

$$H_B\left(\delta, \ell \circ \mathcal{H}_M, \|\cdot\|_{L^2(P)}\right) \leq \log \mathcal{N}\left(\frac{\delta}{2}, \ell \circ \mathcal{H}_M, \|\cdot\|_\infty\right),$$

$$\leq \log \mathcal{N}\left(\frac{\delta}{2\nu}, \mathcal{H}_M, \|\cdot\|_\infty\right) \leq \frac{s+1}{\gamma}\left(\frac{2\nu M}{\delta}\right)^\gamma.$$

For $M \geq 1$, we have

$$\sup_{\ell \circ r \in \ell \circ \mathcal{H}_M} \|\ell \circ r - \ell \circ r^*\|_{L^2(P)} \leq \sup_{\ell \circ r \in \ell \circ \mathcal{H}_M} \|\ell \circ r - \ell \circ r^*\|_\infty$$

$$\sup_{\substack{\ell \circ r \in \ell \circ \mathcal{H}_M \\ \|\ell \circ r - \ell \circ r^*\|_{L^2(P)}\leq \delta}} \|\ell \circ r - \ell \circ r^*\|_\infty \leq \sup_{\ell \circ r \in \ell \circ \mathcal{H}_M} \|\ell \circ r - \ell \circ r^*\|_\infty,$$

and Lemma 6 implies

$$\sup_{\ell \circ r \in \ell \circ \mathcal{H}_M} \|\ell \circ r - \ell \circ r^*\|_\infty \leq \sup_{r \in \mathcal{H}_M} \nu \|r - r^*\|_\infty \leq \nu c_0 M.$$

$\square$

### K.2 BOUNDING THE EMPIRICAL DEVIATIONS

**Lemma 10.** *Under the conditions of Theorem 2, for any $0 < \gamma < 2$, we have*

$$|(\mathbb{E}_{\text{de}} - \hat{\mathbb{E}}_{\text{de}})(\ell_1^{\hat{r}} - \ell_1^{r^*})| = \mathcal{O}_{\mathbb{P}}\left(\max\left\{\frac{\|\hat{r} - r^*\|_{L^2(p_{\text{de}})}^{1-\gamma/2}}{\sqrt{n_{\text{de}}}}, \frac{1}{n_{\text{de}}^{2/(2+\gamma)}}\right\}\right)$$

$$|(\mathbb{E}_{\text{nu}} - \hat{\mathbb{E}}_{\text{nu}})(\ell_1^{\hat{r}} - \ell_1^{r^*})| = \mathcal{O}_{\mathbb{P}}\left(\max\left\{\frac{\|\hat{r} - r^*\|_{L^2(p_{\text{de}})}^{1-\gamma/2}}{\sqrt{n_{\text{nu}}}}, \frac{1}{n_{\text{nu}}^{2/(2+\gamma)}}\right\}\right)$$

$$|(\mathbb{E}_{\text{nu}} - \hat{\mathbb{E}}_{\text{nu}})(\ell_2^{\hat{r}} - \ell_2^{r^*})| = \mathcal{O}_{\mathbb{P}}\left(\max\left\{\frac{\|\hat{r} - r^*\|_{L^2(p_{\text{de}})}^{1-\gamma/2}}{\sqrt{n_{\text{nu}}}}, \frac{1}{n_{\text{nu}}^{2/(2+\gamma)}}\right\}\right)$$

*as $n_{\text{nu}}, n_{\text{de}} \to \infty$.*

*Proof.* Since $0 < \gamma < 2$, we can apply Lemma 11 in combination with Lemma 9 to obtain

$$\sup_{r \in \mathcal{H}} \frac{|(\mathbb{E}_{\text{de}} - \hat{\mathbb{E}}_{\text{de}})(\ell_1^r - \ell_1^{r^*})|}{D_1(r)} = \mathcal{O}_{\mathbb{P}}(1),$$

$$\sup_{r \in \mathcal{H}} \frac{|(\mathbb{E}_{\text{nu}} - \hat{\mathbb{E}}_{\text{nu}})(\ell_1^r - \ell_1^{r^*})|}{D_2(r)} = \mathcal{O}_{\mathbb{P}}(1),$$

$$\sup_{r \in \mathcal{H}} \frac{|(\mathbb{E}_{\text{nu}} - \hat{\mathbb{E}}_{\text{nu}})(\ell_2^r - \ell_2^{r^*})|}{D_3(r)} = \mathcal{O}_{\mathbb{P}}(1),$$

where

$$D_1(r) = \max\left\{\frac{\|\ell_1^r - \ell_1^{r^*}\|_{L^2(p_{\text{de}})}^{1-\gamma/2} I(\ell_1^r)^{\gamma/2}}{\sqrt{n_{\text{de}}}}, \frac{I(\ell_1^r)}{n_{\text{de}}^{2/(2+\gamma)}}\right\},$$

$$D_2(r) = \max\left\{\frac{\|\ell_1^r - \ell_1^{r^*}\|_{L^2(p_{\text{nu}})}^{1-\gamma/2} I(\ell_1^r)^{\gamma/2}}{\sqrt{n_{\text{nu}}}}, \frac{I(\ell_1^r)}{n_{\text{nu}}^{2/(2+\gamma)}}\right\},$$

$$D_3(r) = \max\left\{\frac{\|\ell_2^r - \ell_2^{r^*}\|_{L^2(p_{\text{nu}})}^{1-\gamma/2} I(\ell_2^r)^{\gamma/2}}{\sqrt{n_{\text{nu}}}}, \frac{I(\ell_2^r)}{n_{\text{nu}}^{2/(2+\gamma)}}\right\},$$

Noting that $\sup_{r \in \mathcal{H}} I(r) < \infty$, that $\ell_2, \ell_1$ are Lipschitz continuous, and that $\|\hat{r} - r^*\|_{L^2(p_{\text{nu}})} \leq \left(\sup_{x \in \mathcal{X}} \left|\frac{p_{\text{nu}}(x)}{p_{\text{de}}(x)}\right|\right) \|\hat{r} - r^*\|_{L^2(p_{\text{de}})}$ holds, we have the assertion. $\qquad\square$

Following is a proposition originally presented in van de Geer (2000), which was rephrased in Kanamori et al. (2012) in a form that is convenient for our purpose.

**Lemma 11** (Lemma 5.14 in van de Geer (2000), Proposition 1 in Kanamori et al. (2012))**.** *Let $\mathcal{F} \subset L^2(P)$ be a function class and the map $I(f)$ be a complexity measure of $f \in \mathcal{F}$, where $I$ is a non-negative function on $\mathcal{F}$ and $I(f_0) < \infty$ for a fixed $f_0 \in \mathcal{F}$. We now define $\mathcal{F}_M = \{f \in \mathcal{F} : I(f) \leq M\}$ satisfying $\mathcal{F} = \bigcup_{M \geq 1} \mathcal{F}_M$. Suppose that there exist $c_0 > 0$ and $0 < \gamma < 2$ such that*

$$\sup_{f \in \mathcal{F}_M} \|f - f_0\| \leq c_0 M, \qquad \sup_{\substack{f \in \mathcal{F}_M \\ \|f - f_0\|_{L^2(P)} \leq \delta}} \|f - f_0\|_\infty \leq c_0 M, \quad \text{for all } \delta > 0,$$

*and that $H_B(\delta, \mathcal{F}_M, P) = \mathcal{O}(M/\delta)^\gamma$. Then, we have*

$$\sup_{f \in \mathcal{F}} \frac{\left|\int (f - f_0) d(P - P_n)\right|}{D(f)} = \mathcal{O}_{\mathbb{P}}(1), \ (n \to \infty),$$

*where $D(f)$ is defined by*

$$D(f) = \max\left\{ \frac{\|f - f_0\|_{L^2(P)}^{1-\gamma/2} I(f)^{\gamma/2}}{\sqrt{n}}, \frac{I(f)}{n^{2/(2+\gamma)}} \right\}.$$

### K.3 Bounding the difference of the BR divergence estimators

**Lemma 12.** *Assume $\mathcal{R}_{n_{\mathrm{de}}}^{p_{\mathrm{de}}}(\mathcal{H}) = o(1)(n_{\mathrm{de}} \to \infty)$ and $\mathcal{R}_{n_{\mathrm{nu}}}^{p_{\mathrm{nu}}}(\mathcal{H}) = o(1)(n_{\mathrm{nu}} \to \infty)$. Also assume the same conditions as Theorem 3. Then,*

$$\sup_{r\in\mathcal{H}} |\widehat{\mathrm{nnBR}}_f(r) - \widehat{\mathrm{BR}}_f(r)| = \mathcal{O}_{\mathbb{P}}\left( \exp\left( -\frac{2\alpha^2}{(B_\ell{}^2/n_{\mathrm{de}}) + (C^2 B_\ell{}^2/n_{\mathrm{nu}})} \right) \right)$$

*as $n_{\mathrm{nu}}, n_{\mathrm{de}} \to \infty$.*

*Proof.* First, by combining Lemma 13, the assumption on the Rademacher complexities, and Markov's inequality, there exist $\alpha > 0$ and $n_{\mathrm{de}}^0, n_{\mathrm{nu}}^0 \in \mathbb{N}$ such that for any $n_{\mathrm{de}} \geq n_{\mathrm{de}}^0$ and $n_{\mathrm{nu}} \geq n_{\mathrm{nu}}^0$ and any $\delta \in (0, 1)$, we have with probability at least $1 - \delta$,

$$\sup_{r\in\mathcal{H}} |\widehat{\mathrm{nnBR}}_f(r) - \widehat{\mathrm{BR}}_f(r)| \leq \frac{(1+C)B_\ell L_{\rho-\mathrm{Id}}}{\delta} \exp\left( -\frac{2\alpha^2}{(B_\ell{}^2/n_{\mathrm{de}}) + (C^2 B_\ell{}^2/n_{\mathrm{nu}})} \right).$$

Therefore, we have the assertion. $\qquad\square$

**Lemma 13.** *Assume $\mathcal{R}_{n_{\mathrm{de}}}^{p_{\mathrm{de}}}(\mathcal{H}) = o(1)(n_{\mathrm{de}} \to \infty)$ and $\mathcal{R}_{n_{\mathrm{nu}}}^{p_{\mathrm{nu}}}(\mathcal{H}) = o(1)(n_{\mathrm{nu}} \to \infty)$. Also assume the same conditions as Theorem 3. Then, there exist $\alpha > 0$ and $n_{\mathrm{de}}^0, n_{\mathrm{nu}}^0 \in \mathbb{N}$ such that for any $n_{\mathrm{de}} \geq n_{\mathrm{de}}^0$ and $n_{\mathrm{nu}} \geq n_{\mathrm{nu}}^0$,*

$$\mathbb{E}\left[ \sup_{r\in\mathcal{H}} |\widehat{\mathrm{nnBR}}_f(r) - \widehat{\mathrm{BR}}_f(r)| \right] \leq (1+C)B_\ell L_{\rho-\mathrm{Id}} \exp\left( -\frac{2\alpha^2}{(B_\ell{}^2/n_{\mathrm{de}}) + (C^2 B_\ell{}^2/n_{\mathrm{nu}})} \right)$$

*holds.*

*Proof.* First, we have

$$\mathbb{E}\left[ \sup_{r\in\mathcal{H}} |\widehat{\mathrm{nnBR}}_f(r) - \widehat{\mathrm{BR}}_f(r)| \right]$$

$$= \mathbb{E}\left[ \sup_{r\in\mathcal{H}} \left| \rho(\hat{\mathbb{E}}_{\mathrm{mod}}\ell_1(r(X))) - \hat{\mathbb{E}}_{\mathrm{mod}}\ell_1(r(X)) \right| \right]$$

$$= \mathbb{E}\left[ \sup_{r\in\mathcal{H}} \mathbb{1}\{\rho(\hat{\mathbb{E}}_{\mathrm{mod}}\ell_1(r(X))) \neq \hat{\mathbb{E}}_{\mathrm{mod}}\ell_1(r(X))\} \cdot |\rho(\hat{\mathbb{E}}_{\mathrm{mod}}\ell_1(r(X))) - \hat{\mathbb{E}}_{\mathrm{mod}}\ell_1(r(X))| \right]$$

$$\leq \mathbb{E}\left[ \sup_{r\in\mathcal{H}} \mathbb{1}\{\rho(\hat{\mathbb{E}}_{\mathrm{mod}}\ell_1(r(X))) \neq \hat{\mathbb{E}}_{\mathrm{mod}}\ell_1(r(X))\} \right] \left( \sup_{s:|s|\leq(1+C)B_\ell} |\rho(s) - s| \right),$$

where $\mathbb{1}\{\cdot\}$ denotes the indicator function, and we used $|\hat{\mathbb{E}}_{\mathrm{mod}}\ell_1(r(X))| \leq (1+C)B_\ell$. Further, we have

$$\sup_{s:|s|\leq(1+C)B_\ell} |\rho(s) - s| \leq \sup_{s:|s|\leq(1+C)B_\ell} |(\rho - \mathrm{Id})(s) - (\rho - \mathrm{Id})(0)| + |(\rho - \mathrm{Id})(0)|$$

$$\leq \sup_{s:|s|\leq(1+C)B_\ell} L_{\rho-\mathrm{Id}}|s - 0| + 0 \leq (1+C)B_\ell L_{\rho-\mathrm{Id}},$$

where $\mathrm{Id}$ denotes the identity function. On the other hand, since $\inf_{r\in\mathcal{H}} \mathbb{E}\hat{\mathbb{E}}_{\mathrm{mod}}\ell_1(r(X)) > 0$ is assumed, there exists $\beta > 0$ such that for any $r \in \mathcal{H}$, $\mathbb{E}\hat{\mathbb{E}}_{\mathrm{mod}}\ell_1(r(X)) > \beta$. Therefore, denoting

the support of a function by $\mathrm{supp}(\cdot)$,

$$\mathbb{E}\left[\sup_{r\in\mathcal{H}}\mathbb{1}\{\rho(\hat{\mathbb{E}}_{\mathrm{mod}}\ell_1(r(X)))\neq\hat{\mathbb{E}}_{\mathrm{mod}}\ell_1(r(X))\}\right]$$

$$=\mathbb{E}\left[\sup_{r\in\mathcal{H}}\mathbb{1}\{\hat{\mathbb{E}}_{\mathrm{mod}}\ell_1(r(X))\in\mathrm{supp}(\rho-\mathrm{Id})\}\right]$$

$$=\mathbb{E}\left[\sup_{r\in\mathcal{H}}\mathbb{1}\{\hat{\mathbb{E}}_{\mathrm{mod}}\ell_1(r(X))<0\}\right]$$

$$=\mathbb{E}\left[\mathbb{1}\{\exists r\in\mathcal{H}:\hat{\mathbb{E}}_{\mathrm{mod}}\ell_1(r(X))<0\}\right]$$

$$=\mathbb{P}\left(\exists r\in\mathcal{H}:\hat{\mathbb{E}}_{\mathrm{mod}}\ell_1(r(X))<0\right)$$

$$\leq\mathbb{P}\left(\exists r\in\mathcal{H}:\hat{\mathbb{E}}_{\mathrm{mod}}\ell_1(r(X))<\mathbb{E}\hat{\mathbb{E}}_{\mathrm{mod}}\ell_1(r(X))-\beta\right)$$

$$\leq\mathbb{P}\left(\beta<\sup_{r\in\mathcal{H}}(\mathbb{E}\hat{\mathbb{E}}_{\mathrm{mod}}\ell_1(r(X))-\hat{\mathbb{E}}_{\mathrm{mod}}\ell_1(r(X)))\right).$$

Take an arbitrary $\alpha\in(0,\beta)$. Since $\mathcal{R}_{n_{\mathrm{de}}}^{p_{\mathrm{de}}}(\mathcal{H})\to 0(n_{\mathrm{de}}\to\infty)$ and $\mathcal{R}_{n_{\mathrm{nu}}}^{p_{\mathrm{nu}}}(\mathcal{H})\to 0(n_{\mathrm{nu}}\to\infty)$, we can apply Lemma 14 and obtain the assertion. $\qquad\square$

**Lemma 14.** *Let $\beta>\alpha>0$. Assume that there exist $n_{\mathrm{de}}^0,n_{\mathrm{nu}}^0\in\mathbb{N}$ such that for any $n_{\mathrm{de}}\geq n_{\mathrm{de}}^0$ and $n_{\mathrm{nu}}\geq n_{\mathrm{nu}}^0$,*

$$4L_{\ell_1}\mathcal{R}_{n_{\mathrm{de}}}^{p_{\mathrm{de}}}(\mathcal{H})+4CL_{\ell_1}\mathcal{R}_{n_{\mathrm{nu}}}^{p_{\mathrm{nu}}}(\mathcal{H})<\beta-\alpha.$$

*Then, for any $n_{\mathrm{de}}\geq n_{\mathrm{de}}^0$ and $n_{\mathrm{nu}}\geq n_{\mathrm{nu}}^0$, we have*

$$\mathbb{P}\left(\beta<\sup_{r\in\mathcal{H}}(\mathbb{E}\hat{\mathbb{E}}_{\mathrm{mod}}\ell_1(r(X))-\hat{\mathbb{E}}_{\mathrm{mod}}\ell_1(r(X)))\right)$$

$$\leq\exp\left(-\frac{2\alpha^2}{(B_\ell^2/n_{\mathrm{de}})+(C^2B_\ell^2/n_{\mathrm{nu}})}\right).$$

*Proof.* First, we will apply McDiarmid's inequality. The absolute difference caused by altering one data point in $\sup_{r\in\mathcal{H}}(\mathbb{E}\hat{\mathbb{E}}_{\mathrm{mod}}\ell_1(r(X))-\hat{\mathbb{E}}_{\mathrm{mod}}\ell_1(r(X)))$ is bounded by $\frac{B_\ell}{n_{\mathrm{de}}}$ if the change is in a sample from $p_{\mathrm{de}}$ and $\frac{CB_\ell}{n_{\mathrm{nu}}}$ otherwise. This can be confirmed by letting $\hat{\mathbb{E}}'_{\mathrm{mod}}$ denote the sample averaging operator obtained by altering one data point in $\hat{\mathbb{E}}_{\mathrm{mod}}$ and observing

$$\sup_{r\in\mathcal{H}}\{\mathbb{E}\hat{\mathbb{E}}_{\mathrm{mod}}\ell_1(r(X))-\hat{\mathbb{E}}_{\mathrm{mod}}\ell_1(r(X))\}-\sup_{r\in\mathcal{H}}\{\mathbb{E}\hat{\mathbb{E}}_{\mathrm{mod}}\ell_1(r(X))-\hat{\mathbb{E}}'_{\mathrm{mod}}\ell_1(r(X))\}$$

$$\leq\sup_{r\in\mathcal{H}}\{\mathbb{E}\hat{\mathbb{E}}_{\mathrm{mod}}\ell_1(r(X))-\hat{\mathbb{E}}_{\mathrm{mod}}\ell_1(r(X))-(\mathbb{E}\hat{\mathbb{E}}_{\mathrm{mod}}\ell_1(r(X))-\hat{\mathbb{E}}'_{\mathrm{mod}}\ell_1(r(X)))\}$$

$$\leq\sup_{r\in\mathcal{H}}\{\hat{\mathbb{E}}'_{\mathrm{mod}}\ell_1(r(X))-\hat{\mathbb{E}}_{\mathrm{mod}}\ell_1(r(X))\}.$$

The right-most expression can be bounded by $\frac{B_\ell}{n_{\mathrm{de}}}$ if the change is in a sample from $p_{\mathrm{de}}$ and $\frac{CB_\ell}{n_{\mathrm{nu}}}$ otherwise. Likewise, $\sup_{r\in\mathcal{H}}(\mathbb{E}\hat{\mathbb{E}}_{\mathrm{mod}}\ell_1(r(X))-\hat{\mathbb{E}}'_{\mathrm{mod}}\ell_1(r(X)))-\sup_{r\in\mathcal{H}}(\mathbb{E}\hat{\mathbb{E}}_{\mathrm{mod}}\ell_1(r(X))-\hat{\mathbb{E}}_{\mathrm{mod}}\ell_1(r(X)))$ can be bounded by one of these quantities. Therefore, we have

$$\left|\sup_{r\in\mathcal{H}}\{\mathbb{E}\hat{\mathbb{E}}_{\mathrm{mod}}\ell_1(r(X))-\hat{\mathbb{E}}_{\mathrm{mod}}\ell_1(r(X))\}-\sup_{r\in\mathcal{H}}\{\mathbb{E}\hat{\mathbb{E}}_{\mathrm{mod}}\ell_1(r(X))-\hat{\mathbb{E}}'_{\mathrm{mod}}\ell_1(r(X))\}\right|$$

$$\leq\frac{B_\ell}{n_{\mathrm{de}}}+\frac{CB_\ell}{n_{\mathrm{nu}}},$$

and McDiarmid's inequality implies, for any $\epsilon>0$,

$$\mathbb{P}\left(\epsilon<\sup_{r\in\mathcal{H}}(\mathbb{E}\hat{\mathbb{E}}_{\mathrm{mod}}\ell_1(r(X))-\hat{\mathbb{E}}_{\mathrm{mod}}\ell_1(r(X)))-\mathbb{E}\left[\sup_{r\in\mathcal{H}}(\mathbb{E}\hat{\mathbb{E}}_{\mathrm{mod}}\ell_1(r(X))-\hat{\mathbb{E}}_{\mathrm{mod}}\ell_1(r(X)))\right]\right)$$

$$\leq\exp\left(-\frac{2\epsilon^2}{(B_\ell^2/n_{\mathrm{de}})+(C^2B_\ell^2/n_{\mathrm{nu}})}\right).$$

(10)

Now, applying Lemma 1, we have

$$\mathbb{E}\left[\sup_{r\in\mathcal{H}}(\mathbb{E}\hat{\mathbb{E}}_{\mathrm{mod}}\ell_1(r(X)) - \hat{\mathbb{E}}_{\mathrm{mod}}\ell_1(r(X)))\right]$$

$$\leq \mathbb{E}\left[\sup_{r\in\mathcal{H}}|\mathbb{E}_{\mathrm{de}}\ell_1(r(X)) - \hat{\mathbb{E}}_{\mathrm{de}}\ell_1(r(X))|\right] + C\mathbb{E}\left[\sup_{r\in\mathcal{H}}|\mathbb{E}_{\mathrm{nu}}\ell_1(r(X)) - \hat{\mathbb{E}}_{\mathrm{nu}}\ell_1(r(X))|\right]$$

$$\leq 4L_{\ell_1}\mathcal{R}_{n_{\mathrm{de}}}^{p_{\mathrm{de}}}(\mathcal{H}) + 4CL_{\ell_1}\mathcal{R}_{n_{\mathrm{nu}}}^{p_{\mathrm{nu}}}(\mathcal{H}) =: \mathcal{R}.$$

By the assumption, if $n_{\mathrm{de}} \geq n_{\mathrm{de}}^0$ and $n_{\mathrm{nu}} \geq n_{\mathrm{nu}}^0$, we have $\mathcal{R} < \beta - \alpha$. Therefore,

$$\mathbb{E}\left[\sup_{r\in\mathcal{H}}(\mathbb{E}\hat{\mathbb{E}}_{\mathrm{mod}}\ell_1(r(X)) - \hat{\mathbb{E}}_{\mathrm{mod}}\ell_1(r(X)))\right] < \beta - \alpha < \beta,$$

hence $\beta - \mathbb{E}\left[\sup_{r\in\mathcal{H}}(\mathbb{E}\hat{\mathbb{E}}_{\mathrm{mod}}\ell_1(r(X)) - \hat{\mathbb{E}}_{\mathrm{mod}}\ell_1(r(X)))\right] > 0$. Therefore, we can take $\epsilon = \beta - \mathbb{E}\left[\sup_{r\in\mathcal{H}}(\mathbb{E}\hat{\mathbb{E}}_{\mathrm{mod}}\ell_1(r(X)) - \hat{\mathbb{E}}_{\mathrm{mod}}\ell_1(r(X)))\right]$ in Equation (10) to obtain

$$\mathbb{P}\left(\beta < \sup_{r\in\mathcal{H}}(\mathbb{E}\hat{\mathbb{E}}_{\mathrm{mod}}\ell_1(r(X)) - \hat{\mathbb{E}}_{\mathrm{mod}}\ell_1(r(X)))\right)$$

$$\leq \exp\left(-\frac{2(\beta - \mathbb{E}\left[\sup_{r\in\mathcal{H}}(\mathbb{E}\hat{\mathbb{E}}_{\mathrm{mod}}\ell_1(r(X)) - \hat{\mathbb{E}}_{\mathrm{mod}}\ell_1(r(X)))\right])^2}{(B_\ell^2/n_{\mathrm{de}}) + (C^2 B_\ell^2/n_{\mathrm{nu}})}\right)$$

$$\leq \exp\left(-\frac{2(\beta - \mathcal{R})^2}{(B_\ell^2/n_{\mathrm{de}}) + (C^2 B_\ell^2/n_{\mathrm{nu}})}\right) \leq \exp\left(-\frac{2\alpha^2}{(B_\ell^2/n_{\mathrm{de}}) + (C^2 B_\ell^2/n_{\mathrm{nu}})}\right),$$

where we used $0 < \alpha < \beta - \mathcal{R}$. $\qquad\qquad\square$

