# OpenReview forum: "Non-Negative Bregman Divergence Minimization for Deep Direct Density Ratio Estimation"
_ICLR.cc/2021/Conference — Reject_

### Official Review · AnonReviewer4 · 2020-10-18
**Interesting analysis of density ratio estimation**

**Rating:** 6
**Confidence:** 4

**Review:**

##########################################################################

Summary:

The paper studies density ratio estimation (DRE), addressing the 'train-loss hacking' problems which often arise and hamper estimation when models are too flexible. The authors propose a new risk estimator for DRE, providing a non-negative Bregman divergence estimator, with the non-negative correction. Theoretical analyses are shown with the estimation error and empirical analyses are examined in multiple machine learning problem settings.

##########################################################################

Pros:

- A simple yet practical and principled algorithm for DRE is proposed.
- Although the non-negative correction idea itself is not new, as the authors mentioned in Section 1, using the prior knowledge for the non-negative correction appears novel in DRE.
- All numerical experiments considered in the paper are reasonable and interesting.

##########################################################################

Cons (or questions):

- Although the results in section 4 justify the proposed estimator has vanishing estimation error in some sense, the core idea for the results are basically a simple application of the Mcdiarmid’s inequality. I acknowledge these are highly technical results, but I believe more theoretical analyses of the approximation error would improve the paper. What conditions are needed for r^* (or model space) to have zero approximation error? How can this be explained in terms of a usual reproducing kernel Hilbert space?

- Based on the theories in Section 4, the current algorithm provides an estimator that has a vanishing estimation error. As far as I understand, it might not guarantee the maximization of AUROC (or minimization of pairwise disagreement). Are there any explicit relationships between the 'consistent' density ratio estimation and the AUROC maximization?

- Following the previous question, Figure 3 in the appendix shows the proposed algorithm does not diverge to the negative infinity, but it is not clear if the Bregman divergence evaluated at the proposed estimator is converging to the 'optimal' Bregman divergence value. How can we empirically justify Corollary 1 (even in a very simple simulation setting?)

##########################################################################

Overall, I recommend the weak acceptance. I may well have missed some points in my reading, so clarification is welcome.

---

> ### Author Response · Authors · 2020-11-22
> **Response to AnonReviewer4**
>
> Thank you for your constructive comments.
> Following your comments, we updated our manuscript. The main updates are listed as follows:
> - We showed new theoretical results on the convergence rate of the density ratio under a restriction of the neural network models (**Theorem 2**). The shown convergence rate almost achieves the same convergence rate as that of the nonparametric kernel regression using reproducing kernel Hilbert space (RKHS).
> - We added numerical experiments to support the theoretical results. Using the true value of the density ratio, we calculate the mean squared error between the true value and the estimator of the density ratio.
>
> Our replies to your questions are listed below.
>
> ---
>
> Q1. Although the results in section 4 justify the proposed estimator has vanishing estimation error in some sense, the core idea for the results are basically a simple application of the Mcdiarmid's inequality. I acknowledge these are highly technical results, but I believe more theoretical analyses of the approximation error would improve the paper. What conditions are needed for r^* (or model space) to have zero approximation error? How can this be explained in terms of a usual RKHS?
>
> A1. **In the revised manuscript, we added Theorem 2, which shows the convergence rate of nonparametric density ratio estimation (DRE) under a restricted neural network model (multi-layer perceptron with ReLU activation function (Definition 3))**. This convergence rate is the standard one in nonparametric regression; that is, under an appropriate condition, the convergence rate approaches to $O_p(\min(n_{nu},n_{de})^{-1/2})$. Kanamori et al. (2012) studied the nonparametric DRE with RKHS and showed a similar convergence rate. We summarize the technical points in the proof as follows:
> - We bounded the convergence rate using the **covering entropy bound** shown by Schmidt-Hieber (2019), which studied nonparametric regression using neural networks with ReLU activation function.
> - **The main difference between nonparametric regressions using neural networks and RKHS is based on the covering entropy bound; that is, when we restrict the neural network models, we can show the convergence rate as tight as the RKHS**. However, if we do not restrict the neural network models, we cannot derive such a fast bound.
> - When transforming the generalization error bound of Theorem 1 to $L^2$ convergence rate, the convergence results becomes slower than the RKHS (Kanamori et al. (2012)) because we did not put some restrictions on the neural network models.
> - We derived two generalization error bounds (convergence rates) in Theorem 1 and Theorem 2. **The result of Theorem 1 corresponds to the standard classification results** ([Bartlett et al. (2003)](https://www.jmlr.org/papers/volume3/bartlett02a/bartlett02a.pdf) and Kiryo et al. (2017)). We can derive the error bound of the classification problem from the BR divergence error bound (See PU learning part of Appendix A). This result holds for various neural network models. **On the other hand, in Theorem 2, we put more restrictions on the model and showed faster convergence results**. This result is desirable in nonparametric regression, which wants a nearly $O_p(\min(n_{nu},n_{de})^{-1/2})$ convergence rate (See Kanamori et al. (2012)). We can also relate Theorem 2 to a generalization error bound of the classification problem under a restricted neural network model (Definition 3).
> - We did not discuss more formal optimality, such as the minimax optimality. We consider such a discussion is beyond the scope of this study.
>
> ---
>
> Q2. Based on the theories in Section 4, the current algorithm provides an estimator that has a vanishing estimation error. As far as I understand, it might not guarantee the maximization of AUROC (or minimization of pairwise disagreement). Are there any explicit relationships between the 'consistent' density ratio estimation and the AUROC maximization?
>
> A2. In the revised manuscript, we calculated the MSE of the density ratio in Appendix E using artificially generated datasets with the true density ratio. The setting follows Kanamori et al. (2009).
>
> ---
>
> Q3. Following the previous question, Figure 3 in the appendix shows the proposed algorithm does not diverge to the negative infinity, but it is not clear if the Bregman divergence evaluated at the proposed estimator is converging to the 'optimal' Bregman divergence (BR) value. How can we empirically justify Corollary 1 (even in a very simple simulation setting?)
>
> A3. Following Q2 and Q3, we added numerical experiments investigating the MSE between the true and estimated density ratio in Appendix E. We consider that this experiment also simultaneously investigated the empirical approximation of the BR divergence because the BR divergence's generalization error bound boils down to the estimation error of the density ratio.

---

### Official Review · AnonReviewer3 · 2020-10-29
**The paper suggests a way to estimate the density ratio of two distributions. While the contribution seems potentially useful, it is very hard to follow.**

**Rating:** 6
**Confidence:** 3

**Review:**

The paper addresses an issue that arises in a particular formulation of the density ratio estimation problem. Namely, when one tries to directly fit a density estimation ratio, while minimizing Bergman divergence, it may be that overfitting causes the minimization problem to diverge to minus-infinity.
The paper suggests some form of regularization that uses a bound on the ell_infy norm of the ratio function. The paper justifies the change by proving a bound on Bergman divergence between minimizer computed and the best minimizer within the class.
The authors support their suggestion by experimenting: computing anomalies and learning from positive or unlabeled data.

The main drawback of the paper is that it is very hard to read, overloaded with cumbersome notation and lacks sensible discussion about the meaning of the results. Frankly, I couldn't decipher exactly why the over-fitting problem arises, what the 'theoretical justification' exactly says. While the experiments seems good, I didn't fully understand what was done, why are these the right experiments and what was learned from them.
More serious is that I can't understand how it compares to prior work. The paper claims it extends a work by Kiryo et al. which I'm not familiar with. There are other lines of research that attack the general problem of density estimation, some of them very related.
In particular, one can use arbitrarily strong learning models via the approach of the following paper:
"Estimating Divergence Functionals and the Likelihood Ratio by Convex Risk Minimization" by Ngoyan, Wainright, Jordan.
A comparison with this paper (and the rich literature that cites it) is lacking.
Another line of research for direct estimation comes from the econometric community. A starting point is
"Covariate balancing propensity score" by Imai and Ratkovic.
The latter is less directly related but represents a huge body of work on direct density ratio estimation that should be cited and discussed.

To conclude, the paper's suggestion might be a good one, it is just too hard to asses. It seems to insist on a particular formulation of the problem and then find ways to make it work, but the writing is not clear and important related work is missing.

Specific comments, mostly related to style and presentation:
1. In the introduction you have to discuss better what is the problem you solve exactly. How is direct density estimation different from calibrated classification? What is train-loss hacking and why is it unique to this formulation of the problem? etc.
2. Don't assume that readers are familiar with Kiryo et al.
3. Equation (3) needs to be discussed. That's the heart of the optimization to solve. I think it is somewhat similar to the one in NWJ I mention above, but I'm not sure. In any case, it is a bit 'weird' that r^* could be dropped from the formulation. What does that mean?
4. Figure 1 suggests the issue is over fitting, I didn't understand the claim it is of a different flavor.
5. I thing the assumption that the ratio is bounded is a reasonable one. One can try to justify it better.
6. The last paragraph of Section 3 has to be explained better.
7. I found Section 4 frustrating to read. The lack of numbered equations, the wording and the notation just confused me too much. At the end of the day I didn't fully understand what is the meaning of the theorem. For instance, Lemma 2 assumes the true ratio function is part of the class H. That's a very strong assumption no? How do these bounds compare to previous bounds?
8. In the experiments, besides applications and such, don't you want simply to check the quality of the estimation of the ratio when the ground truth is known? I don't understand how the first experiment in Section 5 achieves that (I think I do, but I'm not sure).

---

> ### Author Response · Authors · 2020-11-15
> **Response to AnonReviewer3**
>
> Thank you for your insightful comments.
>
> Following your comments, we have updated the manuscript with more explanations on the research motivation, train-loss hacking, and the theoretical results.
>
> First, we would like to clarify the following five points.
>
> **(1)** We are interested in density ratio estimation (DRE) using **deep neural networks**, and **not** in the basic formulation of the Bregman (BR) divergence minimization itself, which is formulated by Sugiyama et al. (2011).
>
> **(2)** The BR divergence minimization (Sugiyama et al. (2011)) is a generalization of existing DRE methods, including Nguyen et al. (2010), and widely accepted in the DRE literature.
>
> **(3)** Owing to our typo, we cited Nguyen et al. (2011) instead of Nguyen et al. (2010). Thank you for pointing this out. We have already investigated Nguyen et al. (2010)'s method, referred to as "**UKL**". UKL shares the same objective with KLIEP of Sugiyama et al. (2008). In addition, Sugiyama et al. (2011) proved that the UKL and KLIEP could be generalized as Bregman divergence minimization. Our goal is to conduct the BR divergence minimization procedures, including the method of Nguyen et al. (2010), using neural networks. The implementation of our D3RE with Nguyen et al. (2010) (UKL) is "**nnBR-UKL**" in Sections 3.2 and 5.3.
>
> **(4)** We find that Imai and Ratkovic (2014) is less relevant to our work and DRE for the following reasons:
>
> - From the density ratio estimation (DRE) perspective, Imai and Ratkovic (2014) have little novelty because the method is the same as Gretton et al. (2009), which we cited.
>
> - The motivation of Imai and Ratkovic (2014) is different from DRE. The goal is to perform the generalized method of moments (GMM), including the density ratio as a **nuisance** parameter; that is, they are not interested in estimating the density ratio itself.
>
> - Imai and Ratkovic (2014) is just one instance of the various GMM formulations with the density ratios. We do not have the motivation to focus on it.
>
> - Owing to the non-Donsker property, it is not appropriate to use Iman and Ratkovic (2014) even with naive ridge and lasso regression, not only with neural networks; instead, we need to use the GMM with sample splitting as Chernozhukov et al., (2016), and Kato et al. (2020), which we cited.
>
> In the revised manuscript, we cited the work in Appendix H and explained why it is not important.
>
> **(5)** We do not require the knowledge of Kiryo et al. (2017) and the train-loss hacking problem is explained in Section 3.1. In the revised manuscript, we added more explanations on this problem.
>
> The main updates of the manuscript are as follows:
>
> - By moving some sentences into the Appendix, we have added more explanation of the research motivation in the abstract and introduction and the theoretical results in Section 4.1.
>
> - We fixed the typo of Nguyen et al. (2010) and clarified the relationship between Nguyen et al. (2010) (UKL) and BR divergence.
>
> Our replies to the other questions are listed below.
>
> ---
>
> **Q1**. Equation (3) needs to be discussed. That's the heart of the optimization to solve. I think it is similar to the one in NWJ.
>
> **A1**. The formulation is **not** our proposal, and it is cited from Sugiyama et al. (2011). We added an explanation in the revised manuscript to clarify this point. The BR divergence minimization of Sugiyama et al. (2011) is a generalization of NWJ; therefore, the formulation also includes the technique used in NWJ.
>
> ---
>
> **Q2**. Figure 1 suggests the issue is overfitting, I didn't understand the claim.
>
> **A2**. We also consider that nuance is different from over-fitting and thus call it train-loss hacking. This phenomenon is also reported by Nan et al. (2020). In addition, we could observe that PU-NN diverged as one term of the objective went to negative infinity (Figure 2). Figure 1 is only an illustration, and we explained the problem in more detail in the Introduction of the revised manuscript and Section 3.1.
>
> ---
>
> **Q3**. Lemma 2 assumes the true ratio function is part of the class H. That's a very strong assumption?
>
> **A3**. This is only a technical assumption for simplicity, and it can be easily relaxed. When the hypothesis class does not include the true function, the deviation directly appears in the bound. See "Foundations of Machine Learning" (Mohri et al., 2012). This is also a common assumption adopted in machine learning for simplicity.
>
> ---
>
> **Q4**. don't you want simply to check the quality of the estimation of the ratio when the ground truth is known?
>
> **A4**. The focus of our experiments is on realistic high-dimensional data such as images, and it is a difficult problem setup to investigate through simulations, unlike existing studies. As an alternative to evaluating the estimation error of the ground-truth density ratio in a simulation setting, we experimented with the anomaly detection problem, in which better AUC is an indirect indication of a better estimation of the density ratio.

---

### Official Review · AnonReviewer1 · 2020-10-31
**Both the problem and contribution are interesting, but the writing and explanation need improvement.**

**Rating:** 5
**Confidence:** 3

**Review:**

***

Summary:

The paper addresses learning the ratio between two densities from their samples, with applications to outlier detection and covariate shift adaption. An existing approach is to minimize the Bregman (BR) divergence's empirical approximation while modeling the density ratio function $r^*$ by a flexible hypothesis family, such as neural networks (NNs). A particular issue of such an approach (that the present work aims to resolve) is "train-loss hacking," meaning that the empirical loss can become arbitrarily large and negative. A new loss/objective based on BR divergence has been proposed, appearing on page 4, and is referred to as $\widehat{\text{nnBR}}_f(r)$. The major theoretical result, Theorem 1, states that minimizing the proposed objective effectively minimizes the BR divergence for sufficiently large sample sizes. Following this theorem and its corollary, the paper presents empirical evaluations, showing the new algorithm outperforms prior ones on standard datasets.

***

Reasons for score:

I think the paper is marginally below the acceptance threshold of ICLR. Learning the density ratio is a well-motivated problem with an objective different from density or functional estimation. But it is unclear to me how strong the theoretical claims are, and the writing needs significantly more work. I am also concerned about the reasoning behind several claims and the potential gap between the derived theory and experiments. Please refer to the "Cons" for details.

***

Pros:


1. The paper attempts to resolve a particular issue for employing flexible hypothesis families in density ratio estimation with BR-divergence loss. I think the problem is interesting and practically relevant. In particular, as the BR divergence encompasses several well-known loss functions, the proposed method naturally induces a class of new methods (page 4).

2. The contribution is also meaningful as the algorithm incorporates the hypothesis class of neural networks, known to have strong expressive power, while some prior works don't. Under a sequence of Assumptions (1 to 4), Corollary 1 presents finite-sample guarantees for a specific NN class with bounded complexity. Experimental results further confirm the algorithm's effectiveness on tasks involving labeled images and text contents.

3. I also like how the paper imposes assumptions and identifies/handles the portion of the vanilla BR-type risk contributing to "train-loss hacking," as the rationale seems natural and might apply to other relevant tasks.

***

Cons:

1. It is unclear to me how strong those theoretical claims are.

- First, the main theorem, Theorem 1, holds under a sequence of assumptions. I can see that the first part of Assumption 1 is natural, but assuming an upper bound $\bar R$ on the density ratio might be a problem when combined with other assumptions/settings. Before continuing, I would like to point out a hidden assumption, $\ 0 < C < \frac1R\ $, right before Assumption 2, where $C$ appears in $\partial f(t) = \boldsymbol{C}(\partial f(t) t - f(t)) + \tilde f(t)$, as a key parameter in the definition of $\tilde f$. Note that this assumption translates to $0<R<1/C$. Additionally, observe that $1/5\le C$ in all the experiments and a very small $C$ can "damage the empirical performance" (Remark 1). Given all these constraints, one essentially requires the density ratio to be bounded by a small absolute constant, which seems restrictive.

- Second, I wonder if there is any quantitative assessment of the tightness of Theorem 1 or Corollary 1 (page 5). Let me use the latter as an example. In terms of sample size dependency, the discrepancy upper bound is $\mathcal O(\max \\{ 1/\sqrt n_{\text{de}}, 1/\sqrt{n_{\text{nu}}} \\} )$. For the BR divergence, is such dependency optimal (up to absolute constant factors)? As for $\delta$, the $\sqrt{\log \frac1\delta}$ factor seems tight, but how about the other parameters? In particular, does any constant in the theorem, e.g., $\kappa_1$ or $\kappa_2$, has non-polynomial dependency on $C, B_p, L$, or $B_{W_j}$? I think additional details and explanations should be provided to illustrate the meaning and significance of these results.

- Third, "Computation" would be an important aspect of the proposed heuristic. I don't see how to derive an efficient optimization procedure for finding the actual empirical risk minimizer or an accurate approximation. From a theoretical point of view, such a procedure is crucial for the sample-dependent error bounds in Theorem/Corollary 1 to take effect. For the experimental results' completeness, it also seems reasonable to add relevant details about implementing the proposed algorithm.

2. I also believe the writing of the paper needs improvement.

- As stated above, the assumptions, theorems, and definitions require more explanation. In particular, I suggest adding comments addressing how natural or restrictive the assumptions are, the significance of the theorem/corollary, and their optimality, and the intuitions behind the technical reasoning. For example, page 4 presents the new objective function's construction right after Assumption 2, which currently is a sequence of formulas/definitions together with light comments. In my opinion, this part is crucial for the paper and should show technical insights and novelty of the proposed method, along with relevant intuitions. Besides, the instantiation of the "nnBR divergence" with existing methods shown at the base of the page doesn't seem important to me and somehow introduces redundancy, given Table 1.

- On page 6, the paper claims that "the above theoretical guarantee already suffices for the basic justification of D3RE." I don't fully understand where this sufficiency comes from. Furthermore, I wonder if any prior or related method(s) satisfy a bound similar to that in Theorem 1, namely, if the sample sizes are large, the minimizer is essentially optimal. A concern relevant to this point is that the paper lacks references in a few places. For example, the second last paragraph on page 2 says, "Therefore, various methods for directly estimating the density ratio model have been proposed (Sugiyama et al., 2012), and (Sugiyama et al. 2011b) ...". My thought is that if this is a popular problem, then it should've studied by at least a few research groups instead of one.

3. Other comments, suggestions, and questions.

- On page 2, the original pointwise BR divergence is defined as $BR'_f(t^*\vert\\!\vert t)$. It might be better to avoid using the $(\cdot)'$ notation as it usually serves as an indicator for derivatives.

- In Equation (3) and some other definitions, I found $\boldsymbol x_i$ is used instead of $\boldsymbol x$, while the left-hand side quantities do not involve $i$.

- On page 5, it is said that "for the boundedness and Lipschitz continuity in Assumption 3 to hold ..., a technical assumption $b_r>0$ is sufficient." I wonder how the relevant quantities in Assumption 3 appear as functions of $b_r$.

- On page 4, $\ell_2(t)$ is introduced to represent $-\tilde f(t)$. What is the purpose of this new notation?

It would be nice if the author(s) can address some of my concerns in the rebuttal. Thanks.

***

---

> ### Author Response · Authors · 2020-11-22
> **Response to AnonReviewer1**
>
> Thank you for your constructive comments.
> We have updated the manuscript.
> Our replies are listed as follows.
>
> ---
>
> **Q1** It is unclear to me how strong those theoretical claims are.
>
> **Q1a** the density ratio to be bounded by a small absolute constant...
>
> **A1a** **We do not necessarily require a small absolute constant $R$**. What we require is a parameter $C$, which is not so far away from $1/R$. Although we remarked that very small $C$ can "damage the empirical performance," it does not mean that **$1/R$ is large (R is small)**, but means **$C$ should not be $C\ll 1/R$**. Therefore, we consider that **if $1/R$ is small, $C$ also can be small**. That is, **our requirement is not small $R$ but well guessed $C$**. Note that we do not require the exact guess of $R$. As the experiment shows, roughly guessed $C$ is enough. The message of our remark is **"for $C$, we do not require the exact $1/R$, but we recommend using large $C$ rather than small $C$."** In the next revision, we will add additional experiments supporting this claim.
>
> ---
>
> **Q1b** For the BR divergence, is such dependency optimal?
>
> **A1b** The optimality depends on the application. When considering a classification problem, the bound is a standard one (See Corollary 15 of [Bartlett et al. (2003)](https://www.jmlr.org/papers/volume3/bartlett02a/bartlett02a.pdf)). For the relationship between the BR divergence and classification problem, see Appendix A in the revised manuscript, where we explain PU classification risk and the BR divergence. On the other hand, from a regression perspective, deriving a tighter bound by restricting the neural network model is also standard (Schmidt-Hieber (2020)). In the revised manuscript, we showed $O_p(1/\min(n^{1/2}_nu,n^{1/2}_de))$ rate in Theorem 2 for the nonparametric regression estimator. To discuss the optimality more formally, we should consider mini-max optimality, but such a discussion is beyond the scope of this study.
>
> ---
>
> **Q1c** $\kappa_1$ and $\kappa_2$ has non-polynomial dependency on $C,B_p,L,B_{W_j}$?
>
> **A1c** In Remark 6 of the revised manuscript, we showed the explicit form of the generalization error bound of Theorem 1. As shown there, the dependency is not polynomial. The dependence of $B_p,L,B_{W_j}$ is similar to existing results of classification error bound with Lipschitz function (Corollary 15 of [Bartlett et al. (2003)). Compared with existing studies, this study's only new element is $C$, but the dependency is at most linear.
>
> ---
>
> **Q1d** "Computation" would be an important aspect of the proposed heuristic. ...
>
> **A1d** We adopted the heuristic because existing studies (Kiryo et al. (2017)) adopted it. It slightly improves performance, but the effect is limited. In the revised manuscript, we showed the experiments without the heuristic in Appendix G.1.3. We recommend practitioners to use the heuristic, but readers concerning the effect do not have to use it. In addition, existing studies also have not shown the theoretical analysis, and we also consider that deriving the theoretical analysis of the heuristic is beyond the scope of this study.
>
> ---
>
> **Q2** the writing needs improvement.
>
> **Q2a** the assumptions, theorems, and definitions require more explanation.
>
> **A2a** In the revised manuscript, we added more explanations for the main proposal, especially Section 3.2.
>
> ---
>
> **Q2b** the instantiation of the "nnBR divergence" doesn't seem important ...
>
> **A2b** The part is required by other readers. For such readers, the part is helpful.
>
> ---
>
> **Q2c** claims that "the above theoretical guarantee already suffices for the basic justification of D3RE." ...
>
> **A2c** As answered in A1c, the dependency of sample sizes and Lipschitz constant of the generalization error bound is similar to that of the classification problem (Bartlett et al. (2003)). In the revised manuscript, we explained the relationship between PU learning and the BR divergence. From a regression perspective, in general, $O_p(n^{-1/2})$ convergence rate is desirable, and we derived the rate in Theorem 2 of the revised manuscript. We also explained these points more clearly.
>
> ---
>
> **Q2d** any prior or related method(s) satisfy a bound similar to that in Theorem 1...
>
> **A2d** As mentioned above, the optimality depends on the purpose. When considering the classification problem, the bound of Theorem 1 is similar to Bartlett et al. (2002) and Kiryo et al. (2017). When considering the regression problem, the bound of Theorem 1 is similar to Kanamori et al. (2012), which shows a standard convergence rate of nonparametric regression.
>
> ---
>
> **Q2e** the paper lacks references in a few places...
>
> **A2e** In the previous manuscript, we have already shown reference in Introduction and Appendix "DETAILS OF EXISTING METHODS FOR DRE." We only cited the two papers at that part you pointed out because we considered that citing other studies again is a bit redundant. In the revised manuscript, we show citations as possible.

---

### Official Review · AnonReviewer2 · 2020-10-31
**Delay with my review**

**Rating:** 6
**Confidence:** 2

**Review:**

SUMMARY:
This work introduces a new family of Bregman divergences for density ratio estimation with flexible models that aims at solving the train-loss hacking problem. The contribution is foremost theoretical, but includes an experimental validation on benchmark problems.

STRENGTHS:
- This work introduces a new family of non-negative Bregman divergences.
- The proposed estimator is theoretical justified.
- Experiments show convincing results over the original Bregman divergences (at least for LSIF and PU).
- The paper is well written, although quite difficult to follow without some theoretical background.

WEAKNESSES:
- In Section 3.1, it said the loss functions for other machine learning tasks such as classification are lower-bounded, hence (as far I understand) they do not suffer from the train-loss hacking problem. Shall I therefore understand that estimating the density ratio through classification, using the cross-entropy (BR_BKL), does not suffer from this the train-loss hacking issue? In this case, what is the advantage of using nnBR_BKL over BR_BKL?
- The experiments do not include BR_BKL nor nnBR_BKL, which I would have found quite interesting for the reason mentioned above.
- While appreciate the estimation error bound for D3RE, can you comment on the error bound for the original Bregman divergences? How much does nnBR improve over it?
- How tight is the error bound for D3RE?
- Some recent applications of density ratio estimation that could have been worth mentioning as well as likelihood-free inference approaches based on likelihood ratios.

DISCLAIMER:
Due to the critical conditions in my country due to the pandemic, I must admit that I did not verify the mathematical developments. I am sorry for the limited quality of my review.

---

> ### Author Response · Authors · 2020-11-22
> **Response to AnonReviewer2**
>
> Thank you for reviewing our manuscript in spite of the conditions in your country. Following your constructive comments, we updated the manuscript.
> Our replies are listed below.
>
> ---
>
> **Q1** In Section 3.1, it said the loss functions for other machine learning tasks such as classification are lower-bounded, hence they do not suffer from the train-loss hacking problem. Shall I therefore understand that estimating the density ratio through classification, using the cross-entropy (BR_BKL), does not suffer from this the train-loss hacking issue? In this case, what is the advantage of using nnBR_BKL over BR_BKL?
>
> **A1** Thank you for pointing this out. We found that the BKL is more robust to the train-loss hacking problem, unlike that the other methods, LSIF, UKL, and PU, because the loss is lower bounded. **Following your advice, we also show the experimental result of BKL and nnBR-BKL (nnBR-BKL is also shown in the previous manuscript) in Figures 4 and 7.** From the experimental result, we still confirmed that the proposed non-negative correction improves the performance of BKL. We consider this for the BKL loss function,  both $- \hat{\mathbb{E}}_{de}[\log(\frac{1}{1+r(X)})]$ and $- \mathbb{\hat{E}}_nu[\log(\frac{r(X)}{1+r(X)})]$ can go to $0$ by over-fitting (train-loss hacking). This phenomenon is similar to the problem that when using deep neural networks, estimated conditional probability $p(y\mid x)$ (ex. output of the network with the sigmoid function) tends to stick to $0$ or $1$. We consider that this is also a reason why "bounded LSIF" fails in the experiments with the image dataset (Section 5). In the previous manuscript, when we referred to the train-loss hacking, we mainly meant that the loss function goes to infinite negative. However, even when the loss function does not go to infinite negative, we can still observe such an over-fitting. We conclude that the former problem (loss going to infinite negative) is a more serious version of the latter problem (loss sticking to lower bound). Both problems are serious, but the proposed method solves them. In the revised manuscript, we clarified this point.
>
> ---
>
> **Q2** The experiments do not include BR_BKL nor nnBR_BKL, which I would have found quite interesting for the reason mentioned above.
>
> **A2** In Figures 4 and 7 of the revised manuscript, we showed the experimental results of BKL and UKL. Both methods perform badly owing to the train-loss hacking.
>
> ---
>
> **Q3** While appreciate the estimation error bound for D3RE, can you comment on the error bound for the original BR divergences? How much does nnBR improve over it?
>
> **A3** The nnBR estimation bound does not improve that of the original BR divergence. The implication of the estimation bound is
> - the correction is empirically required for practical DRE because the existing methods fail by the train-loss hacking;
> - Owing to the empirical risk correction, the estimation error bound becomes looser than that of the original BR divergence;
> - However, from a classification perspective, we can obtain a bound similar to Corollary 15 of [Bartlett et al. (2003)](https://www.jmlr.org/papers/volume3/bartlett02a/bartlett02a.pdf) and Kiryo et al. (2017). From regression perspective, we can obtain the $O_p(\min(n_{nu},n_{de})^{-1/2})$, which is a desirable convergence rate of nonparametric regression.
>
> ---
>
> **Q4** How tight is the D3RE error bound?
> Some recent applications of density ratio estimation that could have been worth mentioning (likelihood-free inference approaches).
>
> **A4** The tightness depends on the application. When considering the classification, we want to minimize the classification risk by using flexible models, such as ResNet. When considering the regression, we are more interested in the convergence rate of the estimator even if restricting the model. In the revised manuscript, we separately show Theorem 1 and 2.
> - Theorem 1 corresponds to a classification application, such as PU learning and anomaly detection. The generalization error bound of the BR divergence is similar to the standard one of the classification (Bartlett et al. (2002)).
> - Theorem 2 corresponds to a regression application. The $L^2$ norm bound is $O_p(\min(n_{nu},n_{de})^{-1/2})$, which is standard in nonparametric regression. For instance, as we discussed with reviewer 3, many semiparametric models use density ratio. In econometric literature, Chernozhukov et al. (2016) proposed using machine learning models such as neural networks for a semiparametric model. Chernozhukov et al. (2016) mentioned that for many cases, we want $O_p(\min(n_{nu},n_{de})^{-1/4})$ convergence rate, which is satisfied by our method.
>
> In the next revision, we will also mention likelihood-free inference approaches.

---

### Author Response · Authors · 2020-11-25
**Response to all reviewers: Paper update**

We would like to again thank our reviewers for their valuable comments.
We have updated our manuscript based on their feedback.

We summarize the major updates as follows:

**1.** We explained the motivation of our problem in the Introduction in more detail. In particular, we show an instance of the train-loss hacking problem discussed in Kiryo et al. (2017).

**2.** We elaborated on the derivation of the non-negative Bregman divergence in Section 3.2.

**3.** In the previous manuscript, we mainly cited the related work in the first sections and Appendix. Following the comments of Reviewers 1 and 3, we cited them again in the related part of the main text. In addition, we fixed the typo of Nguyen et al. (2011).

**4.** In the previous manuscript, we mainly showed the generalization error bound of the BR divergence. For the nonparametric regression convergence rate, we only mentioned that "proof techniques such as local Rademacher complexity" would derive a tight bound. In the revised manuscript, under an assumption that the neural networks are multilayer perceptron with ReLU activation function, we derived an instance of such a tight $L^2$ distance bound for nonparametric regression in Theorem 2.

**5.** In the revised manuscript, we explained how Theorem 1 is standard. The generalization error bound depends on several elements, such as the Lipschitz constant. The dependency is similar to the one shown in [Corollary 15 of Bartlett et al. (2003)](https://www.jmlr.org/papers/volume3/bartlett02a/bartlett02a.pdf)), which is a generalization error bound for classification problem with Lipschitz function.

**6.** For Theorem 2, we explained how the result is useful. The convergence rate close to $O_p(\min(n_{nu},n_{de})^{-1/2})$ under some conditions is standard in nonparametric regression. In the context of density ratio estimation, [Kanamori et al. (2012)](https://link.springer.com/article/10.1007/s10994-011-5266-3) showed a similar convergence rate for a case with reproducing kernel Hilbert space in their Theorem 2.

**7.** There is a more rigorous discussion of the optimality of the theoretical result, such as minimax optimality. However, we consider such a discussion is not the scope of this paper.

**8.** A convergence rate shown in Theorem 2 is useful in various applications. As mentioned in Appendix H, [Chernozhukov et al. (2016a)](https://economics.mit.edu/files/12538) showed that for a sample size $n$, the $O_p(n^{-1/4})$ $L^2$ convergence rate of the density ratio suffices to show the asymptotic normality of a semiparametric estimator. For instance, [Uehara et al. (2020)](https://papers.nips.cc/paper/2020/hash/0084ae4bc24c0795d1e6a4f58444d39b-Abstract.html) showed the asymptotic normality of off-policy evaluation estimator using the density ratio estimated by Kanamori et al. (2012)' method with such a convergence rate. We explained these points in Appendix H of the revised manuscript.

**9.** On the other hand, we consider that several methods simultaneously estimating the semiparametric estimator and the density ratio are not directly related to our work because it is not appropriate to use machine learning methods with the approach. For instance, Imai and Ratkovic (2014) adopted this simultaneous approach. This problem comes from the non-Donsker property of the estimators obtained from machine learning algorithms, such as Ridge regression, Lasso regression, random forest, and neural networks. It is not easy to show the asymptotic normality of a semiparametric estimator with such a non-Donsker estimator. This problem is discussed by various existing studies of econometrics and statistics, such as [Klaassen et al. (1998)](https://www.springer.com/gp/book/9780387984735), [Zheng and van der Laan (2011)](https://link.springer.com/chapter/10.1007/978-1-4419-9782-1_27), and Chernozhukov et al. (2016a). Therefore, in the recent semiparametric inference, including econometrics, it is more typical to use two-step methods under about $O_p(n^{-1/4})$ convergence rate when using machine learning algorithms. See also [Chernozhukov (2016b)](https://arxiv.org/abs/1608.00033). Therefore, we did not cite Imai and Ratkovic (2014) in the previous manuscript. Following Reviewer 3's comment, we explained why it is not appropriate to use the methods in Appendix H.

**10.** To support the theoretical results, we added numerical experiments using artificially generated datasets in Appendix E, where we know the true density ratio function.

**11.** We use the gradient ascent/descent heuristic following the existing studies. The heuristic improves the performance, but our proposed method works well even without the heuristic. For readers concerning the effect, we added Appendix G.1.3, an experimental result without the heuristic.

---

### Decision · Program_Chairs · 2021-01-07
**Final Decision**

**Decision:**

Reject

**Comment:**

This paper discusses the likelihood ratio estimation using the Bregman divergence.  The authors consider the 'train-loss hacking', which is an overfitting issue causing minus infinity for the divergence.   They introduce non-negative correction for the divergence under the assumption that we have knowledge on the upper bound of the density ratio.  Some theoretical results on the convergence rate are given.  The proposed method shows favorable performance on outlier detection and covariate shift adaptation.

The proposed non-negative modification of Bregman divergence is a reasonalbe solution to the important problem of density ratio estimation.  The experimental results as well as theoretical justfication make this work solid.  However, there are some concerns also.  The paper assumes knowledge on the upper bound of density ratio and uses a related parameter essentially in the method.  Assuming the upper bound  is a long standing problem in estimating density ratio, and it is in practice not necesssarily easy to obtain.  Also, there is a room on improvements on readability.

Although this paper has some signicance on the topic, it does not reach the acceptance threshold unfortunately because of the high competition in this years' ICLR.